# Evolutions of the seasonal anticyclonic circulation around the Qingdao cold water mass in the China marginal sea and its mechanism

Lin Lin[1, a, *], Hans von Storch[1], Yang Ding[2]

[1]Institute of Coastal Systems, Helmholtz Zentrum Hereon, Geesthacht, 21502, Germany
[2]Frontier Science Center for Deep Ocean Multispheres and Earth System (FDOMES) and Physical Oceanography Laboratory, Ocean University of China, Qingdao, 266100, China
[a]Currently at Max Planck Institute for Meteorology, Hamburg, 20146, Germany

*Correspondence to*: Lin Lin (lin.lin@mpimet.mpg.de).

**Abstract.** The circulation structure surrounding the Qingdao cold water mass in 2019 was investigated using three-
dimensional ensembles of numerical simulations. This study reveals that a cold pool appears in early spring and reaches its peak in late May, and this pool is accompanied by local seasonal anticyclonic circulation. Momentum diagnostics reveal that vertical friction cannot be ignored because of the shallow topography and surface wind stress; as a result, the geostrophic balance is inapplicable in the Qingdao cold water mass region. Seasonal circulation mostly results from the balance of the pressure gradient, Coriolis, and vertical friction forces. The no-tide and no-wind numerical simulation results suggest that
when the tidal forcing is turned off, unrealistically strong currents appear and are caused by the decrease in vertical friction in the no-tide simulation. Moreover, the direction of the eastern side of the anticyclonic circulation is reversed. Furthermore, the seasonal southwesterly monsoon contributes to the magnitude of the anticyclonic circulation, especially in the western portion of the anticyclonic circulation, by piling up the surface water eastward and further changing barotropic pressure gradient force. Additionally, upwelling occurs vertically around the Qingdao cold water mass and is influenced by tidal and wind forcings.
The wind forcing affects upwelling (especially in the western part), which can be explained by Ekman pumping; the tides contribute to upwelling in the eastern part by reshaping the thermocline and further changing the barotropic and baroclinic pressure gradient force. Ensemble simulations are used, and the t-test reveals that 51.29% (89.68%) of points, difference between the control run and the ensemble experiments without tidal forcing (/without wind forcings) is statistically significant.


## 1. Introduction

The Qingdao cold water mass was first identified in 1959 and occurs in deep waters below 25 m in the southeastern offshore region of Qingdao during the spring season (the position of Qingdao and the position mentioned in this paper are shown in Fig. 1) (Ho et al. 1959). This cold water mass is an isolated cold pool characterized by low temperature (6.5–10.0℃), moderate salinity (31.5–32.5), and seasonal variation (Zheng and Zhang 1983). This cold water mass has an important influence on aquaculture in China since it is situated near the Chinese coastline (cunyi Zhang 1986; Y. Zhang and Geng 1989; Zheng and Zhang 1983). The temperature, salinity, and circulation distribution of the Qingdao cold water mass play important roles in the ecosystem, such as in primary production (Q. Wei et al. 2019). Numerous researchers have conducted surveys on the origin and characteristics of the Qingdao cold water mass and reported that it forms in late April, peaks in May, and gradually moves eastward, with both the temperature and salinity increasing. After June, this cold water mass gradually disappears as it merges with the bottom layer of the Yellow Sea (Diao 2015; Yu et al. 2006; Q. Zhang, Yang, and Cheng 1994; Q. Zhang et al. 2004). Although the Qingdao cold water mass lifetime is relatively short, it is considered an independent water mass because of its unique patterns of both formation and evolution. On the basis of observational data, Zhang et al. (2002) analyzed the source and temperature–salinity characteristics of the Qingdao cold water mass, indicating that it formed in the local environment by the Bohai Sea coastal current bypassing the Shandong Peninsula. This cold water mass is characterized by low temperature, moderate salinity, and high dissolved oxygen. Zhang et al. (2004) analyzed the intensity variations in the Qingdao cold water mass. Their results suggest that it disappears gradually in June and July. However, some studies have reported the presence of the Qingdao cold water mass in August (J. Xia and Xiong 2013). Yu et al. (2005, 2006) analyzed the relationship between the Qingdao cold water mass and the Yellow Sea cold water mass and reported that in June, the Qingdao cold water mass was already on the edge of the Yellow Sea cold water mass; it was no longer a local independent water mass but rather a local cold center within the Yellow Sea cold water mass. Huang et al. (2019) used numerical simulations to reveal that the cold and moderate-salinity water in the offshore area between Qingdao and Shidao in the southwestern Yellow Sea was an early form of the Qingdao cold water mass. Additionally, the strong horizontal temperature gradient is the thermodynamic mechanism by which the Qingdao cold water mass forms (Zhang et al., 2016). On the basis of the previous studies mentioned above, conclusions have been drawn regarding the seasonal variations and origins of the Qingdao cold water mass. Previous work has demonstrated that a mesoscale anticyclone exists near the Qingdao coast, and its position is close to the Qingdao cold water mass (Huang, Chen, and Lin 2019; F. Zhang, Mao, and Leng 1987). However, it is unknown whether the formation of this anticyclonic circulation is related to the Qingdao cold water mass.

There are other cold water masses in the Bohai and Yellow Seas, the marginal seas along China, such as the cold pool in the Bohai Sea (Liu et al. 2003; Wan et al. 2004; Zhou et al. 2017), and the Yellow Sea cold water mass (Ho et al. 1959; Hur, Jacobs, and Teague 2000; H. Wei et al. 2010; Yuan et al. 2013). The special circulation structures around the Bohai cold water mass and Yellow Sea cold water mass have been well described in previous studies (Wang et al. 2014; C. Xia et al. 2006; Zhou et al. 2017; Zhu and Wu 2018), but the anticyclone current field analysis near the Qingdao cold water mass still needs

to be investigated. In this work, we investigate the following questions: (1) Does such a seasonal anticyclonic circulation fit the geostrophic balance? (2) What factors influence the morphology, magnitude, and position of the seasonal anticyclonic circulation horizontally and vertically?

The paper is organized as follows. The model configuration and model design are described in Section 2. In this section, the concept and logic of ensemble simulations and the need to consider the internal variability (for the definition of internal variability, see subsection 2.2) of the marginal sea are described. Section 3 shows the results of the ensemble mean of the control simulations with respect to the simulated temperature, salinity, circulation structure, momentum balance in the Qingdao cold water mass, and the potential factors on the horizontal circulation pattern. In Section 4, the upwelling structures of the Qingdao cold water mass, statistic test, the perspective and implications and limitation of this paper are discussed. Finally, conclusions are given in Section 5.

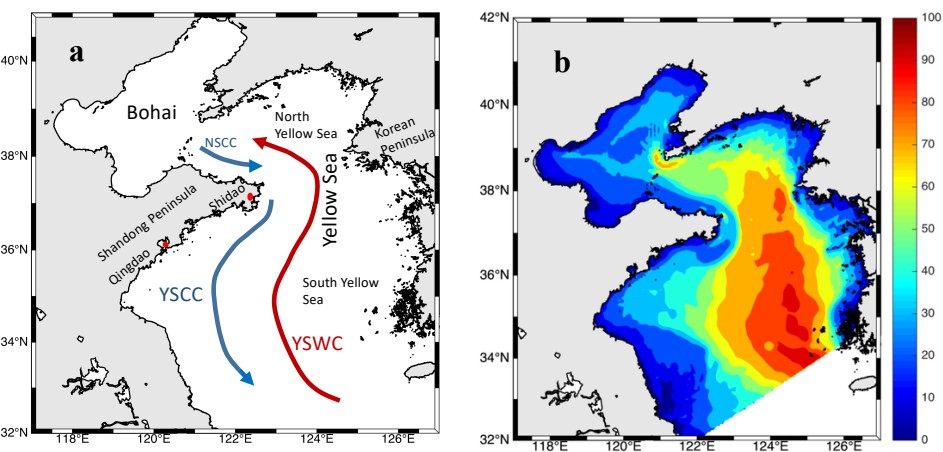

**Figure 1. (a) Maps of the Bohai Sea, Yellow Sea (North and South Yellow Seas), Shandong Peninsula, Korean Peninsula, Shidao, Qingdao, North Shandong Coastal Current (NSCC), and Yellow Sea Coastal Current (YSCC) and (b) the topography and model computation domain.**

## 2. Methods and model

### 2.1 Model configuration

In this work, we used the three-dimensional unstructured grid finite volume coastal ocean model (FVCOM) (Chen, Liu, and Beardsley 2003; Chen, Beardsley, and Cowles 2006). The horizontal grid resolution is approximately 4 and 8 km in the Bohai and Yellow Seas, respectively. Vertically, the model comprises 30 sigma layers. The vertical and horizontal diffusion coefficients were calculated using the Mellor and Yamada level 2.5 (MY-2.5) turbulent closure model (Mellor and Yamada

1982) and the Smagorinsky eddy parameterization method, respectively. The model domain includes the Bohai and Yellow Seas, ranging from 31.885°N–40.942°N and 117.572°E–126.915°E, respectively (Fig. 1b). The open boundary across the Yellow Sea extends eastward from Qidong in China to the southern tip of the Korean Peninsula, and radiation boundary conditions were used. We use the tidal elevation to introduce the effect of the tides. The tidal elevation forcing comprises eight major tidal components ($M_2$, $S_2$, $N_2$ $K_2$, $K_1$, $O_1$, $P_1$, and $Q_1$) derived from the TPXO8 database (Egbert and Erofeeva 2002).

For the control run, the six-hourly surface forcing data utilized in this study are from the National Centers for Environmental Prediction (NCEP) Climate Forecast System Version 2 (CFSv2) data, featuring a global resolution of 0.2° × 0.2° (Saha et al. 2014). This dataset encompasses parameters such as sea surface temperature, cloud cover, air pressure, wind, specific humidity, evaporation, precipitation, and heat flux. The initial data of the model are introduced in the following ensemble experimental design section. The Huanghe, Huaihe, and Haihe are considered, and the river discharges are from the "China Sediment

Bulletin (2019)." The model-simulated elevation, temperature, salinity, and circulation patterns were validated by information in Huang et al. (2019) and Lin et al. (2022, 2023). This study focuses directly on the analysis and discussion of the Qingdao cold water mass.

## 2.2 Ensemble experimental design

An ensemble of control runs was first used to analyze the temperature, salinity, and velocity around the Qingdao cold water mass. Additionally, two further ensembles of model experiments were conducted. The first experiment was performed with tidal forcing turned off to examine the effect of tides on the Qingdao cold water mass, and the second experiment was performed with wind forcing turned off (zero wind stress) to test the effect of the wind. The model output interval was 3 h. A more thorough description of the ensemble simulation configuration can be found in Lin et al. (2022).

Ensemble simulations were conducted for the control run, no-tide run, and no-wind run. Each ensemble simulation comprises four numerical simulation members. In each ensemble, the initial conditions of the four simulations were taken from Nov. 1st of the 2017, Jan. 1st of the 2018, Mar. 1st of the 2018, and Nov. 1st of the 2018 of the separate 10-year climatological simulation. Note that an independent 10-year simulation was conducted to generate slightly different but generally consistent initial conditions for the ensemble simulation. The model starting time of the 10-year climatological simulation was Nov. 1st, 2008,

and the model ending time was Nov. 1st, 2018. The climatological forcing for the climatological run was a smooth annual cycle without interannual variations based on National Centers for Environmental Prediction (NCEP) Climate Forecast System Version 2 (CFSv2) data. For the control, no-tide run, and no-wind run, the results for 2019 were used for further analysis. The model ending dates of the control run, no-tide, and no-wind runs were Dec. 31st, 2019.

    The motivation for using ensemble simulations is based on the observation (Lin et al. 2022; 2023; Penduff et al. 2019) that

deviations from within the ensemble members if the ensemble simulations are conducted with the same model configuration

except for slight perturbations in the initial conditions. In other words, if only one numerical simulation exists, the model output will be a mix of "signal" (external forcing) and random effects, which refer to variation among the ensemble means. The randomness reflects the internal variability. Some spatial features are not repeatable in other ensemble members, even though the model configurations are the same. Averaging across ensemble simulations efficiently reduces the impacts of randomness. Therefore, in Sections 3 and 4, the ensemble means were further analyzed. An ensemble simulation with slightly different initial conditions is an approach to analyzing ocean internal variability. Internal variability is defined as the part that cannot be linked directly to the external forcing but is caused by unforced variability generated within a system. The unforced variability is unprovoked and chaotic. Since this paper focuses on the external forcing imprint on the seasonal anticyclonic circulation around the Qingdao cold water mass, more details about the concept of internal variability can be found in previous studies (Tang, von Storch, and Chen 2020; Lin et al. 2023). For the ensemble simulation configuration in this study, the tradition of generating an ensemble simulation with slightly different initial conditions was adopted (Penduff et al. 2019). The deviations existing between ensemble members due to randomness necessitate testing whether the differences between the ensemble means of the control run and the no-tide run (/no-wind run) may be caused by external forcings (tidal or wind forcings) or randomness. A proper way to do so is hypothesis testing with the null hypothesis: "external forcing has no effect". If this null hypothesis is rejected with a sufficiently small risk, then a valid conclusion is that an external factor has an effect and plays an active role. Here, a t test was performed for the ensemble monthly mean for May.

## 3. Results

### 3.1 Temperature and Salinity of the Qingdao Cold Water Mass

The temperature distribution at a depth of 25 m around the Qingdao cold water mass is shown in Fig. A1(before the formation of Qingdao cold water mass) and Fig. 2a–c. In March, the cold water carried by the Bohai coastal current approached the area around the Shandong Peninsula, gradually evolving to become the Qingdao cold water mass. However, the isotherms did not close, and the Qingdao cold water mass structure did not form at this time (Fig. A1). In April, a low-temperature tongue enveloped by an 8°C isotherm occurs north of 35°45′N and east of 121°45′E, accompanied by a horizontal temperature gradient. During this period, the high-temperature water transported by the northwest path of the Yellow Sea warm current remained in the western region of the South Yellow Sea, forming an apparent oceanfront with the Qingdao cold water mass (Fig. 2a). In May, the location of the cold water mass center does not change at a depth of 25 m (Fig. 2b). Combined with the salinity results shown in Fig. 2d–f, it is clear that the temperature and salinity are relatively low around the Shandong Peninsula. Moderately saline water intrudes toward the south, and the intrusion direction is northeast–southwest, which is consistent with previous research results (Diao 2015). The low-temperature and moderate-salinity water tongue along the northeast of the Shandong Peninsula merges with the ambient warm water left by the Yellow Sea cold water mass beginning in winter. In June (Fig. 2c), intensified solar radiation increases surface temperature, thereby accelerating the convergence of warm and cold

waters (Huang, Chen, and Lin 2019), combined with the downward transfer of net surface heat flux and the horizontal heat input, this process contributes to the disappearance of the Qingdao Cold Water Mass (Q. Zhang et al. 2016). In summary, the Qingdao cold water mass starts to merge in April, continues to develop in May, is accompanied by a strong horizontal temperature gradient, and disappears in June. The cold water mass center is at 122.20°–122.40°E, 36°–36.15°N, and the shape of the cold water mass has a northeast–southwest orientation.

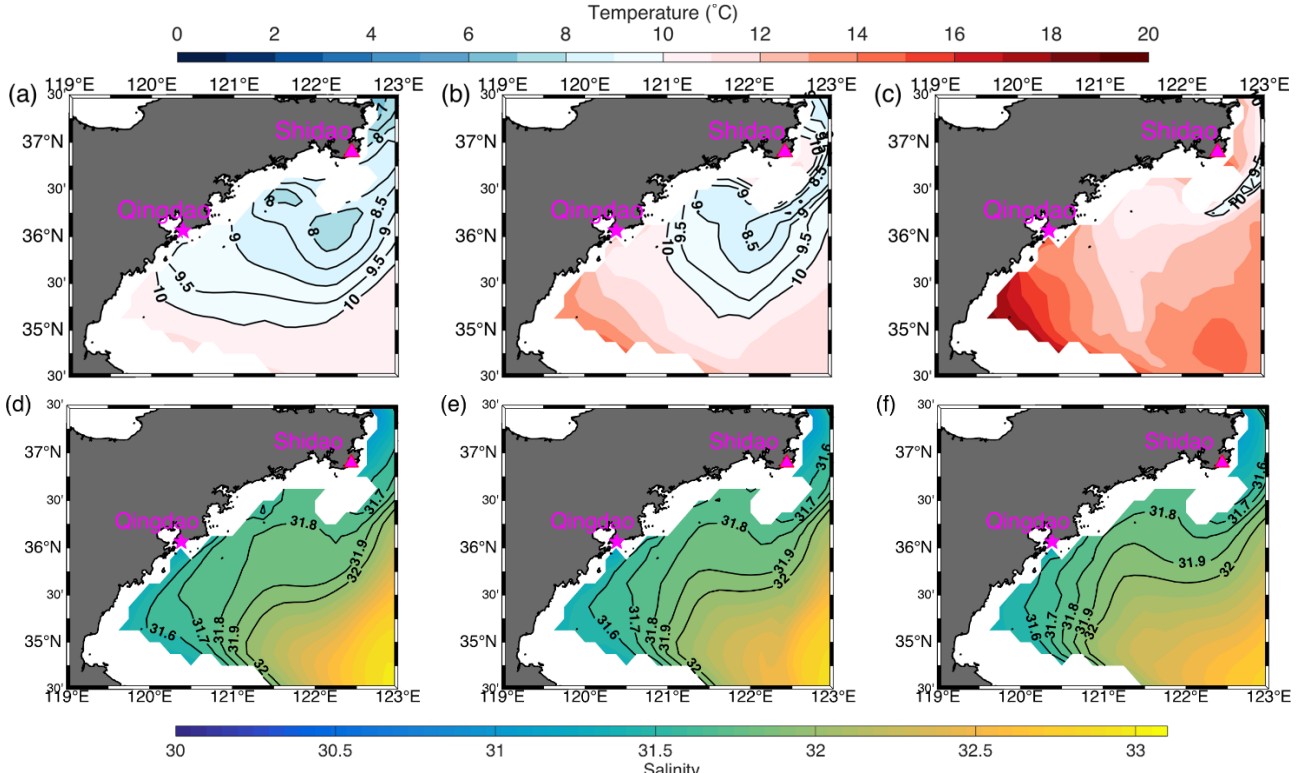

Figure 2. Monthly mean temperature and salinity evolution of the Qingdao cold water mass from April to June at a depth of 25 m according to the ensemble mean of the control runs. Figs. 2a, b, and c show the temperature distributions of the peaks of the Qingdao cold water mass (in April), the gradual disappearance (in May), and the complete disappearance (in June), respectively. Figs. 2d–f are the same as Figs. 2a–c but for salinity. To clearly present the temperature distribution, the isotherms between 8°C and 10°C are marked with 0.5°C intervals.

## 3.2 Circulation Pattern

A seasonal anticyclonic structure exists near the Qingdao cold water mass. Fig. 3 shows the circulation at a depth of 25 m from April to June. In April, the seasonal anticyclonic structure is not closed. In May, a southwesterly monsoon prevails near the Shandong Peninsula (Fig. A3), and a northward current can be seen along the Shandong Peninsula coastline, meeting the

southward current between 122°E and 123°E. In May, the seasonal anticyclonic structure nearly closes, and the northeastward flow on the west side of the anticyclonic structure is stronger than the southwestward current on the east side. The center position of the seasonal anticyclonic circulation is 122.35°-122.45°E, 35.45-°35.55°E, and the velocity at the center of the anticyclonic circulation is very slow. In June, the seasonal anticyclonic circulation disappears when the southward current is strengthened. The above simulation results are consistent with those of previous studies (Xu and Zhao 1999; F. Zhang, Mao,

and Leng 1987).

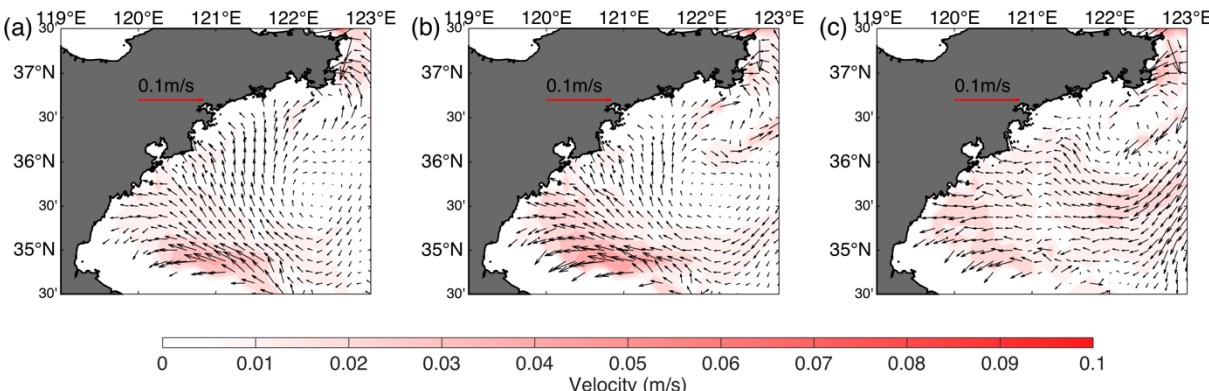

**Figure 3. Horizontal circulation distributions around the Qingdao cold water mass (25 m layer) in (a) April, (b) May, and (c) June.**

The circulation patterns at 25, 30, 35, and 40 m are shown in Fig. 4. The anticyclonic structure can still be observed at depths

of 25 and 30 m. However, at the 35-m layer, the anticyclonic circulation does not close, as the current in the eastern part becomes weaker than that at a 25-m depth. At depths exceeding 35 m, the anticyclonic structure no longer exists. The patterns of the circulation in different layers are different, reflecting the baroclinic feature of the current fields.

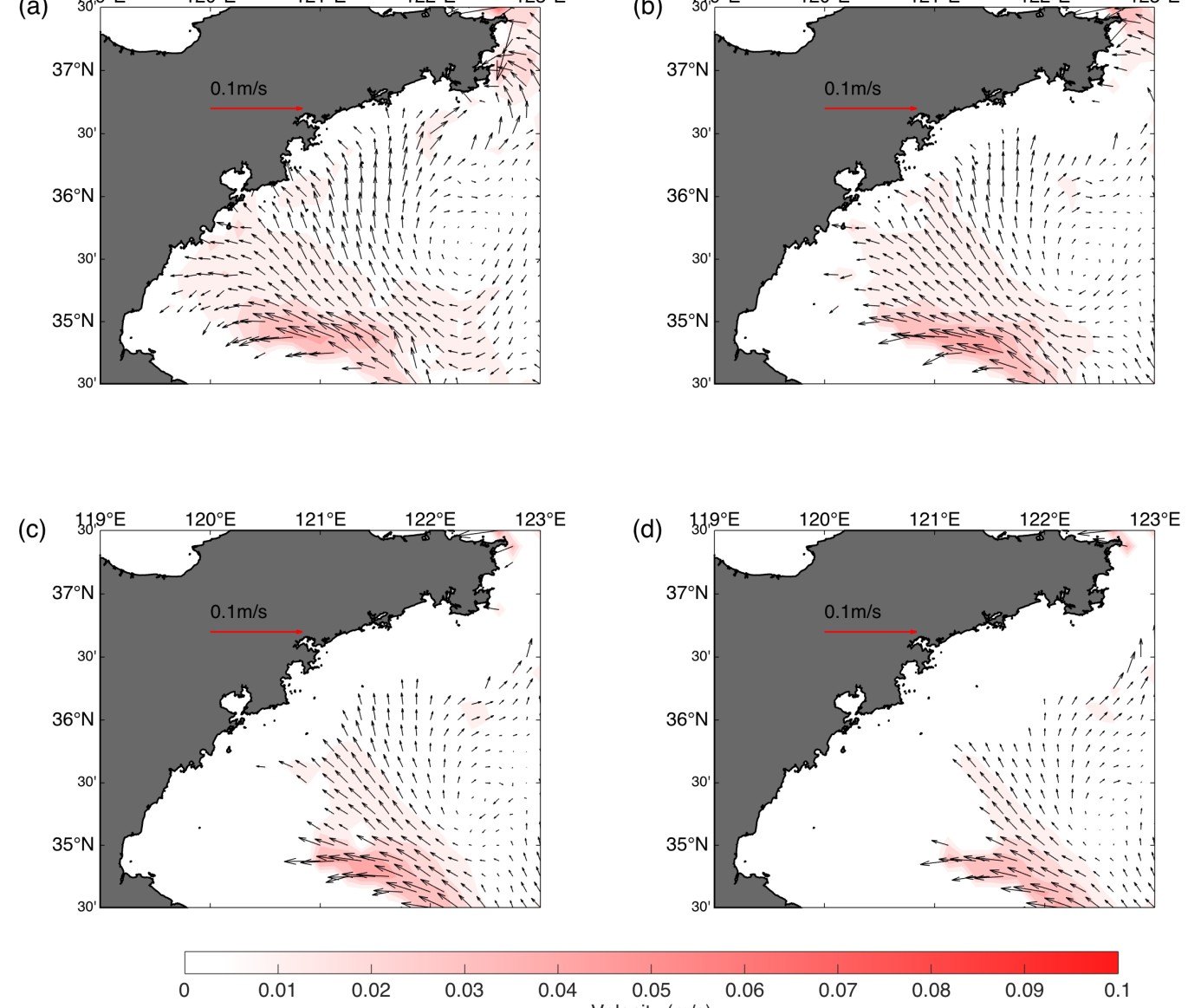

Figure 4. Circulation structure of the Qingdao cold water mass in May at depths of (a) 25, (b) 30, (c) 35, and (d) 40 m.

## 3.3 Momentum Balance

$$\frac{\partial u}{\partial t} = fv - \left(u\frac{\partial u}{\partial x} + v\frac{\partial u}{\partial y}\right) - w\frac{\partial u}{\partial z} - g\frac{\partial \zeta}{\partial x} - \frac{1}{\rho}\frac{\partial}{\partial x}\int_z^\zeta \rho g dz + \frac{\partial}{\partial z}\left(K_m\frac{\partial u}{\partial z}\right) + F_u \qquad (1)$$

$$\frac{\partial v}{\partial t} = -fu - \left(u\frac{\partial v}{\partial x} + v\frac{\partial v}{\partial y}\right) - w\frac{\partial v}{\partial z} - g\frac{\partial \zeta}{\partial y} - \frac{1}{\rho}\frac{\partial}{\partial y}\int_z^\zeta \rho g dz + \frac{\partial}{\partial z}\left(K_m\frac{\partial v}{\partial z}\right) + F_v \qquad (2)$$

$$\frac{\partial w}{\partial t} = -\left(u\frac{\partial w}{\partial x} + v\frac{\partial w}{\partial y}\right) - w\frac{\partial w}{\partial z} - \frac{\rho g}{\rho_0} + \frac{\partial}{\partial z}\left(K_m\frac{\partial w}{\partial z}\right) + F_w \qquad (3)$$

To show the general pattern of the fundamental forces around the Qingdao cold water mass anticyclonic circulation region, a momentum balance diagnostic was conducted (see Equations (1) – (3)). Taking the zonal direction as an example (Equation (1)), the total pressure gradient term is $-g\frac{\partial \zeta}{\partial x} - \frac{1}{\rho}\frac{\partial}{\partial x}\int_z^\zeta \rho g dz$, which comprises the barotropic pressure gradient force $-g\frac{\partial \zeta}{\partial x}$ induced by sea level and the baroclinic pressure gradient force $-\frac{1}{\rho}\frac{\partial}{\partial x}\int_z^\zeta \rho g dz$ induced by density; the Coriolis force fv; the vertical friction term $\frac{\partial}{\partial z}\left(K_m\frac{\partial u}{\partial z}\right)$; the local velocity time variation term $\frac{\partial u}{\partial t}$; the horizontal advection term $-(u\frac{\partial u}{\partial x} + v\frac{\partial u}{\partial y})$; the

vertical advection term $-w\frac{\partial u}{\partial z}$; and the horizontal friction term $F_u$.

In the 25-m layers (Fig. 5), the terms for the barotropic pressure gradient, baroclinic pressure gradient, Coriolis force, and vertical friction are dominant, and the other terms are far smaller. The barotropic gradient force exceeds the baroclinic pressure gradient force and is balanced by the joint effects of the opposing baroclinic pressure gradient force, Coriolis force, and vertical friction force (the sum of the barotropic gradient force, baroclinic pressure gradient force, Coriolis force, and vertical friction

force is shown in Fig. A2). Because the vertical friction can not be ignored in such shallow area, a geostrophic balance no longer exists, especially in areas close to the coast.

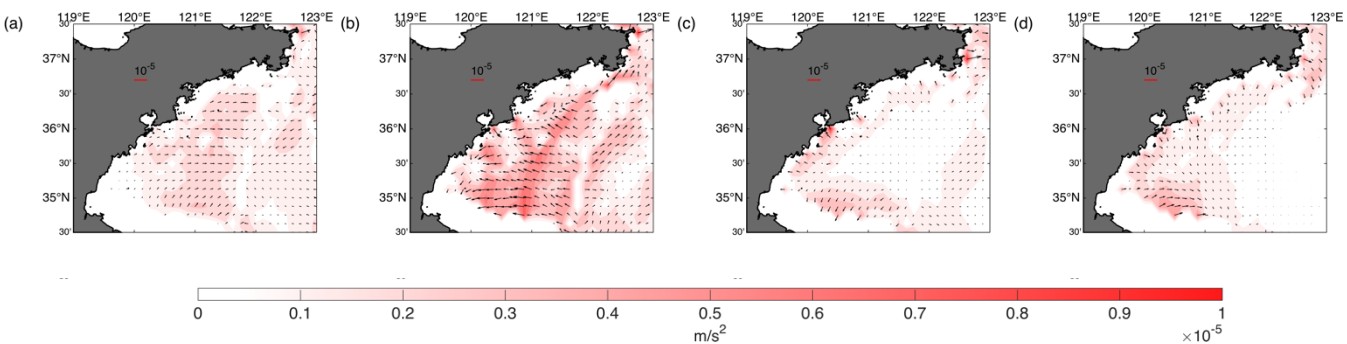

**Figure 5. Horizontal distributions of momentum terms (monthly mean for May) at depths of (a–d) 25 m: (a) Coriolis force ($fv\vec{\imath} -$**

**$fu\vec{\jmath}$), (b) barotropic pressure gradient force ($-g\frac{\partial \zeta}{\partial x}\vec{\imath} - g\frac{\partial \zeta}{\partial y}\vec{\jmath}$), (c) baroclinic pressure gradient force ($-\frac{1}{\rho}\frac{\partial}{\partial x}\int_z^\zeta \rho g dz\,\vec{\imath} -$ $\frac{1}{\rho}\frac{\partial}{\partial y}\int_z^\zeta \rho g dz$), and (d) vertical friction force ($\frac{\partial}{\partial z}\left(K_m\frac{\partial u}{\partial z}\right)\vec{\imath} + \frac{\partial}{\partial z}\left(K_m\frac{\partial v}{\partial z}\right)\vec{\jmath}$). The color represents the magnitude of the momentum term.**

### 3.4 Tidal Effects

Tides play a significant role in the circulation processes of regional seas, contributing substantial amplitude, momentum, and energy flux. Early studies (B.H. Choi 1980; Byung Ho Choi, Eum, and Woo 2003; Moon, Hirose, and Yoon 2009; C. Xia et al. 2006) have shown that tidal characteristics in the Bohai and Yellow Seas are highly complex and play a significant role in

local circulation and modulate dissipation, vertical mixing, and tidal energy. Owing to the large input of tidal energy and the combined effects of bottom friction related to shallow shelf topography, the dissipation of $M_2$ tidal energy (the principal lunar

semidiurnal constituent) in shallow regions reaches 180 GW, accounting for approximately 11.1% of the global total (i.e., 2 TW). No-tide simulation was conducted to analyze the impacts of tidal forcing and wind forcing on anticyclonic circulation.

In this subsection, the tidal effects on the seasonal anticyclonic circulation horizontally are discussed. The circulation pattern of the no-tide experiment in May at a depth of 25 m and the current deviations between the no-tide experiment and the control run (Fig. 6). Northward currents are present on the western side (to the west of $122°E$) in the control run and the no-tide

experiment, but the magnitudes are obviously greater in the no-tide experiment. At the center of the anticyclonic circulation, changes in the current are not evident. When the tidal forcing is turned off, the eastern side of the anticyclonic circulation direction reverses, changing from a southward current (in the control run, see Fig. 3b) to a northward current. The clockwise circulation pattern disappears in the no-tide experiment.

The momentum diagnostic of the no-tide experiments shows that the velocity decreases compared with that of the control run.

When tidal forcing is not considered, the vertical friction along the bottom layer becomes much smaller than that in the control run. Without tidal forcing east of $122°E$, the magnitude of the Coriolis terms increases (Fig. 7a), which is combined with a change in the circulation direction, indicating the significance of the tides around the anticyclonic circulation. In addition, the magnitude of the current in the entire region obviously increases, which is unrealistic, indicating that the dissipation and friction caused by tidal forcing are important for modulating circulation in anticyclonic areas.

The eastern side of anticyclonic circulation direction reverses when the tidal forcing is turned off is related to the tidal forcing impact on the general circulation in the whole Yellow Sea area (Fig. 8). When the tidal forcing is considered, it is dominated by a basin scale anticlockwise gyre in the Yellow Sea at the depth of 25 m. In the eastern part of the Yellow Sea, the mainly current directions are northward. In the western part of the Yellow Sea, the North Shandong Coastal Current (NSCC), and Yellow Sea Coastal Current (YSCC) are present, and the main current direction are southward, except the west of $122°E$,

where is the compensation for the surface layer wind transport (which will discuss in the next subsection). Such observation agreed with previous observations and numerical results (Bearsley et al. 1992, Yangagi and Takashi, 1993, Xia 2006). The northward flow in the eastern part of the southern Yellow Sea is jet-like flow, which is different from the southward flow in the west portion of the southern Yellow Sea, which is much weaker and broader. However, when tidal forcing is removed, the overall circulation shifts to a clockwise gyre (Fig. 8). This large-scale circulation change influences the local flow structure

around the Qingdao Cold Water Mass. Specifically, the reversal of the eastern branch of the anticyclonic circulation (122.5°-123°E 35°-35.5°N) results from the adjustment of the broader-scale Yellow Sea gyre, highlighting the significant role of background circulation in shaping the local current system.

The reason of background anticlockwise circulation in the Yellow Sea when considering tidal forcing has been discussed in previous research: (1) in the middle layer of the Yellow Sea (10-40 m) the flow is quasi-geostrophic. During spring and summer time, strong tidal mixing over the western and central parts of the shelf leads to the formation of a pronounced tidal front, which separates the well-mixed coastal waters from the stratified offshore waters. This front induces strong lateral density gradients, which in turn generate geostrophic currents around the front. This front-associated baroclinic structure promotes the formation of a basin-scale cyclonic (anticlockwise) gyre (C. Xia et al. 2006). (2) the Eulerian residual tidal currents form a cyclonic gyre, implying the Eulerian residual tidal currents also strengthen the cyclonic circulation in the upper layers (C. Xia et al. 2006).

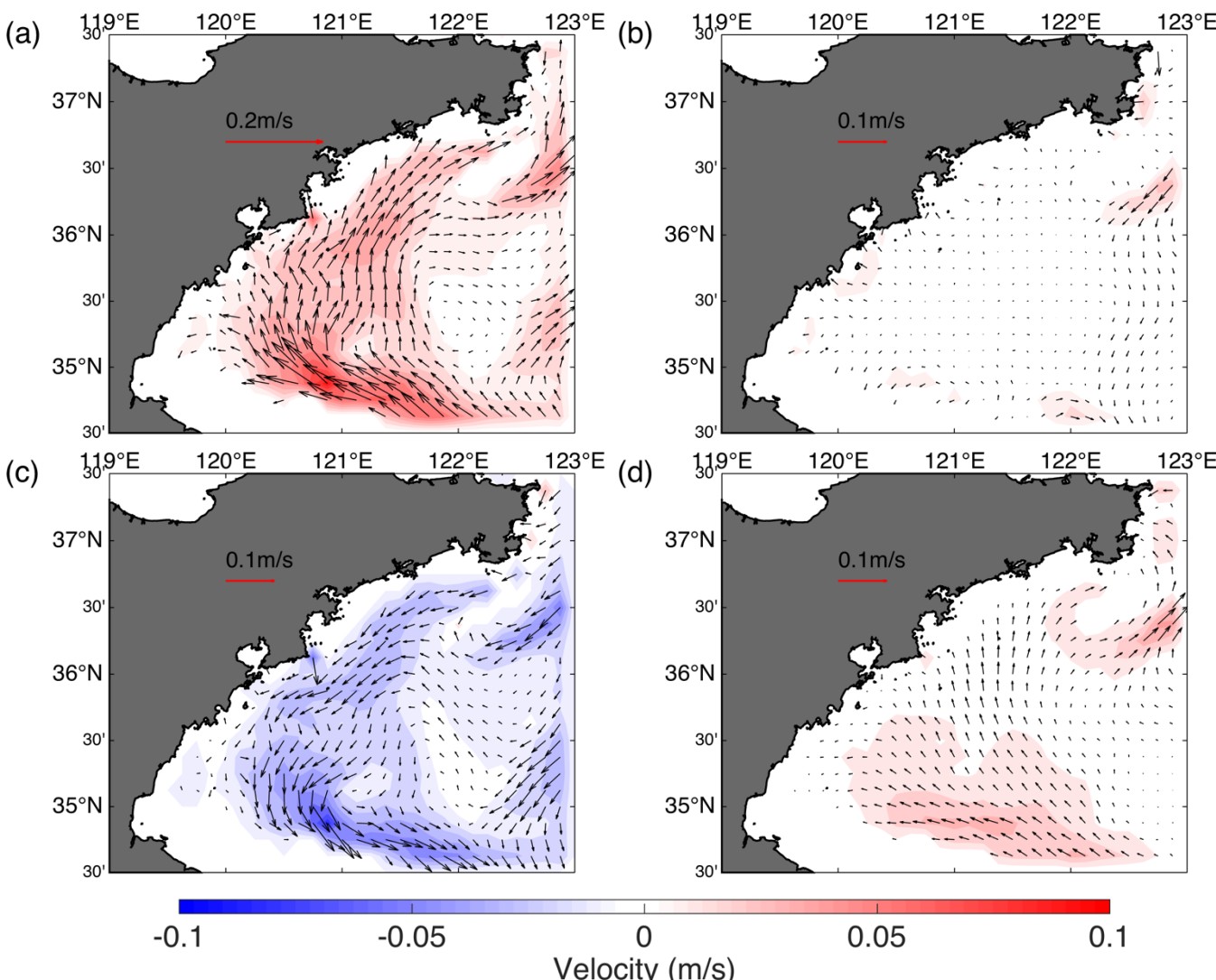

**Figure 6. Impacts of tidal forcing and wind forcings on anticyclonic circulation in May at 25 m (control run minus the no-wind/ no-tide experiment). (a) Circulation pattern of the no-tide experiment, (b) circulation pattern of the no-wind experiment, (c) difference between the control run experiment and no-tide run, and (d) difference between the control run and the no-wind experiment. The color represents the magnitude, and the negative value means that the magnitude of velocity is smaller compared with the no-tide run.**

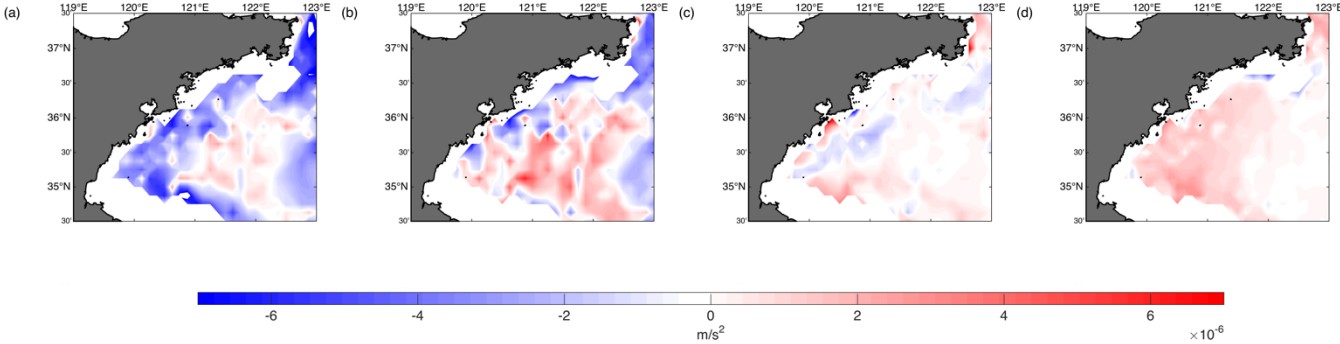

**Figure 7. At 25 m, the magnitude difference of momentum balance terms between the control run and the no-tide experiment (control run minus no-tide experiment). (a) Coriolis force, (b) barotropic pressure gradient force, (c) baroclinic pressure gradient force, and (d) vertical friction term. The results are the monthly means for May.**

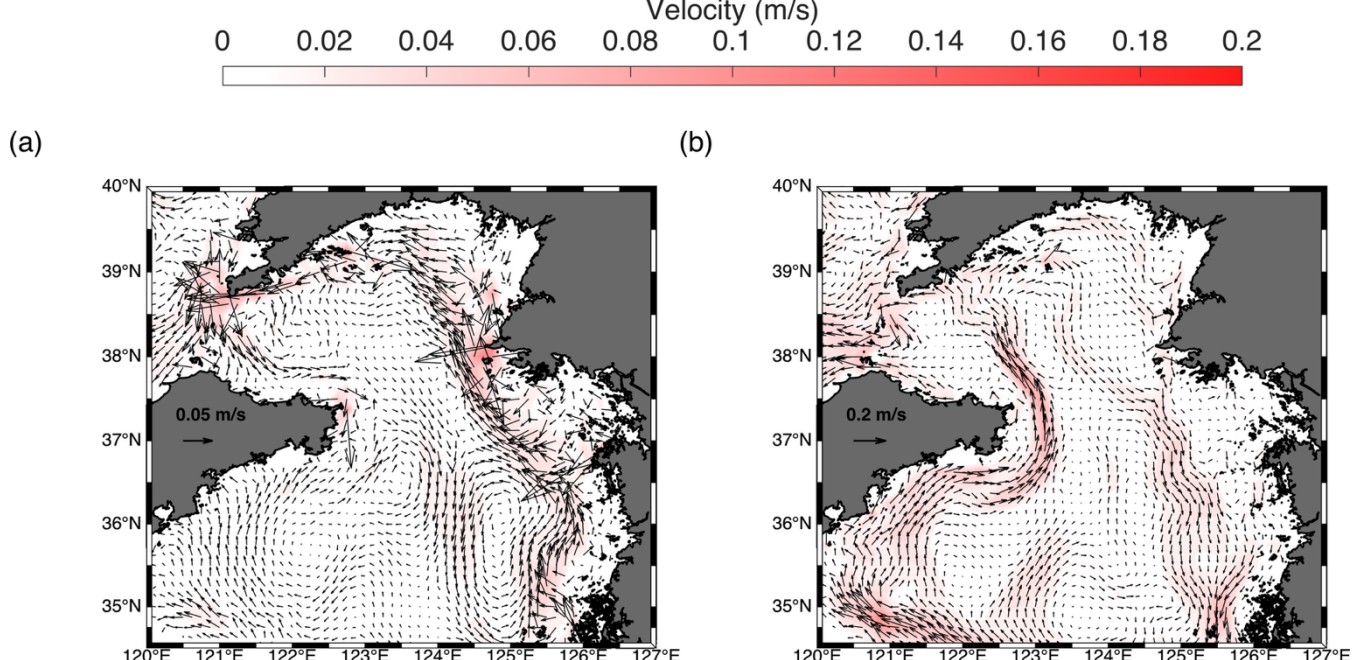

**Figure 8. Impacts of tidal forcing on circulation in the Yellow Sea in May at 25 m (a) Circulation pattern of the control run, (b) circulation pattern of the no-tide experiment.**

## 3.5 Wind Effects

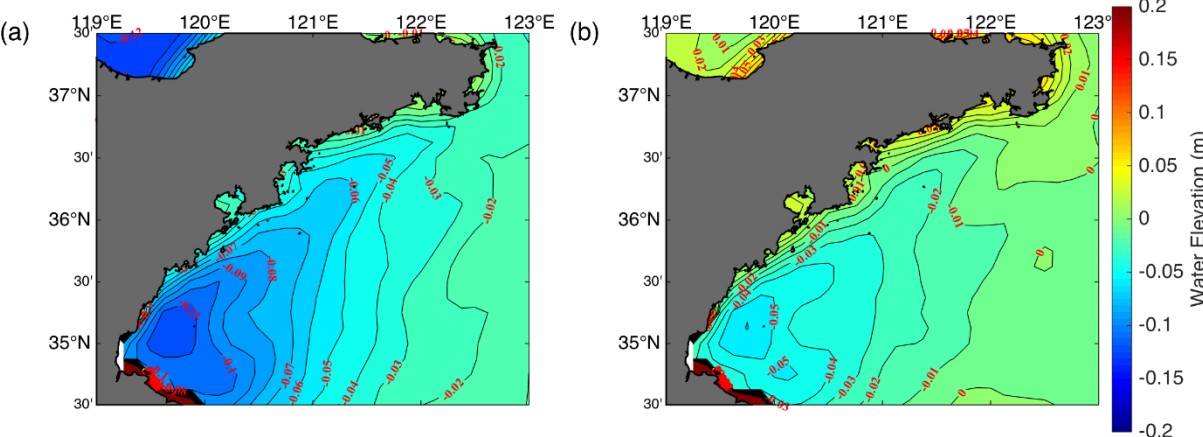

**Figure 9. Surface elevation distributions of the (a) control run and (b) no-wind experiment. Monthly mean for May.**

Xu and Zhao (1999) demonstrated the effect of wind on the seasonal anticyclonic circulation around the Qingdao cold water mass using a two-dimensional numerical model, but a more thorough discussion is needed. Therefore, a no-wind experiment was conducted to examine the effect of wind on the seasonal anticyclonic circulation structure. The results show that wind is a dominant driving force for clockwise circulation. The general magnitude of the current weakens, particularly on the western side of the clockwise circulation (northward current); by contrast, the influence of wind on the eastern side of the circulation is minor, as shown in Figs. 6b and 6d. The direction and magnitude of the eastern side are similar to those of the control run.

The wind effect on the seasonal anticyclonic circulation around the Qingdao cold water mass can be understood that the sea water is piled up eastward in Fig. 9 (121-123°E) under the impact of the southwesterly monsoon. Hence, the landward current (westward) at the western side of the seasonal anticyclonic circulation around the Qingdao cold water mass is primarily driven by the pressure gradient force, with minimal influence from other forcing terms. To prove this, we have examined the surface elevation fields from both the control and no-wind experiments (Fig. 9). The results clearly show that under southwesterly wind forcing (control run), surface waters accumulate to the east. In contrast, the no-wind experiment shows a much flatter sea surface. This indicates that the wind-induced water piling on the eastern side is responsible for the enhanced barotropic pressure gradient force. Further diagnostics of the momentum balance around the seasonal anticyclonic circulation confirm that the cross-shelf barotropic pressure gradient force is the dominant term driving the bottom flow along the coast.

Fig. 10 compares the spatial distributions of the momentum terms in the control (top row) and no-wind (bottom row) experiments. These terms are derived from the x-direction momentum equation, which is particularly relevant since the barotropic pressure gradient force is predominantly directed in zonal direction. In the control experiment, a strong barotropic pressure gradient is established, corresponding to a pronounced landward current at the western side of the anticyclonic circulation around the Qingdao cold water mass. By contrast, the no-wind experiment shows much weaker pressure gradients in the same region. The Coriolis, baroclinic pressure gradient, and vertical friction terms are relatively weak in both

experiments. These results support the interpretation that the southwest monsoon induces water piling eastward, thereby establishing a stronger barotropic pressure gradient that dominates the bottom-layer momentum balance and drives the landward current in the control run on the western side of the seasonal anticyclonic circulation, compared with no-wind run.

 Beyond the local coastal dynamics around the Qingdao Cold Water Mass, previous studies have demonstrated that the wind forcing also plays a key role in shaping the large-scale summertime circulation in the Yellow Sea. For example, a wave–tide–

circulation coupled model is used to reveal a three-dimensional structure characterized by wind-driven surface flows and compensating near-bottom currents (C. Xia et al. 2006). This basin-scale mechanism is consistent with our findings, suggesting that the landward bottom flow observed in our control experiment is not only locally forced but also part of a broader wind-driven circulation system across the Yellow Sea.

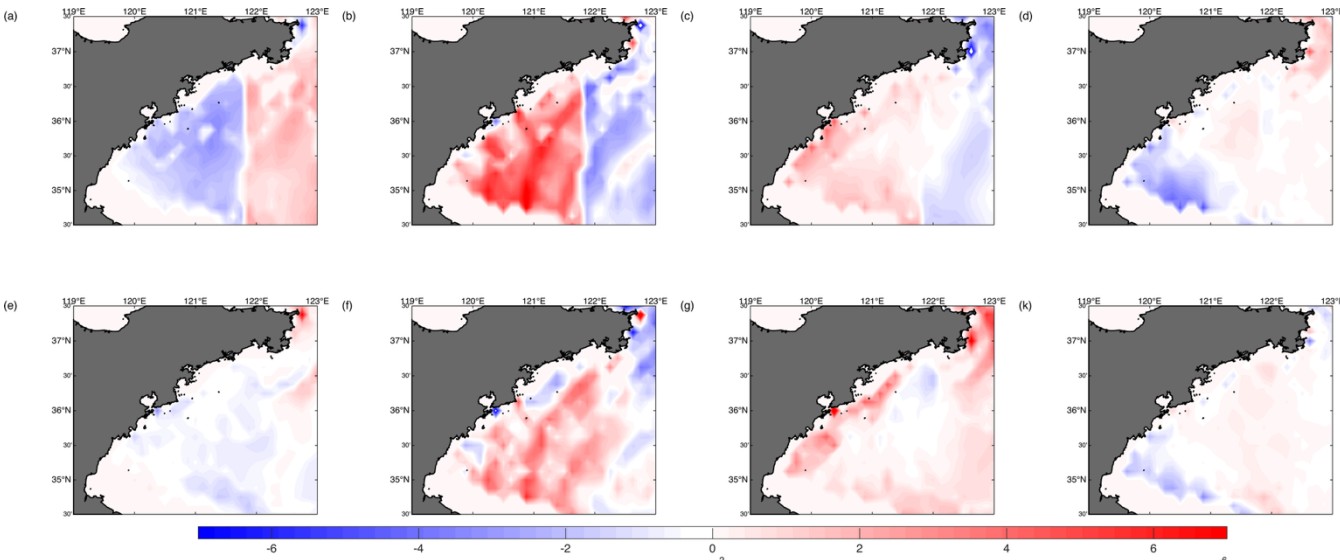

**Figure 10. At 25 m, the zonal momentum balance terms of the control run (a–d) and the no-wind experiment (e-k). (a, e) Coriolis force $\mathbf{fv}$, (b, f) barotropic pressure gradient force $-g\frac{\partial \zeta}{\partial x}\vec{\iota}$, (c, g) baroclinic pressure gradient force $-\frac{1}{\rho}\frac{\partial}{\partial x}\int_z^\zeta \rho g dz$, and (d, k) vertical friction term $\frac{\partial}{\partial z}\left(K_m \frac{\partial u}{\partial z}\right)$. The results are the monthly means for May. The positive values represent the eastward direction.**

## 4. Discussion

### 4.1 Existence of Upwelling Near the Qingdao Cold Water Mass

In this paper, we focus on the upwelling near the Qingdao cold water mass. Fig. 11a shows the vertical circulation structure in May at 35.625°N, which is around the horizontal center of the seasonal anticyclonic circulation. Obvious upwellings occur

near the frontal zones (122.375–122.5°E and 122.625–123°E) on the eastern and western sides of the Qingdao cold water mass, respectively. The seawater lifted by upwelling from the western and eastern sides of the cold pool converges toward the middle

(122.5°E), making the vertical velocity surface layer at 122.5°E relatively small compared with that of the surrounding water.

We hypothesize that tidal forcing and wind forcing both contribute to upwelling around the Qingdao cold water mass. Wind forcings influence upwelling through Ekman transport. When tidal forcing is considered, the thermocline is reshaped, accompanied by changes in the barotropic and baroclinic pressure gradient term, further strengthening the upwelling intensity. To test our hypothesis, we further analyzed the vertical circulation along the 35.625°N profile in May of the control run, the

no-tide experiment, and the no-wind experiment (Fig. 11). Fig. 11 shows that in the control, an eastward flow occurs west of 122.50°E (red rectangle), and westward flow occurs east of 122.50°E (blue rectangle) above 25 m in the vertical direction.

A comparison of Fig. 11a and 11c reveals that upwelling cycles occur in the red and blue rectangles in the control run. However, in the no-wind experiment (Fig. 11c), the magnitude of upwelling cycling obviously decreases. The upwelling caused by the wind forcing can be explained by the Ekman theory because of the predominant southwesterly monsoon.

In the no-tide experiment (Fig. 11b), the magnitude of upwelling decreases as well, but mildly compared with no-wind experiment. In the following, we explain the tidal impacts on upwelling by analyzing the tidal impacts on barotropic pressure gradient force, the tidal impacts on the thermocline and further varying the baroclinic pressure gradient force.

A comparative analysis between the tidal and non-tidal experiments reveals that both the barotropic and baroclinic pressure gradient forces are significantly intensified when tides are included (Fig. 12). The magnitude of the barotropic pressure gradient

force is larger than the baroclinic counterpart. This enhancement in barotropic forcing leads to increased horizontal convergence in nearshore regions, which, through the continuity equation, results in intensified upward motion. Notably, at approximately 122.5°E, surface currents from both the west and east appear to converge (Fig. 11a), as indicated by opposing surface flow directions. This horizontal convergence is accompanied by a strong upward motion below, supporting the interpretation that tidal forcing enhances barotropic pressure gradients, leading to horizontal convergence and subsequent

upwelling in this region.

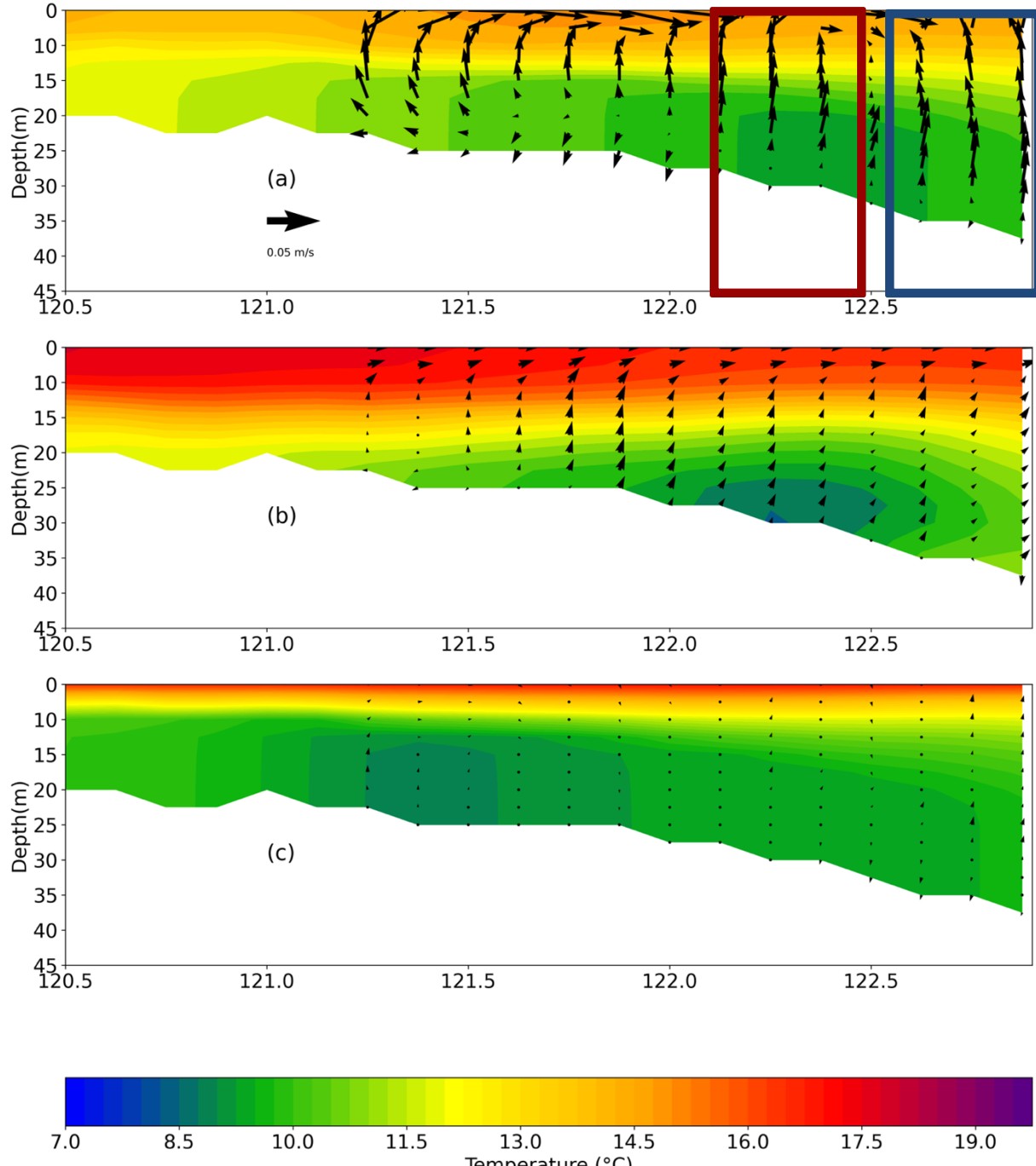

**Figure 11. Temperature (contours) and zonal–vertical current (arrows) along the 35.625°N profile in May in the ensemble means. For better visualization, the vertical velocity is multiplied by 1000 (unit: m/s). (a) Control run, (b) no-tide experiment, and (c) no-wind experiment. The red and blue boxes represent the frontal zones (122.375–122.5°E and 122.625–123°E) on the western and eastern sides of the Qingdao cold water mass.**

Since the baroclinic pressure gradient is influenced by the horizontal temperature distribution, before analyzing the tides impact on the baroclinic pressure gradient, we need to first look at the role of tides in shaping the characteristics of the thermocline, including the width of the front and the maximum temperature gradient. The front serves as a narrow transitional zone that separates stratified deep water from well-mixed shallow water, resulting in temperature (or density) differences across it. Owing to the mixing effects induced by tidal forces at the seafloor, isotherms near the bottom are predominantly orthogonal to the seabed topography at depths below 20 meters. Furthermore, in the control run, the front is both compressed and elevated. This study defines the front as the region where the magnitude of the horizontal and vertical temperature gradient ($\sqrt{\left(\frac{dT}{dx}\right)^2 + \left(\frac{dT}{dy}\right)^2}$) exceeds $1°C/m$. Under tidal forcing, the front is observed to exist within a depth range of approximately 2–20 m, whereas in the absence of tidal influences, the front extends from 2–25 m. A wider front correlates with a reduced intensity, indicating that the front is weaker in simulations without tidal effects. Additionally, the maximum vertical temperature gradient is greater in the control run ($1.78°C/m$) than in the no-tide simulation ($1.73°C/m$). Therefore, the front intensity decreases in the no-tide run, compared with the control run.

Furthermore, the tidal modulation of thermocline amplify the baroclinic pressure gradient force. Although this force is secondary in magnitude compared to the barotropic term (Fig. 12c), it plays a complementary role in shaping the vertical circulation. Figs. 12a and 12b show that around the upwelling area at the depth of 15 m, the lowest-order balance is between the total pressure gradient term and vertical friction term. Fig. 12d indicates that the baroclinic pressure gradient is larger in the control run compared with the no-tide simulation. Previous studies have investigated upwelling and the vertical secondary circulation in the Bohai and Yellow Seas and revealed that upwelling usually occurs in shallow areas along coasts, such as the Subei area and the region near the Korean Peninsula (Q. Wei et al. 2019; Lü et al. 2010). Lü et al. (2010) explained the tidal effect on upwelling, indicating that the front in the control run generates a relatively large baroclinic pressure gradient compared with when the tidal forcing is turned off, which further triggers distinct upwelling. In this work, we found that such an explanation can also explain the upwelling around the Qingdao cold water mass. Our results suggest that the enhanced upwelling is primarily driven by tidal amplification of the barotropic pressure gradient, which modifies the background current structure and promotes convergence-driven vertical motion. The front intensity enhanced in the control run, resulting in the increased baroclinic pressure gradient, which is a complementary reason for the upwelling intensitication when the tides are considered.

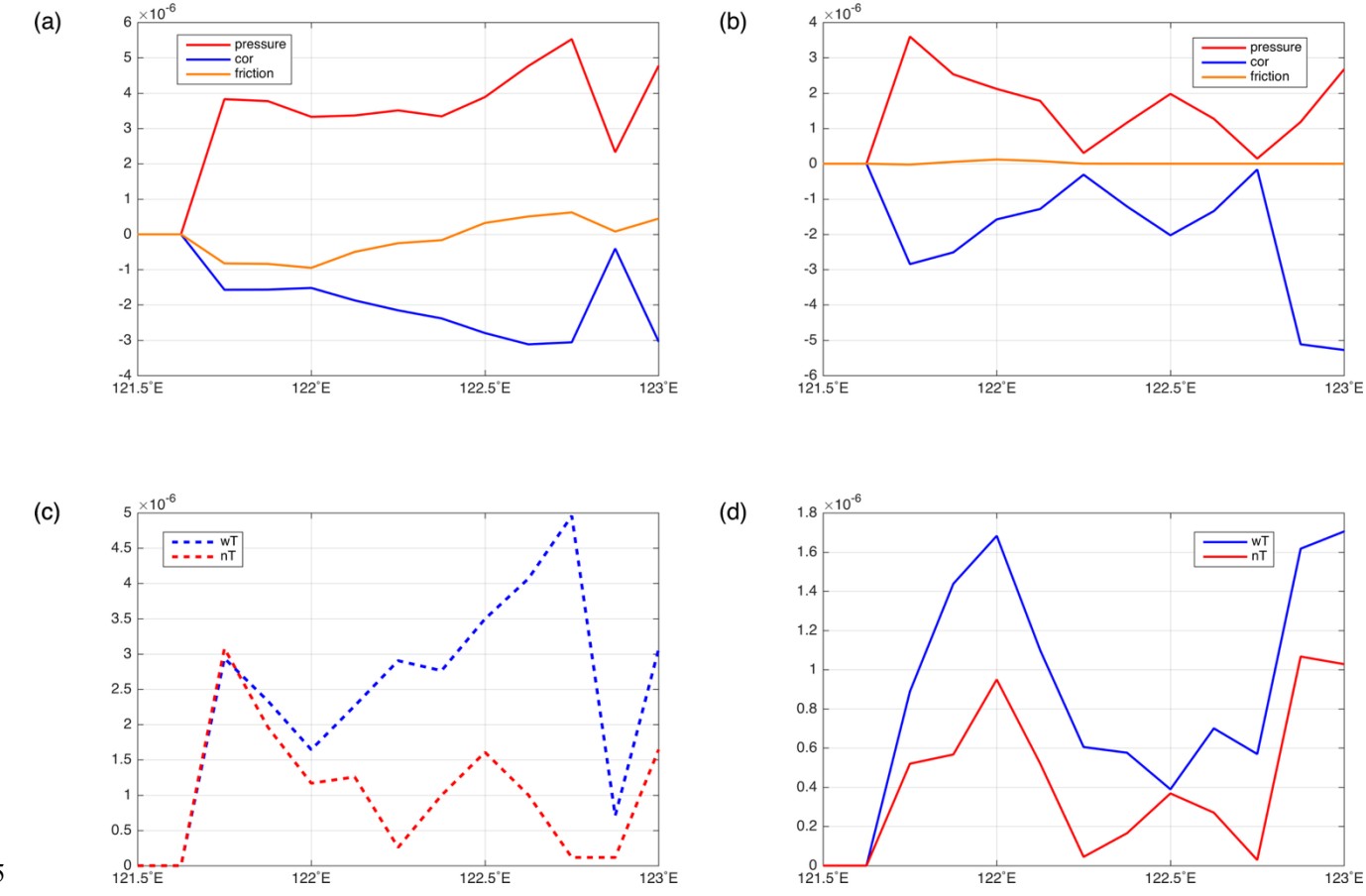

Figure. 12 Momentum terms in zonal direction at the depth of 15 m along 35.625°N for control run and no-tide experiment. (a) Balance among all terms in control run, (b) balance among all terms in the no-tide experiment, (c) the barotropic pressure gradient in control (blue dash line) and no-tide (red dash line) experiments, and (d) the baroclinic pressure gradient in control (blue solid line) and no-tide (red solid line) experiments.

## 4.2 Statistical Tests

As introduced in subsection 2.2, statistical tests have many applications in numerical climate simulations to separate and validate that the variation is caused by internal variability or changes in external forcing or parameterization (Conover 1999; Weisse, Heyen, and von Storch 2000; von Storch and Zwiers 1999; Livezey and Chen 1983; Zwiers and von Storch 1995). However, for the ocean regional simulation community, statistical tests are seldom applied to extinct the variation between internal and external generated.

In this study, a t-test was used to determine whether the differences between the ensemble means of the control run and the no-tide (no-wind run) may be caused by external forcings or can be explained by randomness. The results (Figs. 13 and 14) show the sensitivity of the formation of the spring cold water mass to the presence of tidal forcing and wind forcing. The grid points at which a t-test indicates that the effect of external forcing is significant are marked with a cross. Figs. 13 and 14

demonstrate that the difference between the control run and the no-tide ensemble (or the/no-wind ensemble) is significant, especially where the intraensemble deviations are large. The local percentages of local rejections are 51.29% and 89.68% for the no-wind and no-tide runs, respectively.

When such local tests are conducted, it is expected that even if the null hypothesis is valid, at approximately 5% of grid points, the null hypothesis is rejected (multiplicity of tests, cf. von Storch, and Zwiers, 1999). Since the rejection rate is itself a random

variable, the false rejection rate can be much larger, but more than 20% is very unlikely. A limitation of univariate tests, such as the t-test, is the "multiplicity of tests" problem. This occurs when multiple tests are conducted simultaneously across different points in a field without proper adjustment. This issue is discussed in standard textbooks like von Storch and Zwiers (1999), building on earlier work (H. V. Storch 1982; Livezey and Chen 1983).

The core argument is as follows: If a test has an acceptable false rejection rate (Type I error rate) of, say, 5% when the null

hypothesis is true, then repeating the test multiple times while the null hypothesis remains valid will still yield a 5% chance of false rejection in each individual test. On average, one would therefore expect false rejections in about 5% of the cases. However, since the rejection rate itself is a random variable, the actual proportion of false rejections may exceed 5%, though rates significantly higher than 20% are unlikely.

The situation becomes more complex when the tests are not independent—such as when analyzing spatially correlated data

from a grid. In such cases, nearby grid points exhibit stronger dependencies, meaning that false rejections are less likely to appear as isolated points and more likely to form spatially coherent patterns.

In our analysis, the observed rejection rate is substantially higher than 20% in both scenarios, suggesting that not all rejections can be attributed to the multiplicity effect. Instead, many of these rejections likely reflect genuine signals.


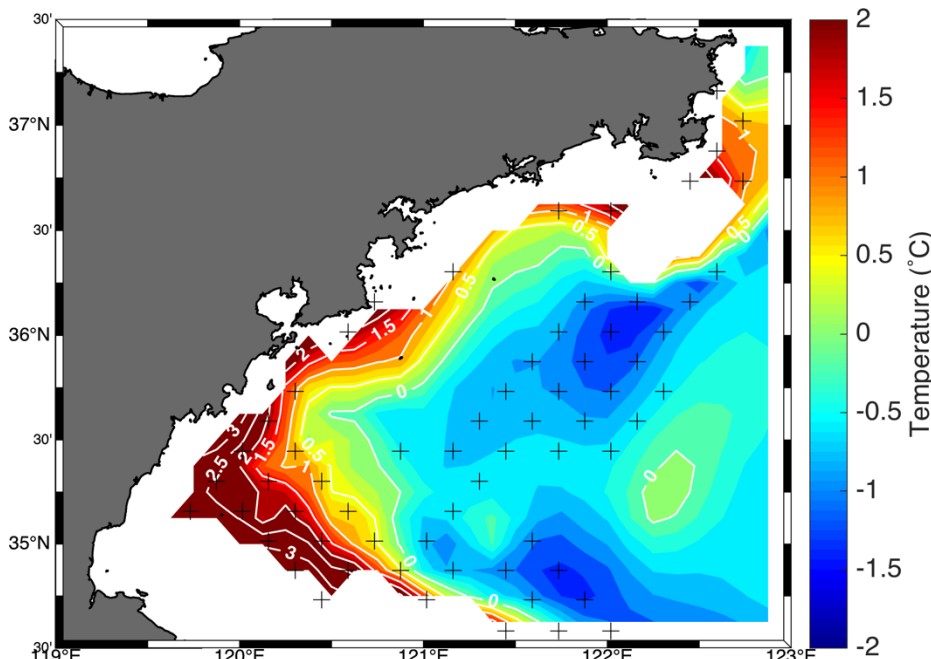

**Figure 13. Temperature difference between the ensemble means of the control runs and the runs without tidal forcing. The crosses represent the areas where the difference between the control run and the run without tidal forcing was significant at the 5% level.**


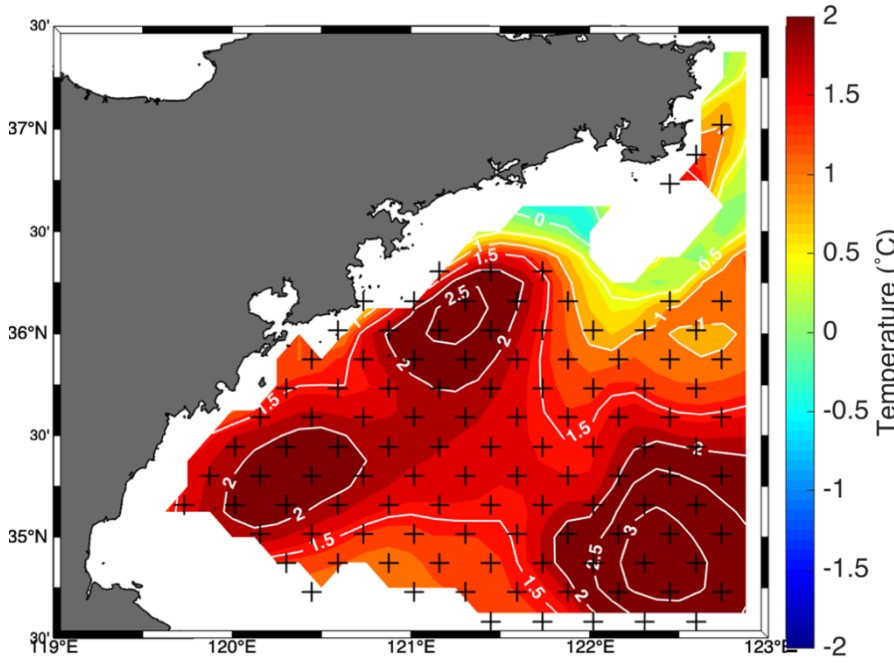

**Figure 14. Temperature difference between the ensemble means of the control runs and the runs without wind forcings. The crosses represent the areas where the difference between the control run and the run without tidal forcing was significant at the 5% level.**

### 4.3 Perspective and Implications of the Model Results

Previous studies have focused mainly on the seasonal variation in the formation of the Qingdao cold water mass and its mechanism (Ho et al. 1959; Q. Zhang et al. 2002; 2004; Yu et al. 2006; 2005; Zhang et al. 2016; Huang, Chen, and Lin 2019).

Our results confirm that the Qingdao Cold Water Mass begins forming in April, develops through May, and dissipates by June, consistent with previous studies (e.g., Diao 2015; Zhang et al. 2016; Huang et al. 2019). The observed northeast–southwest orientation, stable core location near 122.3°E, 36.1°N, and the interaction with Yellow Sea warm water and salinity patterns
further support earlier findings. The simulated seasonal anticyclonic circulation near the Qingdao cold water mass, including its formation in May and disappearance in June, is consistent with earlier findings (Xu and Zhao 1999; Zhang et al. 1987) in terms of both structure and timing.

Some previous studies have reported an anticyclonic circulation around the Qingdao cold water mass (F. Zhang, Mao, and Leng 1987). Early studies (B.H. Choi 1980; Byung Ho Choi, Eum, and Woo 2003; Moon, Hirose, and Yoon 2009; C. Xia et
al. 2006) have shown that tidal characteristics in the Bohai and Yellow Seas are highly complex and play a significant role in local large-scale circulation and modulate dissipation, vertical mixing, and tidal energy. However, tidal and wind forcings impact on the seasonal anticyclonic circulation around it has not been fully understood yet. The numerical results in this paper reveal that both tidal forcing and wind forcing play important roles in horizontal circulation and vertical upwellings. Based on the momentum balance, in this study, although there is a temperature gradient and a mild salinity gradient near the Qingdao
cold water mass, the resulting density gradient from these temperature and salinity variations does not directly drive the observed anticyclonic flow structure, as the geostrophic balance is not maintained.

Our findings are consistent with previous studies (e.g., Lü et al. 2010; Wei et al. 2019), which highlighted the role of tidal forcing in enhancing upwelling via baroclinic pressure gradients along coastal fronts in the Bohai and Yellow Seas. However, in contrast to those studies where baroclinic forcing dominated, our results suggest that in the region around the Qingdao Cold
Water Mass, both barotropic and baroclinic pressure gradient forces play important roles in driving upwelling, with the barotropic component being more pronounced.

In addition to the insight gained regarding the dynamics of the Qingdao water mass, another innovative aspect of this paper is demonstrating how to separate random variations and signals in marginal seas caused by the experimental setup. The method was introduced in the climate sciences in 1974 (Chervin, Gates, and Schneider 1974), but it is rarely used when evaluating the
results of numerical experiments in marginal seas.

Additionally, an ensemble simulation methodology has been applied in coastal ocean model simulations, which has rarely been applied and discussed in previous studies. Ensemble simulations can make the conclusion more robust and exclude the random phenomenon in a single simulation maximally.

**4.4 Limitations of the study**

In this study, the 2019 model results were used as an example. Huang et al. reported similar basic characteristics of the temperature and salinity of the Qingdao cold water mass in 2010 (2019). This study is limited to the 2019 simulation. Future work would investigate the interannual variation characteristics and mechanisms of the interannual variation in this anticyclonic circulation.

Also, this study focused on submesoscale horizontal and vertical circulation around the Qingdao cold water mass, and further
exploration of boundary layer theories is worthwhile since topography and bottom friction play a role in the coastal area. Future analyses will investigate the boundary layer variation caused by tidal forcing.

**5. Conclusions**

Based on numerical models, the main characteristics and related dynamic processes of the local seasonal anticyclonic circulation around the Qingdao cold water mass during spring and summer were analyzed in this study. The Qingdao cold
water mass emerges in April and is fully developed in May, combined with a seasonal anticyclonic circulation. The major conclusion of the manuscript is that: (1) The seasonal anticyclonic circulation does not satisify the geostrophic balance because of the vertical friction cannot be ignored (Fig. 5). (2) Horizontally, the tidal forcing and wind forcing play important roles in anticyclonic circulation structure (Fig. 6). The wind effects can be understood by that sea water s piled up under the impact of the southwesterly monsoon (Fig. 9), further trigger a strong barotropic pressure gradient (Fig. 10), corresponding to a
pronounced landward current at the western side of the anticyclonic circulation around the Qingdao cold water mass. When the tidal forcing is turned off, the eastern side of anticyclonic circulation direction reverses (Fig. 3b and 6a), because the background currents of the Yellow sea changed by the tidal forcing (Fig. 8). The magnitude of the current of the entire region increases unrealistically when the tidal forcing is excluded. (3) Vertically, the tidal forcing causes the intensities of the barotropic and baroclinic pressure gradient increase (Fig. 12), further trigerring upwelling; wind forcing contributes to Ekman
upwelling.

*Author contributions.* LL formulated all the simulations, analyzed the results, and wrote the manuscript. HvS provided guidance and wrote and revised the manuscript, particularly the statistical test portion. DY helped with the numerical model momentum balance output and provided guidance in drafting and revising the manuscript.

*Competing interests*. The contact author has declared that none of the authors has any competing interests.

*Acknowledgments.* We are grateful to the German Climate Computer Center (DKRZ) for providing computer resources. Yang Ding was supported by the National Natural Science Foundation (NSFC, No. 42130403 and 42076010).

*Data availability*. The datasets generated during and/or analyzed during the current study are available from the corresponding author upon reasonable request.

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

**Appendix**

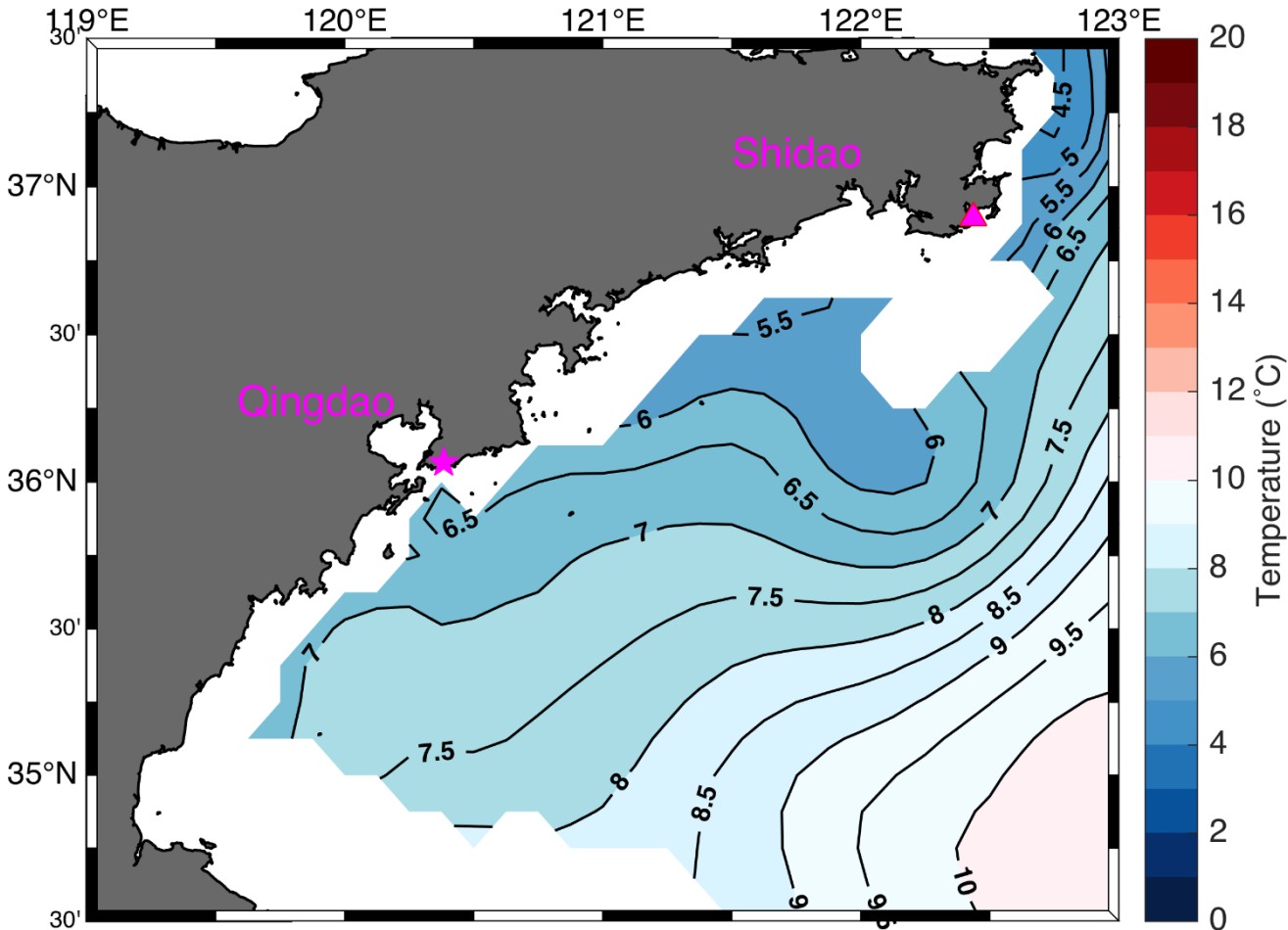

**Figure A1. Monthly mean temperature evolution of the Qingdao cold water mass in March at a depth of 25 m, as derived from the ensemble mean of the control runs.**


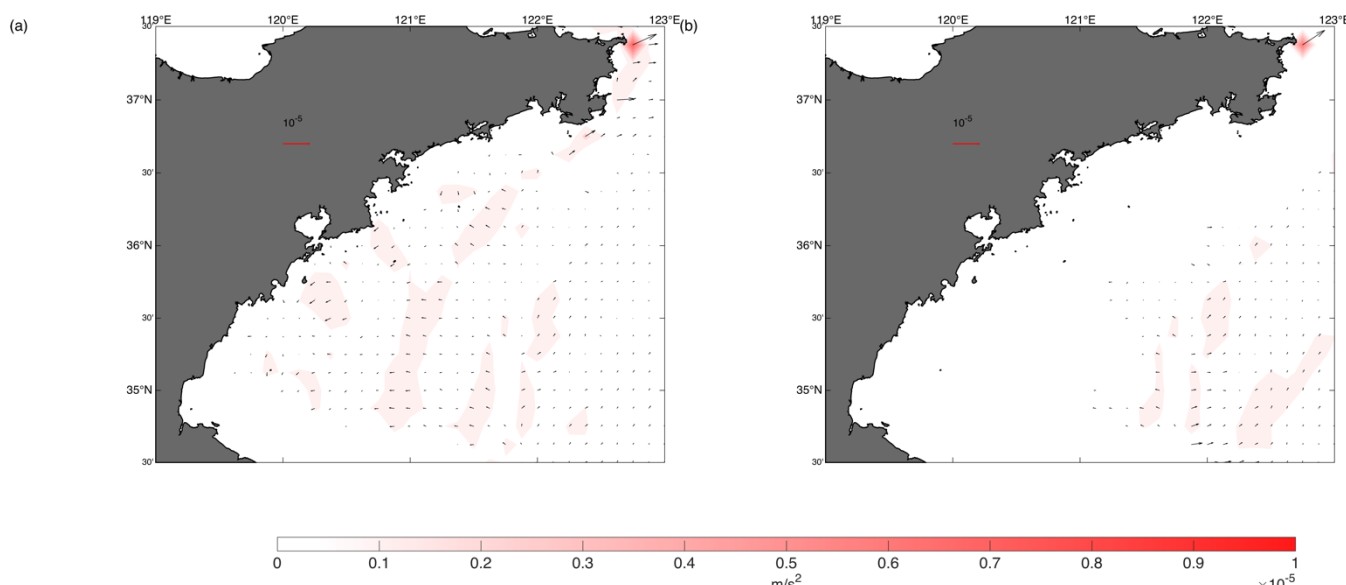

**Figure A2.** Horizontal distributions of the sum of the barotropic gradient force $-g\frac{\partial \zeta}{\partial x}$, baroclinic pressure gradient force $-\frac{1}{\rho}\frac{\partial}{\partial x}\int_{z}^{\zeta}\rho g dz$, Coriolis force $fv$ and vertical friction force $F_u$ (monthly mean for May) at depths of (a) 25 and (b) 40 m.

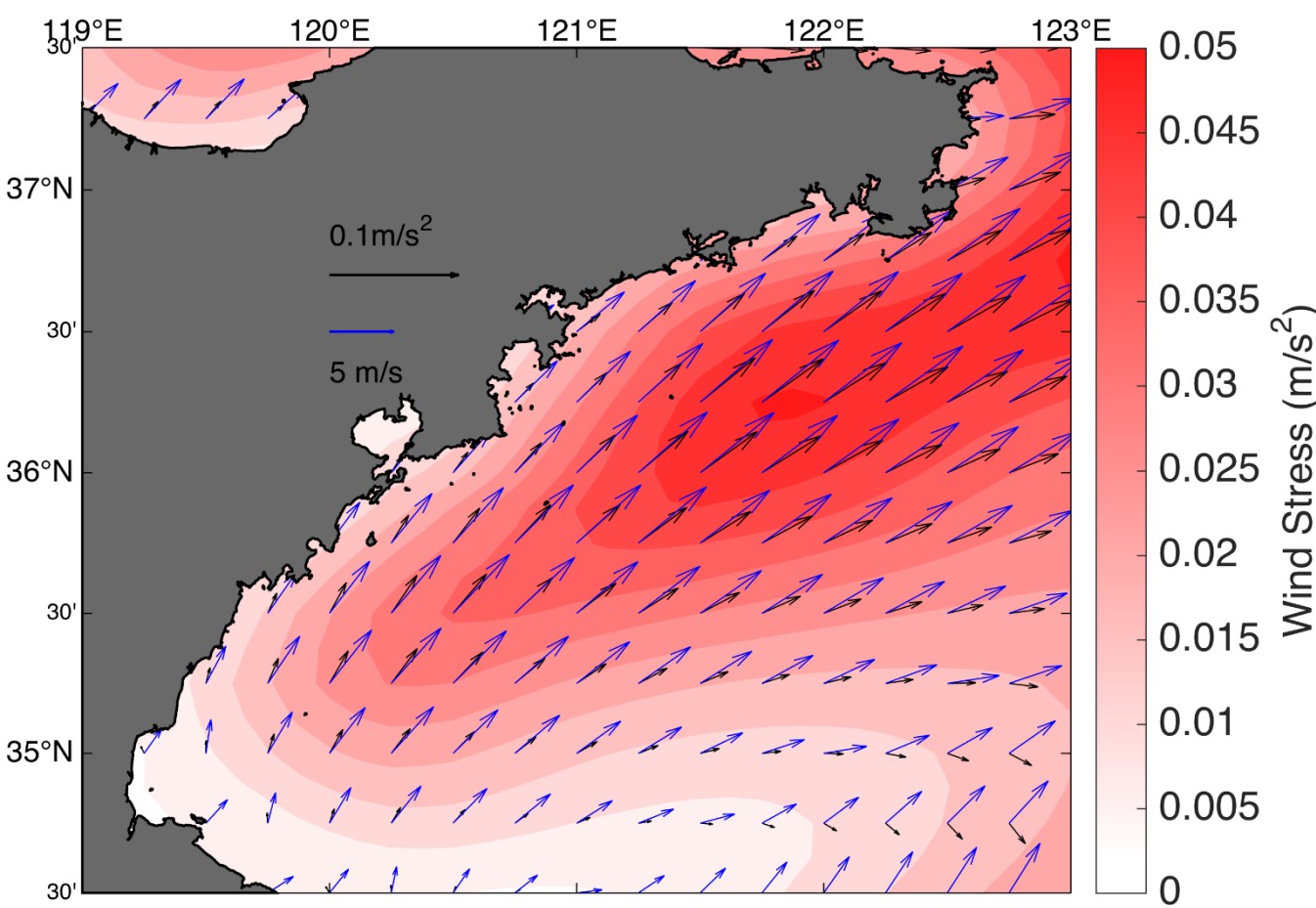


**Figure A3. Mean surface wind stress in May 2019 used in the control run and the no-tide ensembles. (The magnitudes of the wind stress and wind are different. To show both in the same diagram, two scales were used. The blue and black arrows represent the magnitudes of the wind and wind stress, respectively.)**
