# Peer review of "Evolutions of the seasonal anticyclonic circulation around the Qingdao cold water mass in the China marginal sea and its mechanism"

_EGUsphere, 2024_

## Author Comment (AC1)

**Thank you very much for your effort and time to read our paper and help us improve it. The comments are very insightful and constructive for us. We have revised the manuscript point by point to the comments as listed below.**

**Review1 General comments:**

*This work designs a series of numerical experiments using FVCOM model to investigate the drivers of Qingdao cold water mass. The study finds that both tidal forcing and winds are important in the formation of cold water. However, the manuscript is not well organized and include the information not directly related to this work. The objectives of this study are not very clear. Figure 12 is not even mentioned in the manuscript. In Conclusions, the authors confused "anti-clockwise" with "anti-cyclonic". Therefore, I would recommend the publication of the manuscript after major revision. Below are my detailed comments and suggestions.*

**Response to the comment:**

Many thanks for your helpful review, perspectives, and comments. We have removed the content that is not strongly related to the work. To express the objective of this study more clearly, we have also rewritten the scientific questions based on your suggestions. We apologize for the confusion regarding Figure 12; we have corrected it and added a description. Additionally, we have made the necessary modifications to ensure consistency regarding the "anti-clockwise" and "anti-cyclonic" terms. Please see the detailed response below.

**Minor Comments:**

*1.      Lines 35 and 47: line 35 states that the cold water is featured by moderate salinity, while line 47 says it is a low salinity water mass. Please be consistent.*

**Response to the comment:**

Thanks for the correction. We have corrected and changed the low salinity to moderate salinity.

*2.        Lines 39-40 and Lines 47-49: information duplication.*

**Response to the comment:**

Thanks. We have deleted that "It forms from March to April, reaches its peak in May" in lines 47-49 (in old version).

*3.        Lines 59-70: I would recommend the authors to rewrite this portion. First, other cold water masses are not directly related to this work, so it is unnecessary to include so many details. Please be concise and remove those examples. Second, the question need to be answered in a study should be very specific. Why you still use an example to express the question?*

**Response to the comment:**

We have removed the examples and rewritten this part to make it concise. Please see lines 62-66:

*Bohai Sea (Liu et al., 2003; Wan et al., 2004; Zhou et al., 2017) and the Yellow Sea cold water mass (Ho et al., 1959; Hur et al., 2000; Wei et al., 2010; Yuan et al., 2013). The special circulation structures around the Bohai cold water mass and Yellow Sea cold water mass have been well described in previous research (Wang et al., 2014; Xia et al., 2006; Zhou et al., 2017; Zhu and Wu, 2018), but the anticyclone current field analysis near the Qingdao cold water mass still needs to be investigated.*

*4. Lines 71-77: This portion is poorly organized. In question one, why you say it is a special cold pool structure? Question two is too vague. Influence what features of the anti-cyclonic gyre? Morphology, Duration, Magnitude, Timing, or something else? There is no need to list only two questions using one paragraph.*

**Response to the comment:**

Regarding scientific question 1, we have deleted the "it is a special cold pool structure" and rewritten the scientific question to "Is the Qingdao cold water mass causing the local seasonal anticyclonic structure?" (lines 66-67). For scientific question 2, we have added "the morphology, magnitude, and position of" to make it more specific. Additionally, we have combined this paragraph with the previous paragraph.

*Lines 66-68: In this work, we investigate the following questions: (1) Is the Qingdao cold water mass causing the local seasonal anticyclonic structure? Does such a seasonal*

*anticyclonic circulation fit the geostrophic balance? (2) What factors influence the morphology, magnitude, and position of seasonal anticyclonic circulation horizontally and vertically?*

*5. Line 82: cite the wrong paper. It should be Chen's 2003 or 2006 paper. Also there are a lot of citation format issues in this work. Please double check your citations.*

*Chen, C., Liu, H. and Beardsley, R.C., 2003. An unstructured grid, finite-volume, three-dimensional, primitive equations ocean model: application to coastal ocean and estuaries. Journal of atmospheric and oceanic technology, 20(1), pp.159-186.*

*Chen, C., Beardsley, R. and Cowles, G., 2006. An unstructured grid, finite-volume coastal ocean model (FVCOM) system. Oceanography, 19(1), pp.78-89.*

**Response to the comment:**

Sorry, we have corrected it.

*6. Line 82: remove "while".*

Corrected.

*7. Line 89: provide citation of CFSv2 dataset.*

Corrected.

*8. Lines 93-94: why only one sentence in this paragraph? Is it necessary to use one paragraph to state tidal forcing?*

Corrected.

*9. Line 101: Very confusing. Please rewrite this sentence.*

**Response to the comment:**

Sorry about this confusion. We have rewritten this sentence, please see lines 103-104:

*Ensemble simulations are conducted for the control run, no-tide run, and no-wind run. Each ensemble simulation consists of four numerical simulation members.*

*10.     Lines 103-104: why you choose these four time frames? Also there is no information regarding your 9-year climatological simulation.*

**Response to the comment:**

This is a good question, which reminds us to address the motivation of ensemble simulation. We have added a paragraph to describe why we need the ensemble simulation and why we conduct ensemble simulation in such a way. In short, we would like to seed a perturbation in the ensemble simulation, so we used four slightly different initial conditions, namely shift in model start time. Additionally, the details of how the perturbations are seeded do not matter, they can be seeded by slightly different model start times or by the simulations with the same model configurations conducted in the different clusters (Geyer et al. 2021). We have addressed the motivation for ensemble simulation in lines 113-121 in the manuscript.

*The motivation for using ensemble simulations is based on the observation (Lin et al., 2022, 2023; Penduff et al., 2019) that deviations form within the ensemble members if the ensemble simulations are conducted with the same model configuration except for slight perturbations in the initial conditions. In other words, if we have only one numerical simulation, the model output will be a mix of "signal" (external forcing) and random effects. Some spatial features are not repeatable in other ensemble members, even though the model configurations are the same. Averaging across ensemble simulations efficiently reduces the random impacts of randomness. Therefore, in Sections 3 and 4, we consider the ensemble means for further analysis. An ensemble simulation with slightly different initial conditions is one way to analyze ocean internal variability. For the ensemble simulation configuration in this study, we follow the tradition of generating an ensemble simulation with slightly different initial conditions (Penduff et al., 2019).*

We used the 9-year climatological simulation to provide slightly different but consistent initial conditions. All the initial conditions for the ensemble simulation members are taken from the same climatological simulation, even though the time shifts a bit. The model starting time of the 9-year climatological simulation is 1st Nov. 2008, and the model ending time is 31st Dec. 2019. The climatological forcing for the climatological run is a smooth annual cycle without weather variations based on National Centers for Environmental Prediction (NCEP) Climate Forecast System Version 2 (CFSv2) data. We have added the description of the climatological simulation in lines 106-110:

*Note that we conducted an independent 9-year simulation to generate slightly different but generally consistent initial conditions for the ensemble simulation. The model starting time of the 9-year climatological simulation was 1st Nov. 2008, and the model ending time was 31st Dec. 2019. The climatological forcing for the climatological run is a smooth annual cycle*

*without weather variations based on National Centers for Environmental Prediction (NCEP) Climate Forecast System Version 2 (CFSv2) data.*

*11.        Line 105: why you choose year 2019? If the cold water mass has strong interannual variability, can 2019 scenario represent the normal year condition?*

**Response to the comment:**

Thanks for your question. 2019 was chosen as an example for analysis. If we choose to simulate the results of other years (for example, we have simulated the result of 2010 discussing the characteristics of the seasonal variation of Qingdao cold water mass, published in a Chinese journal (Huang et al., 2019)), the basic characteristics will be similar, and the simulation of this anticyclonic circulation in other years is currently running, which will be the next step of work and the focus of the next step, namely, discuss the interannual variation characteristics and mechanisms of the interannual variation this anticyclonic circulation. We have mentioned this limitation and next step work in the paper.

*Lines 375-379: In this paper, we use the model result of 2019 as an example. Huang et al. reported similar basic characteristics of the temperature and salinity of the Qingdao cold water mass in 2010 (2019). Currently, we limit ourselves to the simulation of 2019; in the future, we plan to discuss the interannual variation characteristics and mechanisms of the interannual variation in this anticyclonic circulation.*

*12.        Lines 115-116: why you say "these results are not different if we consider the mean T …"? In your example the benchmark's mean T is around 5 while the sensitivity test's mean T is ~7.5. The rationale behind this setup is very tricky.*

**Response to the comment:**

Sorry about this confusion, we have deleted this sentence and rewritten why we do a t-test for the control run, no-wind, and no-tide simulation. Hope it can express more clearly in the new version. Please see lines 122-134.

*Because the deviations exist between ensemble members caused by the randomness, we need to test whether the differences between the ensemble mean of the control run and the no-tide run (/no-wind run) may be caused by external forcings (tidal or wind forcings) or could only because of randomness. A proper way to do so is statistical hypothesis testing with the null hypothesis: "external forcing has no effect". If this null hypothesis is rejected with a sufficiently small risk, then a valid conclusion is that an external factor has an effect and plays*

*an active role. Here, a t-test is done for the ensemble monthly mean for May. The results (Figs. 2 and 3) show the sensitivity of the forming of the spring cold water mass to the presence of tidal forcing and wind forcing. Those grid points, at which a t-test indicated that the effect of external forcing is significant, are marked with a cross. Figs. 2 and 3 demonstrate that the difference between the control run and the no-tide ensemble (or the /no-wind ensemble) is significant, especially where the intra-ensemble deviations are larger.*

[Figure]

**Figure 2. The difference in temperature between the ensemble means of the control runs and those of the runs without tidal forcing. The crosses represent the areas where the difference between the control run and the run without tidal forcing was significant at the 5% level.**

[Figure]

**Figure 3. The difference in temperature between the ensemble means of the control runs and those of the runs without wind forcings. The crosses represent the areas where the difference between the control run and the run without tidal forcing was significant at the 5% level.**

*Additionally, the explicit logic behind the significance of the statistical test under review (the preprint: https://www.preprints.org/manuscript/202407.2261/v1; section5.2). As the main topic of this paper is the seasonal anticyclonic circulation around the Qingdao cold water mass, so we only briefly describe the ocean's internal variability related to the definition and understanding in the paper.*

*13.      Line 122: Results.*

 Corrected.

*14.      Lines 124-126: the structure of this part is weird. There is no need to have this portion as a paragraph, especially when it does not provide too much detailed information.*

**Response to the comment:**

We have deleted this portion, and have rewritten the model validations can be found in the previous publication in one sentence in section 2.1.

*15.      Lines 129-130: you already included the similar information in Introduction, no need to repeat it here.*

**Response to the comment:**

We have deleted this part.

*16.    Lines 131-133: the figure should be included in the supplementary material to help readers better understand your work.*

**Response to the comment:**

We have added the figure as suggested.

*17.    Line 137: remove "moreover". "the shape of ... has a ... direction"? Rephrase this sentence.*

**Response to the comment:**

We have deleted this sentence.

*18.    Line 138: it should be Shandong coastal current, not Bohai coastal current or Bohai coast current.*

Corrected.

*19.    Line 139: In Fig. 2, why not show data in June?*

**Response to the comment:**

We have added a panel showing the salinity data in June. Please see the Fig. 4 in the revised manuscript.

*20.    Line 141: "Diao et al" citation issue.*

Corrected.

*21.    Line 144: the figures directly jump from Fig. 2 to Fig. 15?*

**Response to the comment:**

Sorry, we have moved Fig. 15 in the old version to Fig. 5 in the revised version and adjusted the figure number correspondingly.

*22.    Line 147: Additionally?*

We have deleted the "Additionally".

*23.     The color gradient of colorbar and the x axis labels are not well matched.*

Thanks. We have replotted the figures. Please see Figs. 9 and 10. And we have avoided this kind of problem in the revised paper.

*24.    Line 161: to July?*

Corrected.

*25.    Line 163: it is hard to tell the anticyclonic structure is closed from the figure.*

**Response to the comment:**

Sorry for the confusion, it should be the anticyclonic structure closes in May, rather than "in May and June". To be more accurate, we have rewritten the sentence, and the sentence in the revised version is "In May, the anticyclonic structure almost closes".

*26.    Lines 199: In Figs. 5d and 6d, I don't think the statement in the manuscript "vertical friction term plays a role, especially east of 122E" match the figure.*

**Response to the comment:**

Ok. The significance of the vertical friction terms is mostly described in section 3.4, so we decided to delete the "vertical friction term plays a role, especially east of 122E" in section 3.3 and address the importance of the vertical friction later in section 3.4.

*27.    Line 210: the subtopic is too long. Please be concise.*

**Response to the comment:**

We have changed the subtopic to "Relation between the seasonal anticyclone circulation pattern and the Qingdao cold water mass".

*28.    Line 216: why you choose 35.5 N transect? If you look at Fig. 1 this transect does not cross the center of the cold water mass.*

**Response to the comment:**

35.5 ºN was chosen because it is around the center of the anticyclonic structure. The center of the Qingdao cold water mass is not exactly the same as the center of the anticyclonic structure. Additionally, we have examined the results if we shift the transect 0.5º northward or southward. It does not change the conclusion that the geostrophic balance is not satisfied because the topography of the Qingdao cold water mass is shallow, thus inducing strong vertical friction.

*29.    Line 280: I don't think the authors discussed Fig. 12 in the manuscript.*

**Response to the comment:**

Sorry for this. The Fig. 12 in the old version manuscript is now Fig. 14 in the revised version. This figure is used to show that the center temperature of the Qingdao cold water mass decrease by 2°C when the wind forcing is turned off, so I have the description of it in the manuscript.

*Lines 314-317: On the other hand, in the absence of mixing caused by surface winds, the temperature of the Qingdao cold water mass decreased by 2°C (Fig. 14), resulting in an increase in the baroclinic pressure gradient force around the location of the Qingdao cold water mass in the no-wind experiment. The vertical friction somewhat decreases because of the lack of wind stress in the no-wind experiment.*

*30.        Lines 310-311: rephrase this sentence "for the eastern portion,….but for the eastern portion,…"*

**Response to the comment:**

Thanks for the correction. The correct sentence should be "For the western portion (122.625-123°E), the wind forcing still plays a role, but for the eastern portion, tidal forcing contributes more to upwelling". We have corrected it in the manuscript.

*31.        The two transects show very similar results. I think one transect is enough.*

**Response to the comment:**

We have modified it as suggested, please see Fig. 5 in the manuscript.

*32.        Line 353: be concise. I feel this section does not relate to the topic of this research. I would recommend to remove it.*

**Response to the comment:**

Deleted as suggested.

*33.        Line 369: Conclusions*

Corrected.

*34.        Lines 372, 373, 378: anti-clockwise or anti-cyclonic?*

**Response to the comment:**

Sorry about this. It should be anticyclonic. We have corrected and double-checked it in the whole manuscript.

---

## Author Comment (AC2)

**Reviewer 2 General comments:**

*This study presents the results of a local anticyclonic gyre around Qingdao cold water based on 3D numerical simulations. In short, the authors found that wind and tide are major factors influencing the anticyclonic structure, not the temperature and salinity gradients caused by the cold-water mass. The manuscript is generally well written but clarifications or more explanations and rearrangements of figures are suggested. There are also some typos. Please see below the detailed comments.*

**Response to the comment:**

We appreciate the reviewer for the time and effort in reviewing the manuscript. Thanks for the confirmation of the paper and suggestions which help us to improve the manuscript significantly.

*1. A large bathymetry map identifying locations (e.g., Shandong Peninsula, Korean Peninsula), coastal seas (e.g., Yellow Sea, South Yellow Sea, Bohai Sea), and coastal currents (e.g., Bohai coast current, Yellow Sea warm current) mentioned in the paper are suggested to provide a big picture for readers who are not familiar with this region.*

**Response to the comment:**

We have added a map which identifies the locations, please see Fig.1.

*2. Please supplement explanations on why ensemble experiments are needed. Why not directly using the realistic simulation but initializing the model for four times to get the ensemble?*

**Response to the comment:**

Thanks for your question, which reminds us to address the motivation for ensemble simulation. We have added a paragraph to describe why we need the ensemble simulation and why we conduct ensemble simulation in such a way. Essentially, we conduct the ensemble simulation to reduce the randomness caused by a single numerical simulation; and initializing the model four times a way to generate an ensemble simulation, such a way follows the tradition of previous research (Büchmann and Söderkvist, 2016; Geyer et al., 2021; Penduff et al., 2019). Additionally, the details of how the perturbations are seeded do not matter, they can be seeded by slightly different model start times or by the simulations with exactly the same model

configurations conducted in the different clusters (Geyer et al. 2021). We have addressed the motivation of ensemble simulation in lines 113-121 in the manuscript.

*The motivation for using ensemble simulations is based on the observation (Lin et al., 2022, 2023; Penduff et al., 2019) that deviations form within the ensemble members if the ensemble simulations are conducted with the same model configuration except for slight perturbations in the initial conditions. In other words, if we have only one numerical simulation, the model output will be a mix of "signal" (external forcing) and random effects. Some spatial features are not repeatable in other ensemble members, even though the model configurations are the same. Averaging across ensemble simulations efficiently reduces the random impacts of randomness. Therefore, in Sections 3 and 4, we consider the ensemble means for further analysis. An ensemble simulation with slightly different initial conditions is one way to analyze ocean internal variability. For the ensemble simulation configuration in this study, we follow the tradition of generating an ensemble simulation with slightly different initial conditions (Penduff et al., 2019).*

We used the 9-year climatological simulation to provide slightly different but consistent initial conditions. All the initial conditions for the ensemble simulation members are took from the same climatological simulation, even though the time shifts a bit.

*3. is the anticyclonic structure formed at depth shallower than 25 m?*

**Response to the comment:**

We have plotted the current pattern at the depth 5m and 15m, please see below (Figs. 1* and 2*). The shape of the anticyclonic structure at the depth of 15m can still be visible, but the center is not very clear compared to that at 25m. The anticyclonic structure disappears completely at 5m depth. Thus, we choose a 25m layer for detailed analysis because the boundary and center of the anticyclonic are much clearer.

[Figure]

**Figure 1*. The horizontal circulation distribution around the Qingdao cold water mass (5 m layer) in April (a), May (b), and June (c).**

[Figure]

**Figure 2\*. The horizontal circulation distribution around the Qingdao cold water mass (15 m layer) in April (a), May (b), and June (c).**

*4. "temperature or salinity gradient" was used several times, are these horizontal or vertical temperature/salinity gradient?*

**Response to the comment:**

Sorry about this. We have clarified the horizontal or vertical temperature/salinity gradient accordingly in the manuscript.

*5. The short Section 3.1 Model validation could be merged into Section 2.1.*

**Response to the comment:**

We have merged the brief model validation into Section 2.1 as suggested.

*6. '-fu' in Equation (2) should be '+fu'. The expression of the barotropic pressure gradient force should be g\*dζ/dx in Line 190.*

**Response to the comment:**

Corrected. Thank you for the correction.

*7. the identified different roles of wind and tide on upwelling that occurred east or west of the cold water mass is interesting. Yet, what leads to the different roles of wind and tide on different sides of the cold water mass seems lacking. The tide-induced front was mentioned. Could you add a plot showing where the tide-induced front is?*

**Response to the comment:**

We have added a summary of how the wind and tide affect the upwelling at the end of section 4.3. The model results show that the western portion of upwelling is contributed by wind forcing and the eastern portion of upwelling is contributed mostly by the tidal forcing, but wind forcing plays a role as well. The upwelling caused by the wind force can be explained by the Ekman theory because of the predominant southeasterly monsoon; and the upwelling

contributed by the tidal forcing is because of the tide-induced front (Fig. 3*), which affects the horizontal (Fig. 3*) and vertical (Figs. 5a and 5b) temperature distribution. The temperature variation will change the density distribution. The density redistribution will further influence the baroclinic pressure gradient force change accordingly around the front zone, which further triggers upwelling (when the tidal forcing is considered). The mechanism of how tidal forcing changes the baroclinic pressure gradient force and in the end triggers the upwelling is explained in previous research (Lü et al., 2010), in this paper, we found that such explanation can interpret the upwelling around the Qingdao cold water mass as well.

[Figure]

**Figure 3\* The temperature distribution in the control run (a) and no-tide experiment(b). In the control run(a), there is a front around the cold water mass. When the tidal forcing is turned off, such front disappears (b).**

*8. All "anti-clockwise" in Conclusion should be "anti-cyclonic" or "clockwise".*

**Response to the comment:**

Sorry, we have corrected it.

*9. Line 35: the cold pool is characterized by "moderate salinity" but later, it was described as "low salinity" (Lines 47, 53). Please check it.*

**Response to the comment:**

Corrected. To be consistent, we have changed to "moderate salinity".

*10.        Line 41: "emergence" to "mergence".*

Corrected. Thanks!

*11.        Line 56: add reference(s) after "…the Qingdao cold water mass forms".*

*Response to the comment:*

We have added the reference as suggested.

*12.        Line 57: "Yellow Sea cold water mass" should be "Qingdao cold water mass"?*

Corrected.

*13.        Line 70: "anticyclone" to "anticyclonic"*

Corrected. Thanks!

*14.        Line 115: "However, these results are not different…" seems to be not correct. In the example, the mean temperature of this grid node's ensemble is 4.875 ℃ for the control run and 7.375 ℃ for the no-tide run. Their difference is 2.5 ℃, which is different from the "3 ℃".*

**Response to the comment:**

Sorry for the confusing sentence. We have deleted it and rewritten the significance of the statistical test in lines 122-134. Additionally, we have an explicit version of why we need statistical test in another under-review paper with the topic of "Significance of Internal Variability for Numerical Experimentation and Analysis" (section 5.2; preprint: https://www.preprints.org/manuscript/202407.2261/v1). To explain more clearly, we would like to show part of the content in the reply to the review comments.

*Lines 122-134: Because deviations exist between ensemble members because of randomness, we need to test whether the differences between the ensemble means of the control run and the no-tide run (/no-wind run) may be caused by external forcings (tidal or wind forcings) or could be caused only by randomness. A proper way to do so is statistical hypothesis testing with the null hypothesis: "external forcing has no effect". If this null hypothesis is rejected with a sufficiently small risk, then a valid conclusion is that an external factor has an effect and plays an active role. Here, a t test is performed for the ensemble monthly mean for May. The results (Figs. 2 and 3) show the sensitivity of the formation of the spring cold water mass to the presence of tidal forcing and wind forcing. Those grid points, at which a t test indicates that the effect of external forcing is significant, are marked with a cross. Figs. 2 and 3 demonstrate that the difference between the control run and the no-tide ensemble (or the /no-wind ensemble) is significant, especially where the intraensemble deviations are large.*

*When such local tests are conducted, one has to expect that even if the null hypothesis is valid, at approximately 5% of grid points, the null hypothesis is rejected (multiplicity of tests, cf. von Storch, and Zwiers, 1999). Since the rejection rate is itself a random variable, the false*

*rejection rate can be much larger, but more than 20% is very unlikely. Here, the rate is considerably greater in both cases.*

*15.     Line 141: (Diao, 2015) to (2015)*

Corrected. Thanks!

*16.     Line 144: Figures should be mentioned in ordered sequence in the text.*

**Response to the comment:**

We have corrected it and avoided this problem in the revised manuscript.

*17.     Line 161: "July" should be "June" (?)*

Corrected.

*18.     Lines 160, 168: "southeast monsoon" and "southward monsoon" are a bit confusing. Is it southeasterly monsoon or northward monsoon?*

**Response to the comment:**

Sorry, we have changed to southeasterly monsoon. We would like to express that a southeasterly monsoon prevails around the Shandong Peninsula, and a northward current exists along the Shandong Peninsula.

*19.     Lines 178-179: please clarify "meaning that the Qingdao cold water mass is less affected by horizontal disturbances".*

**Response to the comment:**

We have deleted this unclear sentence.

*20.     Lines 199-200: "east of 122 ˚E" should be "west of 122 ˚E"? change "southwest wind stress" to "southwesterly wind stress," "northern current" to "northward current".*

Corrected. Thanks!

*21.     Line 299: please check "a decrease in the baroclinic pressure gradient force around the location of the Qingdao cold water mass in the no-wind experiment". Is it "increase"?*

**Response to the comment:**

Thanks for your correction. We have corrected it.

*22.      Line 335: "west side" should be "east side"?*

Corrected.

*23.      Add references in Section 4.4 for the Yellow Sea basin-scale cyclonic circulation.*

**Response to the comment:**

As we consider that the description of the Yellow Sea basin-scale cyclonic circulation is not strongly related to the main topic of this paper, we decide to delete section 4.4 in the revised version.

*24.      Question about the terminology: the anticyclonic circulation studied here disappears as the seasonal Qingdao cold water mass disappear. It's not a permanent feature, maybe call it a 'seasonal anti-cyclonic gyre' is more accurate or just use 'anti-cyclonic circulation'?*

**Response to the comment:**

Thanks for bringing this discussion to us. In our opinion, the gyre usually refers to a large-scale feature, such as the sub-tropical gyre. In our paper, maybe it is more appropriate to use seasonal anti-cyclonic circulation. Further discussion about this issue is warmly welcomed. Additionally, in the revised version, we have added seasonally in front of the anti-cyclonic circulation.

**Specific comments about figures:**

*1.      Some figures can be combined for better visualization and comparison of results, for example, the following figures can be put together: Figures 1 and 2, Figures 5 and 6 (those two can be re-formatted into two rows and each row shows four plots), Figures 10 and 11.*

**Response to the comment:**

We have replotted the above diagrams as suggested.

*2. the unit of the momentum term in several figures is wrong: Figures 5, 6, 10, 11. It should be m/s2.*

**Response to the comment:**

Thanks for the correction. We have corrected it.

*3. For those figures showing features at 25 m, some of them masked the nearshore region (I believe these are shallower than 25 m), like Figures 1-3, 5-6, but others have valid values in this shallow region, like Figure 10-11. Please check it.*

**Response to the comment:**

Many thanks for your comments. We have checked this problem and found the visualization problem is caused by unsuitable color bar. We have replotted the above diagrams.

*4. Figure 7: correct (e, f, g) to (d, e, f). Which direction does the positive value represent?*

**Response to the comment:**

Corrected. Thank you!

The positive value represents northward direction. We have added this informatio in the manuscript in line 233.

*5. Figure 8: there should be no unit for Ekman ratio.*

**Response to the comment:**

Corrected.

*6. Figure 11: why is "time average over two M2 tide cycle" used here, rather than using "monthly mean for may" as Figure 10?*

**Response to the comment:**

Thank you for pointing out this problem. We have modified it and both diagrams are monthly mean.

*7. Figure 12 is not cited in the main text. Figure 12a is the same plot as Figure 1b, keep one figure is fine.*

**Response to the comment:**

Sorry about this mistake. We have added a description for Fig. 12 and removed Fig. 12a (now Fig. 14 in the revised version) as suggested.

*8. Figure 14 is repeated with one of the plots in Figure 15. You may consider put Figure 15 into supplementary. It's also recommended to reduce the vector numbers in Figures 14-15 to make them easier to see.*

**Response to the comment:**

We have removed Fig. 14 in the revised version.

*9. Figure A1: which period of the surface wind stress is shown in this figure? The caption indicates that this \*constant\* wind field was used in the control run (?)*

**Response to the comment:**

Sorry about this. It is the monthly mean of surface wind in May. We use the realistic wind forcing, not the constant, as we described in the model configuration part. Sorry for not expressing this clearly. Fig. A1 (now Fig. A2 in the revised version) is only used to provide a direct feeling for the readers of the wind stress direction.

---

## Referee Report (RR1)

**General comments:**

This study investigated the formation and main drivers of the Qingdao cold water mass by using numerical simulations from a series of ensemble experiments. The study pointed out that the geostrophic balance is no longer applicable in the Qingdao cold water mass region due to the considerable friction terms. A seasonal anticyclonic circulation system was detected around the cold water mass. Wind and tides were found to be the main factors to this circulation. As for the method, the authors introduced an ensemble methodology, which is seldom applied and discussed in many numerical studies. I think, it is an advantage that can help this study to stand out. However, some major points still remain unclear and need to be resolved before the publication. Therefore, I recommend the publication of the manuscript after major revision.

**Major comments:**

1. The logic of the manuscript is unclear. I spent a lot of time trying to figure out how the authors organize the manuscript. My understanding is (1) that there is a seasonal anticyclonic circulation system observed around the Qingdao cold water mass; (2) that the geostrophic balance is no-longer applicable in coastal Qingdao and cannot be used for explanation of the formation of this anticyclonic circulation system; (3) that this anticyclonic circulation system mostly results from the balance of pressure gradient force, the Coriolis force, and the friction force with an emphasize of the non-negligible friction in the shallow water and (4) that the wind and tidal forcings contribute significantly to the evolution of this anticyclonic circulation system through the adjustments to the friction term. The authors, for example, put a part of discussion of wind and tidal impacts at the beginning (section 2.2), which, I think, is a scattered way of thinking and makes the readers hard to capture the main points of this study. Similar problems were found in the abstract and conclusion sections (see detailed comments below). Another example of this erratic or scattered way of thinking is reflected in the orders of figures shown. Figure 2,3, and 5b, 5c should go to the discussion section, while figure 5 should be merged with figure 15.

2. The manuscript is lack of in-depth discussion although a "Discussion" section is performed. Each subsection in section 4 seems like other result section without detailed quantification and comparison (against previous studies). Please see the detailed comments below.

3. There are some findings or points listed in the manuscript that are contradicted to each other weakening statements. I have listed them below.

4. The manuscript is lack of quantitative analysis and comparisons no matter in the results but also the discussion sections. This can largely weaken the convincibility of the findings.

**Detailed comments:**

1. The title is not specific enough. Saying "the seasonal anticyclonic circulation" may be too general. Per my understanding, the authors are investigating the mechanisms of the evolutions of this circulation. So, it could be better to point out the main purpose of the study in the title.

2. Lines 14-30. The abstract is not concise enough and is lack of a "main clue" to guide the reader to rapidly capture the findings (please see the above comments).

3. Lines 19-20. The authors should point out directly what results in the anticyclonic circulation rather than saying the cool poor is not the main cause.

4. Lines 22-24. Too verbose. Try to concise it.

5. Line 105. 11 year? The simulation period of the climatological run is from Nov 2008 to Dec 2019.

6. Lines 127-134. This part should belong to the discussion section and same for figures 2–3.

7. Line 159. "Northeast" or "southeast"? Please double check. If "northeast", the corresponding temperature and salinity distribution should be shown, i.e., enlarge the coverage in Fig. 4.

8. Lines 163 and 167. The location of the cold water mass center was mentioned twice but with different longitudes.

9. Lines 163–165. My understanding of the formation of cold water body is somehow slightly different from the explanation here. Firstly, during winter, water column is homogenized forming cold water from surface to bottom. Secondly, as in early summer, fast heating on the ocean surface induces strong and rapid stratification sealing the cold water below the thermoclines. Thirdly, vertical mixing due to wind disturb or upwellings is not strong enough to homogenized the water column, which results in the maintenance of near-bottom cold water mass.

So, it may not be appropriate to emphasize only the role of thermocline to the cold water mass formation. In addition, citations are needed here as there is no clue shown in Fig. 4 that thermocline leads to the formation of the cold water mass.

10. Figure 5 should be merged with Figure 15, as they contain largely overlapping information. Arrows in figure 5 are not clear enough.

11. Lines 183–185. Is the southeasterly wind strong enough to induce vertically-homogenized the northward currents?

12. Line 188. Could you also show the June pattern to support this statement?

13. Lines 190–192. This is an incomplete sentence. "As the water depth is shallow in the western portion of the anticyclonic circulation and the dominant southeasterly monsoon…" This part is incomplete. This is a conclusion-like statement. Model evidence are needed to support this causality.

14. Lines 200–201. I cannot see velocity increases with depth. Perhaps the authors need to replot the Figure 7 showing 5 separate 2D panels rather than using a 3D plot. The Figure 7 shown makes current arrows hard to compare. Also, the panels do not align with the corresponding depth, e.g., the first panel does not align with depth 25 m and so on.

15. Equation (1)–(2). Please also show the momentum equation in the vertical direction. Also please decompose the force terms in (1) and (2) into the term shown in Figure 8 and put all force terms to the right-hand side. It is important to provide the readers with the detailed mathematic expressions of the force terms shown in Figure 8 and others.

16. Lines 215–216. Although horizontal friction is usually negligible in open ocean, in the coastal region this friction may not be negligible. The Qingdao cold water mass occurs quite nearshore. So, please provide quantitative comparisons of all friction terms before removing any of them out of the analysis and discussion.

17. Lines 216–218. Lack of evidence. At least, the difference of barotropic gradient force and the sum of baroclinic gradient force, Coriolis force, and vertical friction force are needed before addressing the balance of these terms.

18. Line 219. The "southwesterly wind stress" contradict against the "southeasterly monsoon" in Line 183. In the northern hemisphere, the southeasterly wind most

likely induces southerly wind stress rather than southwesterly ones. Please overlap the wind patterns in Fig. A2.

19. Lines 220–222. It is more like a hypothesis or an inference without necessary evidence. Also I doubt that the horizontal friction also considerably affects the Qingdao cold water mass which locates quite nearshore.

20. Figure 8. Please also show the mathematic expression of each term in each subplot because the signs of these shown force terms are also important to readers.

21. Line 226. Format is wrong.

22. Figure 9. Same suggestion as for Figure 8.

23. Lines 235. The difference of pressure gradient force and the sum of Coriolis force and friction force is needed to support this statement.

24. Lines 239–243. Lack of horizontal friction. See above comment.

25. Lines 244–253. I think this part is a very important finding of this study and also the reason why the author wanted to dig deeper in the wind and tidal impacts in the discussion section. I suggest the authors provide more explanations and details here.

26. Line 244–246. It is not surprised that the geostrophic balance is not satisfied as it is a state of large-scale ocean flows in open ocean. In the coastal region, the authors may want to focus more on the theories of boundary layer. So then the friction is very important and may need more analysis and discussion.

27. Lines 270–271. But along the anticyclonic circulation, friction is lower in the no-tide experiment.

28. Liens 274–275. "…and Coriolis terms change direction…"? The force direction seems not change. For example, the changes in Coriolis force in the no-tide experiment is at an order of magnitude of 10E-6, while the Coriolis force in the control experiment is at an order of magnitude of 10E-5.

29. Figure 11b. Why control-no_tide? It different from other difference terms. Please make the calculation consistent.

30. Lines 310–312. I did not understand this sentence.

31. Lines 312–314. The causality is misleading. It is not the decreases in water temperature that cause an increase in baroclinic pressure gradient force but the weaker mixing processes when wind is eliminated.

32. Lines 321–323. I did not observe a downwelling system from figure 5a.

33. Line 325. "For the western portion (122.625–123ºE)" or "For the eastern portion (122.625–123ºE)"?

34. Line 326. The tidal mixing font is a new term here (not mentioned in previous sections). So, where is this tidal mixing font? Seems like it is a part of the decomposition of mixing. I am not sure. Please convince me.

35. Lines 330–331. Below 20m, all flows seem eastward.

36. Lines 332–341. The part is not convincible to me. I would suggest the authors compare the difference of current velocity between control and no-tide runs and the difference between control and no-wind runs.

37. Lines 349–351. There is still temperature font found in Figure 5b, which means that when tides are excluded, the cold water mass maintains at the similar location. So, how can the tidal impacts be important to the cold water mass? As I observed, the impacts of winds outperform tidal impacts, as in Figure 5c, the location of the cold water mass shift westward to 121.5ºE.

---

## Referee Report (RR2)

**General comments:**

I can see the improvement of the manuscript. However, I am still confused on some critical points. Therefore, I recommend the publication of the manuscript after major revision.

**Major comments:**

1. The logic of the manuscript is still unclear. In section 3.1 and 3.2, the authors showed the temperature and salinity features and circulation pattern of the cold water, trying to demonstrate the evolution of the cold water and the related mechanisms. However, there is a lack of supported evidence. I will list those points out later. In section 3.3, the authors tried to prove that geostrophic balance is not applicable for the maintains of the anticyclonic circulation in lower layer in May due to the non-negligible friction in the shallow water. Isn't it obvious in a shallow shelf? I am not quite sure why the authors spent a section to demonstrate this question. It may be better to merge this section into section 3.1 and 3.2? In section 3.4, the authors demonstrated that the anticyclonic circulation is NOT caused by the cold water mass which induces changes in baroclinic conditions. So, why not just to demonstrate which factor cause the anticyclonic circulation? As I observed from the figures shown in the manuscript, there are at least two compensate currents (secondary currents) which, I think, are important for the anticyclonic circulation development and the cold water mass evolution, i.e., the landward current at near bottom layers (westward current) and the upwelling system. Also, the background current system outside the shown region is very important. In section 3.5 and 3.6, the tidal and wind effects on the anticyclonic circulation are studied. But I am not quite satisfied with the explanation provided. I will list my concerns later. In section 4.1, the upwelling effects on the cold water mass is discussed, which I think is very important and should be move to section 3.3. More focus should be put to this part. And I found it is hard to follow the tidal effects on the upwelling and the mixed layers (lines 334–360). I will list my questions. Section 4.2 is too short to be a section alone. Instead, as supporting evidence for the tidal and wind effects, it may be better to merge this section into the sections discussing the tidal and wind effects (section 3.5 and 3.6). The same problem raises to section 4.3.

2. I am a bit confused as to why the authors devoted so much effort to comparing all the forcing terms. Based on Figure 5, the barotropic term appears to dominate in the near-bottom layer, which I believe is likely due to water piling up on the

eastern side of the domain under the influence of the southwesterly monsoon. As a result, the landward flow in the near-bottom layers along the coastal region is primarily driven by this pressure gradient force, with minimal influence from other forcing terms. This aligns with what I mentioned earlier regarding the compensating current in my previous comment. So, it seems that too much discussion on other terms (baroclinic, Coriolis, friction) is not necessary. Could you please clarify the rationale and necessity for such extensive discussion of these terms (as presented in Figures 5, 6, 7, 9, and 13)?

3. There are still some findings or points listed in the manuscript that are contradicted to each other.

**Detailed comments:**

1. Figure 1. Please overlap the bathymetry used in the model and also provide the coverage of the entire computation domain.

2. Section 2.2. There is a lack of description of the lateral boundary conditions and riverine forcings, which I think are very important to shelf dynamic simulations.

3. Line 103. Please use the specific years here, e.g., use November 1st, 2017 instead of November 1st of the 9th year. The latter expression may confuse the readers if they count the 2008 as the 1st year.

4. Lines 141–142: Evidence is needed to support the claim that the merging of warm and cold water (i.e., the deformation of the cold water mass) is primarily due to increased solar radiation and not other factors, such as mixing caused by the sustained monsoon. Additionally, doesn't surface heating contribute to maintaining the cold water mass near the bottom by reinforcing strong stratification?

5. Lines 142–145. Although the author added my understandings on the cold water mass formation, it is just the knowledge from my previous studies in physical oceanography. The authors need to provide evidence to support this statement. I see that Figure 12 is a helpful clue. And also, the author may need to add more plots in the appendix showing the evolution of this cold water mass from April to June (may be evolution of the transect as depicted in Figure 12?). So, it is better to move Figure 12 to here.

6. Lines 141–142 contradict to Lines 142–145. The former one emphasize the importance of solar radiation, while the latter emphasize the role of mixing and upwelling.

7. Section 3.2. Please confirm the direction of the summer monsoon. According to the Figure A3, it is southwesterly wind, but southeasterly wind is used during the discussion, e.g., Line 166. The inconsistence is found almost all over the manuscript. Please double check.

8. Lines 159–161. The southwesterly wind induces the northward current on surface, but the authors indicate that the near bottom northward current is also induced by this wind pattern. I doubt that. This is the reason why I asked, "Is the southeasterly wind strong enough to induce vertically-homogenized the northward currents?" during the last round of revision. It is better to show the current pattern from surface to near bottom. Now, I would argue that it is the compensate current that induces the northward current near bottom but not the direct effects from the surface winds.

9. Lines 166–167. This sentence is an incomplete one grammatically. The "southeasterly" contradicts to the previous description and Figure A3. The causality shown confused me. Why do the shallow water and southeasterly monsoon lead to stronger current at the west part of the anticyclonic circulation? Here, the authors may miss the boundary effects but only focusing on the local wind effects. So, how does the current system like over the entire computational domain?

10. Lines 171–174. What is the purpose to show the circulation below 25m?

11. Line 174–175. I don't think the baroclinic pressure gradient forces are contribution to the vertical structure of the current patterns near bottom. As I observed from the Figure 4, current patterns are quite similar over these near bottom layers (northwestward and northward). Isn't is due to the landward compensate currents or the current from the south boundary? A well-known Subei current system usually intrudes into the Yellow Sea in summer and merge with the cold water mass. That is, a larger picture of current systems needs to be discussed.

12. Lines 175–176. I don't understand why the anticyclonic circulation contribute to the formation of the cold water mass near bottom? Does it contradict to Lines 142–145?

13. Line 194–195. The vertical friction shown in figure 5 is the friction at depth of 25m. So, the wind stress at the surface layer does not contribute to this friction term but the current speed gradient around the 25 m depth does.

14. Lines 195. I doubt that the surface wind can impact the current pattern near bottom through direct dragging. If so, the whole water column can be well mixed with not cold water exists at near bottom.

15. Lines 190–199. I see that the discussion is conducted mixing the current patterns at surface and near bottom layers, which should be discussed separately as they should not be the same as shown by Figure 3 and Figure A3.

16. Line 200. The expression of the vertical friction force is wrong and same for Line 219.

17. Section 3.4. I don't think discussion on geostrophic balance is useful for linking the anticyclonic circulation and the cold water mass.

18. Line 227. The term "significant" has statistical meaning. So, "significant" usually come along with statistical tests. Please use another term. Please check throughout the manuscript for this issue.

19. Lines 227–232. The geostrophic balance should not be the theory discussed here as in this region, wind forcing cannot be ignored. Instead, it is the wind stress that causes the pressure gradient forcing (higher SSH on the east than on the west evident by the westward barotropic term) and the compensate current in lower layer but not the violation of geostrophic balance by tidal forcing. The explanation here conflicts with my knowledge. Holp the authors can convince me if I am wrong.

20. Lines 233–236. What does the "dynamic reason" refer to? And I don't see the anticyclonic circulation contribute to the evolution of the cold water mass, rather, based on my study on the figures show, both of which (anticyclonic circulation and cold water mass) are the results of the compensate currents induced by the surface wind forcings. Again, please convince me if I am wrong.

21. Section 3.5. In-depth discussion is lacking. I suspect that the elimination of tidal forcings result sin the changes of the background current systems which affect the anticyclonic circulation near Qingdao. The authors, however, only provide the description of how the anticyclonic circulation changes due to the absence of tides.

Additionally, when removing the tidal forcing (that is in the no-tide experiment), how the boundary conditions are configured? The tidal forcing here should be the tidal signal generated within the computational domain, right? If tidal signal is not removed from the open boundary conditions, the no-tide experiment still contains tidal signal generated from outside.

22. Line 265. Please be consistent of the difference terms. Based on the caption, Figure 8c is flow differences of no-tide and control runs (no-tide minus control), Figure 8d is flow differences of control and no-wind runs (control minus no-wind). And what does the colored patches represent? Magnitudes? Or directions?

23. Section 3.6. The authors demonstrated that the wind is the dominant driving force for the anticyclonic circulation, which I agree with. But I don't agree with the rest. Firstly, there is a westward shift of the Qingdao cold water mass when wind is removed. It should be pointed out and discussed but the authors did not. Instead, the authors mentioned that the temperature of the cold water decreases by 2°C which is due to the weaker wind mixing when wind is removed. Rather, I think, it should be related to the location changes in cold water mass.

24. Line 309. Based on Figure 12a, the upwelling does not reach the surface but stop at depth around 10m

25. Lines 312–313. As I observed, the upwelling occurs over the entire transect but with different strength.

26. Lines 313–315. Yes, it is true. But the convergence is not pronounced. Instead, the most remarkable point is the eastward surface current and the associated secondary westward current at lower layers over the east of the 122E and the strong upwelling system.

27. Line 317–318. Based on Figure 12a–12b, the upwelling system is strengthened over studied transect but not just the east side when tides are considered, right?

28. Lines 320–321. The westward or eastward flows in the rectangle zone are not the most remarkable point but the upwelling, instead, westward current at lower layers over the east of the 122E may be more interesting.

29. Line 325. Southeasterly monsoon? The upwelling is contributed by tidal forcings, not by wind forcing?

30. Lines 309–326. These paragraphs should be more concise as they only highlight two points: the upwelling system and how it is related to wind and tide. Apparently, the authors did not provide convincible evidence of how tide affect the upwelling here.

31. Lines 334–360. I lost here. Instead of believing the tide induce thermal fronts, I would rather believe that the background current system (like the Subei coastal current) is affected by the tides. So, when tides are removed, the background current system change a lot leading to the changes in upwelling and also anticyclonic circulation around the Qingdao cold water mass. Also, I don't know why the authors focus on the relationship between tide and baroclinic pressure gradient force which is a very small term comparing to the barotropic pressure gradient force. Again, the tidal effects can be reflected by changing the background current pattern which, I guess, results in more pronounced changes in the barotropic term than the baroclinic term.

32. Figure 13. Please double check the caption.

33. Line 369. No. Oceanic modelers use statistic tests a lot.

34. Lines 370–380. Is t-test an appropriate test for this study? You only have 4 ensemble members. That is, for example, for the temperature at a given grid point, you are testing if the 4 temperature values in the control run are significantly different (assuming two-side testing) from the 4 temperature values in the no-wind (or no-tide) run. So, sample size is very small. Hope my understanding to your t-test is correct.

35. Lines 377–380. I don't get it. Could you provide more detailed explanation?

36. Lines 394–396. The anticyclonic circulation discussed in this study locate at near bottom, right? The northeastward current induced by summer southwesterly monsoon is at the surface. So, how does this surface current affect the near bottom current system?

37. Section 4.3. This section is weak as the authors did not do a great comparison between the finding from this study and the previous studies, e.g., which points agree with the previous findings, and which do not.

38. Figure A1. The term "evolution" is not expected as only one map is shown.

39. Figure A3. Blue arrow is for wind while the black for the wind stress, right?

---

## Referee Report (RR3)

**General comments:**

The manuscript has been improved; however, several critical issues remain unaddressed. My main concerns lie with the main current patterns in the Yellow Sea, and the effects of wind and tidal forcing on these patterns. The current version lacks a logical, thorough, and convincing analysis of these dynamics. I therefore recommend a major revision.

**Major comments:**

1. What made me confused during the revision for the last two version is the main current patterns in the Yellow Sea in summer. After some literature studies, now, I have some understandings. In Figure 1, the authors show only the YSCC, which is corresponding to the southward current in Figure 8a 122.5E–124E. This is the eastern boundary of the Qingdao cold water mass. However, the northeastward current in the Lu'nan coast (seeing figure below) and the northward current in Subei coastal current were not shown in Figure 1 and discussed thoroughly. These two currents are at the western boundary of the Qingdao cold water mass. The listed 3 currents together generate the summer anticyclonic circulation discussed in this work. Besides, the currents shown in the below figure and the Figure 1 in the manuscript are surface currents, however, the authors aimed to analysis the current patterns at near bottom (25 m). First, the authors should prove that the patterns of these 3 current systems are similar at surface and near bottom. This is what I have pointed out in the last revision. However, in the updated manuscript, the authors still didn't provide detailed analysis on it. Secondly, before the discussion on any effects on the current patterns, the authors should identify and label the current systems in their simulations (i.e., label the current system in Figure 8a). Additionally, the current system in Figure 8a should cover the entire computational domain.

[Figure]

图1 黄海流系示意图(据文献[6,9,14]重绘)

**Fig. 1 Diagram of coastal current systems in the Yellow Sea(Redrawn after references[6,9,14])**

1.辽南沿岸流;2.鲁北沿岸流;3.青岛—石岛近海的反气旋中尺度涡旋,青岛冷水团;4.山东南部沿岸的东北向流动;5.黄海西部沿岸流;6.苏北沿岸水;7.夏季苏北沿岸的北向流动;8.东北向扩展的长江冲淡水;9.台湾暖流前缘水;10.黄海暖流;11.朝鲜半岛西部沿岸流

1. Liaonan coastal current; 2. Lubei coastal current; 3. Mesoscale anticyclonic eddy in Qingdao-Shidao offshore, Qingdao cold water mass; 4. Northeastward current in Lu'nan coast; 5. Yellow Sea western coastal current; 6. Subei costal current; 7. Northward current in Subei coast in summer; 8. Northward extension of Yangtze River diluted water; 9. Taiwan warm current; 10. Yellow Sea warm current; 11. Korean peninsula western coastal current

Figure from Wei et al., (2011)

Wei, Q., Yu, Z., Ran, X., & Zang, J. (2011). *Characteristics of the Western Coastal Current of the Yellow Sea and Its Impacts on Material Transportation*. Advances in Earth Science, 26(2), 145–156. doi: 10.11867/j.issn.1001-8166.2011.02.0145

2. I am lost during the exploration of the wind and tidal effects on the anticyclonic circulation. Based on my understanding in physical oceanography and my careful revision on this manuscript, I am providing my points of view on how the wind and tides affects the studied anticyclonic circulation.

**Wind effects**. As shown in Figure A3 (please show the wind and wind stress patterns over the entire computational domain), surface currents in the west Yellow Sea are northeastward and eastward. There are at least three effects. (1) Waters piles up on the east side leading to barotropic pressure gradient forces pointing westward. Going down into deeper layers, as the existence of the Qingdao cold water mass, the temperature gradient is pointing westward at the west side of the cold water mass, generating a baroclinic pressure gradient force pointing

westward. So, at 25 m depth, both barotropic and baroclinic gradient forces are negative, as denoted by Figure 10b-10c (but I think the authors messed up the signs in these plots). Such **local wind effects** on the deep water will generate a northward current. (2) As the waters move offshore, a basin-wide (at least over west of 123E because over this region, depth is shallower in the west than in the east as shown in Figure 1b) upwelling system will be stimulated. This is the results due to Ekman transports. (3) Summer monsoon in the Yellow Sea and East China Sea should be similarly southwesterly (I am not quite sure if it is true in summer 2019). The currents from the southern boundary are highly affect by East China Sea coastal currents, which are highly affected by the same southwesterly monsoon. This is the **remote wind effects**. The authors, however, did not provide any discussion on it.

**Tidal effects**. (1) As shown in the Figures 8a-8b, northward coastal currents on the west side of the Qingdao cold water mass are weaken when tides are considered, while the currents (the YSCC) on the east side of the cold water mass turn to northward when tides are turned off. These patterns have been pointed out in the manuscript, which is good. The authors provide discussion on tidal-induced changes in barotropic and baroclinic terms. But what are the specific tidal effects that cause such changes? Residual tidal current? Tidal mixing? Or others? (2) In section 4.1, the authors aimed to discuss the tidal effects on the upwelling. However, discussion on changes in barotropic and baroclinic terms does not answer which tidal processes affect the upwelling system. To my understanding, upwelling systems are compensate current due to the surface water divergence across coastal shelf and the mass balance, but not due to the changes in barotropic and baroclinic conditions at deep layers. Instead, alike the upwelling, the changes in barotropic and baroclinic conditions at deep layers are also the results of changing surface current patterns.

**Detailed comments:**

1. Figure 1a. Please provide a reference for this plot. It seems like from this work:

Liu, Shichu, et al. "Interannual variation in winter thermal front to the east of the Shandong Peninsula in the Yellow Sea." Journal of Sea Research 193 (2023): 102370.

2. Line 176-177. Please show evidence to support the baroclinic effects on the differences in current patterns along vertical direction. If not, please remove this sentence.

3. The arrows in Figure 5 are hard to see.

4. Line 195-196. The geostrophic balance is hardly violated in the cold water area (122E-123E, 34.5N-36N), which is shown in Figure 5d and is also mentioned in Line 239.

5. Line 213-214. Should be "between the control and the no-tide experiment (Fig. 6)"

6. Line 219. As shown in Figure 8a and 8b, the velocity is greater in the no-tide experiment than in the control ones.

7. Line 221. Without tidal forcing east of $122°E$, the magnitude of the barotropic term also increase.

8. Line 219-214. I don't understand what the main conclusion or the main purposes of this paragraph is. I lost here. Please see the major comments 2 for the tidal effects.

9. Lines 224-237. Seems that the authors try explain why the YSCC is reversed when tides are off, but I didn't find the logic of for this paragraph and cannot come up with a clear conclusion.
Let me clarify my confusion.
On line 226-227, the anticlockwise gyre is the one around 124-126E, 34.5-26.5N. It is beyond the studied clockwise circulation around the Qingdao cold water mass.

Line 228-230, the norward current west of 122E is not the compensation currents but the northeastward current in the Lu'nan coast and the northward current in Subei coastal current. Please correct me if I am wrong.

Line 231-233, the "northward flow in the eastern part of the southern Yellow Sea" is around 125E, while the "southward flow in the west portion of the southern Yellow Sea" (or the YSCC) is at around 122-124.5E. Do they have any linkages when you compare them?

All the above changes are the so-called changes in broader-scale Yellow Sea circulation. But they still not answer what tidal processes affect changes in broader-scale circulation, and then affect the YSCC. Although the authors list reasons in 238-245 citing other works. But what the authors get from their simulation are still not well addressed.

10. Line 243, where is the mentioned "basin-scale cyclonic gyre"?

11. Lines 284-289. The analysis conflict with the signs shown in Figure 10.

12. Figure 10. Please double check the signs. For example, the positive (eastward) barotropic term in Figure 10b conflict with the westward barotropic term shown in Figure 5b.

13. Section 4.1. Please see the major comment 2 related to the discussion of upwelling. As shown in Figures 11a and 11b, surface currents (0-10 m) are eastward moving waters offshore. I would insist that the authors should focus on the adjustment of basin-scale mass balance rather than on the changes in barotropic and baroclinic terms over a transect or a profile (Figure 12) when linking the tidal effects and the upwelling system.

14. Line 345-346. My understanding on oceanic front is that front is determined by horizontal gradients of temperature or salinity or density, but not based on gradient on the x-z panel, especially for the Qingdao cold water mass, which has strong horizontal temperature gradient.

15. Line 354. As shown in Figures 12a -12b, the vertical friction is near zeros. So, at this location, pressure gradient forces balance with the Coriolis force, which means that the geostrophic balance is met.

16. Figure 12. Why do you show the momentum at depth=15 m when the rest of the analysis is based on the currents pattern on depth =25m?

17. Lines 374-375. Distinguish instead of extinct? This is an incomplete sentence.

18. Line 381-382. As I observed on Figures 13-14, 51.29% is for no-tide conditions, while 89.68% for no-wind conditions. Also, rounding to integer is enough.

19. Figure 13-14, are they for depth=25 m? Please update the captions.

20. Lines 424-426. The geostrophic balance maintains over the cold water mass.

---

## Author Response (AR2)

**Thank you very much for your effort and time to read our paper and help us improve it. The comments are very insightful and constructive for us. We have revised the manuscript point by point to the comments as listed below.**

**Review1 General comments:**

*This study investigated the formation and main drivers of the Qingdao cold water mass by using numerical simulations from a series of ensemble experiments. The study pointed out that the geostrophic balance is no longer applicable in the Qingdao cold water mass region due to the considerable friction terms. A seasonal anticyclonic circulation system was detected around the cold water mass. Wind and tides were found to be the main factors to this circulation. As for the method, the authors introduced an ensemble methodology, which is seldom applied and discussed in many numerical studies. I think, it is an advantage that can help this study to stand out. However, some major points still remain unclear and need to be resolved before the publication. Therefore, I recommend the publication of the manuscript after major revision.*

**Response to the comment:**

Many thanks for your helpful review, perspectives, and comments. We have changed the manuscripts according to your comments, please see the details below.

**Major comments:**

*1. The logic of the manuscript is unclear. I spent a lot of time trying to figure out how the authors organize the manuscript. My understanding is (1) that there is a seasonal anticyclonic circulation system observed around the Qingdao cold water mass; (2) that the geostrophic balance is no-longer applicable in coastal Qingdao and cannot be used for explanation of the formation of this anticyclonic circulation system; (3) that this anticyclonic circulation system mostly results from the balance of pressure gradient force, the Coriolis force, and the friction force with an emphasize of the non-negligible friction in the shallow water and (4) that the wind and tidal forcings contribute significantly to the evolution of this anticyclonic circulation system through the adjustments to the friction term. The authors, for example, put a part of discussion of wind and tidal impacts at the beginning (section 2.2), which, I think, is a scattered way of thinking and makes the readers hard to capture the main points of this study. Similar*

*problems were found in the abstract and conclusion sections (see detailed comments below). Another example of this erratic or scattered way of thinking is reflected in the orders of figures shown. Figure 2,3, and 5b, 5c should go to the discussion section, while figure 5 should be merged with figure 15.*

**Response to the comment:**

Thanks for your suggestions. We have moved the discussion of wind and tidal impact at the beginning to the discussion section in the updated manuscript. The figures mentioned above have been reorganized as well. The problems mentioned in the abstract and conclusion sections have also been modified, please see the replies to the minor comments.

*2. The manuscript is lack of in-depth discussion although a "Discussion" section is performed. Each subsection in section 4 seems like other result section without detailed quantification and comparison (against previous studies). Please see the detailed comments below.*

**Response to the comment:**

The wind and tidal forcing effects on the Qingdao cold water mass which were in the discussion section, now has moved to result section. And the discussion has been modified as four sub-sections: The existence of upwelling in the vicinity of the Qingdao cold water mass, statistic test, perspective and implication of the model results, limitations of the study. Additionally, we have added the comparison against previous studies. Please see lines 303-308, 309-315, 391-400:

*Lines 303-308: Previous studies have investigated upwelling and the vertical secondary circulation in the Bohai and Yellow Seas and revealed that upwelling usually occurs in shallow areas along coasts, such as the Subei area and the region near the Korean Peninsula (Q. Wei et al. 2019; Lü et al. 2010). Lü et al. (2010) explained the tidal effect on upwelling, indicating that the front in the control run generates a relatively large baroclinic pressure gradient compared with when the tidal forcing is turned off, which further triggers distinct upwelling. In this work, we found that such an explanation can also explain the upwelling around the Qingdao cold water mass.*

*Lines 309-315: In this study, we also find that upwelling occurs from the surface layer to the bottom layer near the Qingdao cold water mass. Additionally, a refined analysis of the factors influencing upwelling around the Qingdao cold water mass has not yet been performed.*

*Lines 366-369 As introduced in subsection 2.2, statistical tests have many applications in numerical climate simulations to separate and validate that the variation is caused by internal variability or changes in external forcing or parameterization (Conover 1999; Weisse, Heyen, and von Storch 2000; von Storch and Zwiers 1999; Livezey and Chen 1983; Zwiers and von Storch 1995). However, for the ocean regional simulation community, statistical tests are seldom applied.*

*Lines 391-400: Previous studies have focused mainly on the seasonal variation in the formation of the Qingdao cold water mass and its mechanism (Ho et al. 1959; Q. Zhang et al. 2002; 2004; Yu et al. 2006; 2005; Zhang et al. 2016; Huang, Chen, and Lin 2019) but not the seasonal anticyclonic circulation around it. Some previous studies have reported an anticyclonic circulation around the Qingdao cold water mass (F. Zhang, Mao, and Leng 1987). The numerical results reveal that the northeastward current induced by the summer southwesterly.*

*However, the link between the Qingdao cold water mass and the seasonal anticyclonic circulation has not been analyzed quantitatively. Based on the momentum balance, in this study, although there is a significant temperature gradient and a mild salinity gradient near the Qingdao cold water mass, the resulting density gradient from these temperature and salinity variations does not directly drive the observed anticyclonic flow structure, as the geostrophic balance is not maintained.*

*3. There are some findings or points listed in the manuscript that are contradicted to each other weaking statements. I have listed them below.*

**Response to the comment:**

Thanks for catching the problems. We have corrected and please see the replies below.

*4. The manuscript is lack of quantitative analysis and comparisons no matter in the results but also the discussion sections. This can largely weaken the convincibility of the findings.*

**Response to the comment:**

We have made modification as suggested, please see the replies in the minor comments.

**Minor Comments:**

*1. The title is not specific enough. Saying "the seasonal anticyclonic circulation" may be too general. Per my understanding, the authors are investigating the mechanisms of the evolutions of this circulation. So, it could be better to point out the main purpose of the study in the title.*

**Response to the comment:**

O.K. We have changed the title to "The evolutions of the seasonal anticyclonic circulation around the Qingdao cold water mass in the China marginal sea and its mechanism".

*2. Lines 14-30. The abstract is not concise enough and is lack of a "main clue" to guide the reader to rapidly capture the findings (please see the above comments).*

**Response to the comment:**

We have rewritten the abstract.

*3. Lines 19-20. The authors should point out directly what results in the anticyclonic circulation rather than saying the cool poor is not the main cause.*

**Response to the comment:**

We have improved the abstract, please lines 18-24:

*Lines 18-24: Seasonal circulation mostly results from the balance of the pressure gradient, Coriolis, and vertical friction forces. The no-tide and no-wind numerical simulation results suggest that when the tidal forcing is turned off, unrealistically strong currents appear and are caused by the decrease in vertical friction in the no-tide simulation. Moreover, the direction of the eastern side of the anticyclonic circulation is reversed. Furthermore, the seasonal southwesterly monsoon contributes to the magnitude of the anticyclonic circulation, especially in the western portion of the anticyclonic circulation. Additionally, upwelling occurs vertically around the Qingdao cold water mass and is influenced by tidal and wind forcings.*

*4.    Lines 22-24. Too verbose. Try to concise it.*

**Response to the comment:**

We have added how many of the grid points are statistical in the manuscript. 51.29% and 89.68% of grid points of no-wind and no-tide runs are proven to be local statistically significant.

*5. Line 105. 11 year? The simulation period of the climatological run is from Nov 2008 to Dec 2019.*

**Response to the comment:**

Sorry about this confusion. The climatological run ending time is 1st Nov. 2018. It is 10-year. After 1st Nov. 2018, we used the realistic surface forcings (NCEP CFSv2) to run the ensemble simulation. We have corrected the manuscript.

*6. Lines 127-134. This part should belong to the discussion section and same for figures 2–3.*

**Response to the comment:**

We have moved it to the discussion (section 4.3).

*7. Line 159. "Northeast" or "southeast"? Please double check. If "northeast", the corresponding temperature and salinity distribution should be shown, i.e., enlarge the coverage in Fig. 4.*

**Response to the comment:**

It is southeast, thank you for catching this problem.

*8. Lines 163 and 167. The location of the cold water mass center was mentioned twice but with different longitudes.*

Corrected.

*9. Lines 163–165. My understanding of the formation of cold water body is somehow slightly different from the explanation here. Firstly, during winter, water column is homogenized forming cold water from surface to bottom. Secondly, as in early summer, fast heating on the ocean surface induces strong and rapid stratification sealing the cold water below the thermoclines. Thirdly, vertical mixing due to wind disturb or upwellings is not strong enough to homogenized the water column, which results in the maintenance of near-bottom cold water mass. So, it may not be appropriate to emphasize only the role of thermocline to the cold water mass formation. In addition, citations are needed here as there is no clue shown in Fig. 4 that thermocline leads to the formation of the cold water mass.*

**Response to the comment:**

The reviewer describes the formation of the cold water body more thoroughly, we have deleted the sentence which emphasizes the role of thermocline, and we have also added the reviewer's understanding of the formation of the cold water body in the manuscript. Thank you.

*Lines 142-147: At this time, fast heating on the ocean surface induces strong and rapid stratification, sealing the cold water below the thermoclines. The vertical mixing due to wind disturbance or upwelling is not strong enough to homogenize the water column, resulting in the maintenance of a near-bottom cold water mass. In summary, the Qingdao cold water mass starts to merge in April, continues to develop in May, is accompanied by a strong horizontal temperature gradient, and disappears in June. The cold water mass center is at 122.20°– 122.40°E, 36°–36.15°N, and the shape of the cold water mass has a northeast–southwest orientation.*

*10.    Figure 5 should be merged with Figure 15, as they contain largely overlapping information. Arrows in figure 5 are not clear enough.*

**Response to the comment:**

Sorry about this. We have replotted Fig. 5 (now Fig. 12 in the new manuscript). Figure 15 in the old manuscript has already been deleted since it contains largely overlapping information.

*11.    Lines 183–185. Is the southeasterly wind strong enough to induce vertically-homogenized the northward currents?*

**Response to the comment:**

We can compare the Fig.3 (b) control run result in May with Fig. 8 (d) no-wind result. There is no obvious northern current in the no-wind run. Given on this model result, we conclude that southeasterly monsoon is causing the northward currents (25m layer).

*12.    Line 188. Could you also show the June pattern to support this statement?*

**Response to the comment:**

Please see Fig. 3 in the manuscript, the circulation pattern in June is shown there.

*13.    Lines 190–192. This is an incomplete sentence. "As the water depth is shallow in the western portion of the anticyclonic circulation and the dominant southeasterly monsoon…" This part is incomplete. This is a conclusion-like statement. Model evidence are needed to support this causality.*

**Response to the comment:**

We have deleted this sentence, since it is confusing for the readers.

*14.   Lines 200–201. I cannot see velocity increases with depth. Perhaps the authors need to replot the Figure 7 showing 5 separate 2D panels rather than using a 3D plot. The Figure 7 shown makes current arrows hard to compare. Also, the panels do not align with the corresponding depth, e.g., the first panel does not align with depth 25 m and so on.*

**Response to the comment:**

We have replotted Fig. 7 in the old version, with 2D panels. Please see Fig. 4 in the latest version. We found it is true that it is not very obvious that velocity variation with depth, so we have deleted this sentence. This is a sentence describing the vertical variation of the velocities, which does not influence the main conclusion of this paper if we delete it.

*15.   Equation (1)–(2). Please also show the momentum equation in the vertical direction. Also please decompose the force terms in (1) and (2) into the term shown in Figure 8 and put all force terms to the right-hand side. It is important to provide the readers with the detailed mathematic expressions of the force terms shown in Figure 8 and others.*

**Response to the comment:**

O.K. We have added a momentum equation in the vertical direction, please see equation (3). All force terms are put to the right-hand side. The detailed mathematic expressions of the force terms are explained in Lines 185-189.

*Taking the zonal direction as an example (Equation (1)), the total pressure gradient term is $-g\frac{\partial \zeta}{\partial x} - \frac{1}{\rho}\frac{\partial}{\partial x}\int_z^\zeta \rho g dz$, which comprises the barotropic pressure gradient force $-g\frac{\partial \zeta}{\partial x}$ induced by sea level and the baroclinic pressure gradient force $-\frac{1}{\rho}\frac{\partial}{\partial x}\int_z^\zeta \rho g dz$ induced by density; the Coriolis force $fv$; the vertical friction term $\frac{\partial}{\partial z}\left(K_m \frac{\partial u}{\partial z}\right)$; the local velocity time variation term $\frac{\partial u}{\partial t}$; the horizontal advection term $-(u\frac{\partial u}{\partial x} + v\frac{\partial u}{\partial y})$; the vertical advection term $-w\frac{\partial u}{\partial z}$; and the horizontal friction term $F_u$.*

*16.   Lines 215–216. Although horizontal friction is usually negligible in open ocean, in the coastal region this friction may not be negligible. The Qingdao cold water mass occurs*

*quite nearshore. So, please provide quantitative comparisons of all friction terms before removing any of them out of the analysis and discussion.*

**Response to the comment:**

We have checked the horizontal friction term intensity, which is 1-2 orders smaller than the main terms, namely, the Coriolis force, barotropic pressure gradient force, baroclinic pressure gradient force, and vertical friction force. If we plot the horizontal friction term distribution with the same colorbar as that for other main terms (the same colorbar as Fig. 5), the whole pattern for horizontal distribution is almost white, since it is much smaller than $0.1e-5$ $m/_{s^2}$. To be quantitative, the magnitude of the area-mean horizontal friction term and Coriolis force at the layer 25m are $1.15 \times 10^{-8}$ $m/_{s^2}$, $7.68 \times 10^{-7}$ $m/_{s^2}$, respectively. At the depth of 40m, the magnitude of the area-mean horizontal friction term and Coriolis force are $3.35 \times 10^{-9}$ $m/_{s^2}$, $6.01 \times 10^{-7}$ $m/_{s^2}$.

*17.    Lines 216–218. Lack of evidence. At least, the difference of barotropic gradient force and the sum of baroclinic gradient force, Coriolis force, and vertical friction force are needed before addressing the balance of these terms.*

**Response to the comment:**

We have added a diagram showing the sum of the barotropic gradient force, baroclinic gradient force, Coriolis force and vertical friction force is shown in Fig. A3.

*18.    Line 219. The "southwesterly wind stress" contradict against the "southeasterly monsoon" in Line 183. In the northern hemisphere, the southeasterly wind most likely induces southerly wind stress rather than southwesterly ones. Please overlap the wind patterns in Fig. A2.*

**Response to the comment:**

Thanks for catching this point. We have overlapped the wind patterns in Fig. A2. Additionally, we have changed the direction of wind stress to "southwesterly wind stress" to make it consistent.

*19.    Lines 220–222. It is more like a hypothesis or an inference without necessary evidence. Also I doubt that the horizontal friction also considerably affects the Qingdao cold water mass which locates quite nearshore.*

**Response to the comment:**

Here is the same comment regarding the horizontal friction, please see the reply to comment 16.

*20.     Figure 8. Please also show the mathematic expression of each term in each subplot because the signs of these shown force terms are also important to readers.*

**Response to the comment:**

We have added the mathematic expression as suggested, please see the Fig. 5 and other momentum diagnose related diagrams as well.

*21.     Line 226. Format is wrong.*

Corrected.

*22.     Figure 9. Same suggestion as for Figure 8*

We have added the mathematic expression as suggested.

*23.     Lines 235. The difference of pressure gradient force and the sum of Coriolis force and friction force is needed to support this statement.*

**Response to the comment:**

Thank you for pointing out this problem. We have added a new diagram Fig. 1* showing the difference of pressure gradient force and the sum of Coriolis force and friction force. We find a large value to the east of $121.75°E$, there is a large value in the surface, so have mentioned this in the lines 212-213.

[Figure]

Figure A1. Vertical distribution of the difference in the pressure gradient force $-g\frac{\partial \zeta}{\partial x} - \frac{1}{\rho}\frac{\partial}{\partial x}\int_{z}^{\zeta}\rho g dz$ and the sum of the Coriolis force **fv** and friction force $\mathbf{F_u}$ along the 35.5°N profile of the control run.

*Lines 212-213: east of 121.75°E, the Coriolis force and vertical friction has high values, caused by the large current magnitude and strong wind*

*24.    Lines 239–243. Lack of horizontal friction. See above comment.*

**Response to the comment:**

Please see the reply to the comment 16.

*25.    Lines 244–253. I think this part is a very important finding of this study and also the reason why the author wanted to dig deeper in the wind and tidal impacts in the discussion section. I suggest the authors provide more explanations and details here.*

**Response to the comment:**

Thanks for the confirmation. We have rewritten this part, please see below.

*Lines 225-245: The geostrophic balance describes the equilibrium between the horizontal pressure gradient and Coriolis forces. Temperature and salinity distributions directly influence*

*water density, altering the pressure gradient and affecting oceanic geostrophic currents. Near a Qingdao cold water mass, there are significant temperature and mild salinity gradients. However, owing to the lack of geostrophic balance, the density gradient caused by the temperature and salinity gradients associated with the Qingdao cold water mass is not the direct cause of the anticyclonic flow structure. This trend explains why, under the geostrophic balance, a cold center in the Northern Hemisphere would typically be associated with a cyclonic (counterclockwise) circulation, whereas the flow around the Qingdao cold water mass would exhibit a clockwise circulation pattern. The friction induced by tidal forcing disrupts the original geostrophic balance.*

*To some extent, the seasonal anticyclonic circulation is the dynamic reason for the Qingdao cold water mass. As mentioned in subsection 3.2, this type of anticyclonic structure prohibits heat transfer between the cold temperature center and the surrounding water column (Huang et al., 2019), but the seasonal anticyclonic circulation is not caused by the special temperature and salinity structure around the Qingdao cold water mass.*

*Since the geostrophic balance is not the direct reason for the circulation, other causes of the seasonal anticyclonic circulation around the Qingdao cold water mass should be identified. Tides play a significant role in the circulation processes of regional seas, contributing substantial amplitude, momentum, and energy flux. Early studies (B.H. Choi 1980; Byung Ho Choi, Eum, and Woo 2003; Moon, Hirose, and Yoon 2009; C. Xia et al. 2006) have shown that tidal characteristics in the Bohai and Yellow Seas are highly complex and play a significant role in local circulation and modulate dissipation, vertical mixing, and tidal energy. Owing to the large input of tidal energy and the combined effects of bottom friction related to shallow shelf topography, the dissipation of $M_2$ tidal energy (the principal lunar semidiurnal constituent) in shallow regions reaches 180 GW, accounting for approximately 11.1% of the global total (i.e., 2 TW). Additionally, the local seasonal monsoon may influence the structure of the anticyclonic circulation around the Qingdao cold water mass. Consequently, no-tide and no-wind simulations were conducted to analyze the impacts of tidal forcing and wind forcing on anticyclonic circulation.*

*26.  Line 244–246. It is not surprised that the geostrophic balance is not satisfied as it is a state of large-scale ocean flows in open ocean. In the coastal region, the authors may want to focus more on the theories of boundary layer. So then the friction is very important and may need more analysis and discussion.*

**Response to the comment:**

Since topography and bottom friction play a role in the coastal area, it is worthwhile to further explore boundary layer theories. We put it in the discussion section 4.4 limitations of the study and will have another analysis about smaller scale features around the seasonal anticyclonic circulation around the Qingdao cold water mass. In this paper, we limited ourselves to the relatively large-scale feature of such anticyclonic circulation.

*Lines 414-416: Also, this study focused on submesoscale horizontal and vertical circulation around the Qingdao cold water mass, and further exploration of boundary layer theories is worthwhile since topography and bottom friction play a role in the coastal area. Future analyses will investigate the boundary layer variation caused by tidal forcing.*

*27.    Lines 270–271. But along the anticyclonic circulation, friction is lower in the no-tide experiment.*

**Response to the comment:**

We try to express that in most of the area in Fig. 12 (d) (in the previous version before revision), when the tidal forcing is turned off, the friction intensity decreases, especially, where the depth is shallow, say, to the west of 122°E. If looking at the magnitude of the circulation, it is significantly increased to the west of 122°E, accordingly, the vertical friction decreases when the tidal forcing is turned off. But since this sentence causes confusion, we decided to delete it.

*28.    "…and Coriolis terms change direction…"? The force direction seems not change. For example, the changes in Coriolis force in the no-tide experiment is at an order of magnitude of 10E-6, while the Coriolis force in the control experiment is at an order of magnitude of 10E-5.*

**Response to the comment:**

Thanks for pointing out this problem. You are right. This diagram shows the difference of the magnitude. We have plotted the Coriolis force for control run and no-tide run. The result show that the negative value of the difference is because of the magnitude of the Coriolis force in no-tide run is larger than control run. We have changed the expression in the manuscript, please see lines 258-261. Additionally, the order of the magnitude of the Coriolis force in control run and no-tide run are still at the same level ($10^{-6}$).

[Figure]

Figure. 2*. The difference of the magnitude of the Coriolis force in the control and no-tide runs. (a) The magnitude of the Coriolis force in control run; (2) the magnitude of the Coriolis force in the no-tide run; (3) the difference of magnitude (control run minus no-tide run).

*Lines 258-261: Without tidal forcing east of 122°E, the magnitude of the Coriolis terms increases (Fig. 9a), which is combined with a change in the circulation direction, indicating the significance of the tides around the anticyclonic circulation. In addition, the magnitude of the current in the entire region significantly increases, which is unrealistic, indicating that the dissipation and friction caused by tidal forcing are important for modulating circulation in anticyclonic areas.*

*29.    Figure 11b. Why control-no_tide? It different from other difference terms. Please make the calculation consistent.*

**Response to the comment:**

We have changed it to the control run minus no-tide run, as suggested. Please see Fig. 8.

*30.    Lines 310–312. I did not understand this sentence.*

**Response to the comment:**

We have reorganized this sentence, please see lines 293-295.

*Lines 293-295: With a southwesterly wind (Fig. A3), to the west of 121°E, the surface elevation is lower in the control run (Fig. 10). Accordingly, the barotropic pressure gradient force, which is the force exerted due to horizontal pressure differences caused by variations in sea surface height, also decreases when the wind forcing is turned off (Fig. 9g).*

*31.    Lines 312–314. The causality is misleading. It is not the decreases in water temperature that cause an increase in baroclinic pressure gradient force but the weaker mixing processes when wind is eliminated.*

**Response to the comment:**

Thank you for pointing out this problem, we have rewritten this sentence, please see lines 295-299.

*Lines 295-299: In contrast, in the no-wind run, the temperature of the Qingdao cold water mass decreased by 2°C (Fig. 11) in the absence of mixing between the warm surface water and cold bottom water. These weaker mixing processes cause an increase in the baroclinic pressure gradient force in the area around the Qingdao cold water mass (120.5°-121.75°E, 35.5°-36.5°N) when the wind is eliminated.*

*32.    Lines 321–323. I did not observe a downwelling system from figure 5a.*

**Response to the comment:**

Thanks for the correction. We have corrected the expression of downwelling.

*33.    Line 325. "For the western portion (122.625–123oE)" or "For the eastern portion (122.625–123oE)"?*

**Response to the comment:**

Thanks for catching it. It should be the eastern portion. We have corrected it.

*34.    Line 326. The tidal mixing font is a new term here (not mentioned in previous sections). So, where is this tidal mixing font? Seems like it is a part of the decomposition of mixing. I am not sure. Please convince me.*

**Response to the comment:**

Considered comment 34 and 37, we have reorganized the tides effects in section 3.5 and 4.1.

*35.    Lines 330–331. Below 20m, all flows seem eastward.*

**Response to the comment:**

To be more precise, we have added "over 25m".

*36.    Lines 332–341. The part is not convincible to me. I would suggest the authors compare the difference of current velocity between control and no-tide runs and the difference between control and no-wind runs.*

**Response to the comment:**

Sorry for the unclear visualization for Fig. 5 in the old version, we have plotted it (Fig. 12 in the new version). It seems that this time it is clearer that the "upwelling cycles occur in the red and blue rectangles in the control run, but in the no-wind experiment, the magnitude of upwelling cycling in the red rectangle decreases, as shown in Fig. 12".

*37.    I think one transect is enough. Lines 349–351. There is still temperature font found in Figure 5b, which means that when tides are excluded, the cold water mass maintains at the similar location. So, how can the tidal impacts be important to the cold water mass? As I observed, the impacts of winds outperform tidal impacts, as in Figure 5c, the location of the cold water mass shifts westward to 121.5oE.*

**Response to the comment:**

We have added a sub-section about the tidal forcings influence on the Qingdao cold water mass and the seasonal anticyclonic circulation in Discussion, please see lines 334-344.

*Lines 334-344: Tidal forcing plays a crucial role in shaping the characteristics of the thermocline, including the width of the front and the maximum temperature gradient. The front serves as a narrow transitional zone that separates stratified deep water from well-mixed shallow water, resulting in significant temperature (or density) differences across it. Owing to the mixing effects induced by tidal forces at the seafloor, isotherms near the bottom are predominantly orthogonal to the seabed topography at depths below 20 meters. Furthermore, in the control run, the front is both compressed and elevated. This study defines the front as the region where the magnitude of the horizontal and vertical temperature gradient ($\sqrt{\left(\frac{dT}{dx}\right)^2 + \left(\frac{dT}{dy}\right)^2}$) exceeds $1°C/m$. Under tidal forcing, the front is observed to exist within a depth range of approximately 2–20 m, whereas in the absence of tidal influences, the front extends from 2–25 m. A wider front correlates with a reduced intensity, indicating that the front is weaker in simulations without tidal effects. Additionally, the maximum vertical temperature gradient is greater in the control run ($1.78°C/m$) than in the no-tide simulation ($1.73°C/m$). Therefore, the front intensity decreases in the no-tide run, compared with the control run.*

**Review2 General comments:**

*I appreciated the efforts the authors have made to reply to my comments. I have several follow-up suggestions and questions for this version. The line number refers to the tracked version.*

**Response to the comment:**

We appreciate the reviewer's time and effort to read and give comments on our manuscript again.

**Minor Comments:**

*1.      I would suggest adding a sentence like "Yet, it is unknown if the formation of this anticyclonic circulation is related to the Qingdao cold water mass" at the end of line 60 to state the research gap.*

**Response to the comment:**

Thanks, we have added it as suggested.

*2.      please explain "internal variability" (line 70) and "weather variations" (line 109).*

**Response to the comment:**

O.K. We have added the internal variability in lines 117-121.

*Lines 117-121: Internal variability is defined as the part that cannot be linked directly to the external forcing but is caused by unforced variability generated within a system. The unforced variability is unprovoked and chaotic. Since this paper focuses on the external forcing imprint on the seasonal anticyclonic circulation around the Qingdao cold water mass, more details about the concept of internal variability can be found in previous studies (Tang, von Storch, and Chen 2020; Lin et al. 2023).*

For "weather variation," it may be more accurate to use the term "interannual variations." In realistic external forcing, daily weather typically differs from the same day in another year, but this aspect of variability is removed in the climatological forcing.

*3.    the sentence "The model…" in line 112 is repeated with line 101.*

**Response to the comment:**

Thanks for catching the problem, we have corrected.

*4.     please add more explanations on "random effects" and "randomness" in the section of Ensemble experimental design.*

**Response to the comment:**

O.K. We have added what "random effects" refer to and the cause of randomness.

*5. The "low-salinity" in line 157 should be "moderate-salinity"?*

**Response to the comment:**

Corrected. Thanks!

*6. The center of the cold water mass is 122.4˚E (line 163) and 122.75˚E (line 167). It's better to be consistent or describe the center location as a range.*

**Response to the comment:**

We have corrected. Additionally, we have changed it into a range "122.20°-122.40°E 36°-36.15°N".

*7. line 179, the locations of the red and blue boxes in Fig. 5a don't match the caption "121.5-122" and "122.625-123". It's also not easy to see the arrow direction. In addition, what's behind the westward shift of the center of the cold water in the no-wind run (Fig. 5c)?*

**Response to the comment:**

Thanks for catching the typos. The caption should be 122.375-122.5°E and 122.625-123°E.

We have replotted the diagram to see the arrow direction.

We did not understand the meaning of the last sentence. Could give us more hints about "what's behind the westward shift of the center of the cold water in the no-wind run (Fig. 5c)?"

*8. "(not show)" in line 189 should be "(Fig. 6c)"?*

**Response to the comment:**

Indeed, you are right. We actually show the diagram. Now we have made it consistent.

*9. cite Fig 9d-f in lines 274-276.*

**Response to the comment:**

Corrected. Thanks.

*10. line 335, there is no Fig 15f.*

**Response to the comment:**

Fig. 15 has been deleted because of the overlap information of the vertical temperature and salinity distribution.

*11. Question about the direction of the Coriolis force in Fig 9a (I may have some misunderstanding so hope the authors could help explain it). Between 121˚E-122˚E, the current at 25m flows mainly toward the north (v>0, u≈0) averaged in May (Fig. 6b), so the Coriolis force -fv will be negative but it is positive in Fig 9a (?).*

**Response to the comment:**

Sorry for causing such confusion, we should have added the mathematic expressions of each term earlier. Please see the caption. The Coriolis force for this diagram is expressed as $fv\vec{\imath} - fu\vec{\jmath}$, so the Coriolis force is mostly positive.

---

## Author Response (AR3)

**General comments:**

*I can see the improvement of the manuscript. However, I am still confused on some critical points. Therefore, I recommend the publication of the manuscript after major revision.*

Many thanks for the reviewer's efforts to review and give comments. We have replied all the comments, please see the details below.

**Major comments:**

*1. The logic of the manuscript is still unclear. In section 3.1 and 3.2, the authors showed the temperature and salinity features and circulation pattern of the cold water, trying to demonstrate the evolution of the cold water and the related mechanisms. However, there is a lack of supported evidence. I will list those points out later. In section 3.3, the authors tried to prove that geostrophic balance is not applicable for the maintains of the anticyclonic circulation in lower layer in May due to the non-negligible friction in the shallow water. Isn't it obvious in a shallow shelf? I am not quite sure why the authors spent a section to demonstrate this question. It may be better to merge this section into section 3.1 and 3.2? In section 3.4, the authors demonstrated that the anticyclonic circulation is NOT caused by the cold water mass which induces changes in baroclinic conditions. So,why not just to demonstrate which factor cause the anticyclonic circulation? As I observed from the figures shown in the manuscript, there are at least two compensate currents (secondary currents) which, I think, are important for the anticyclonic circulation development and the cold water mass evolution, i.e., the landward current at near bottom layers (westward current) and the upwelling system. Also, the background current system outside the shown region is very important. In section 3.5 and 3.6, the tidal and wind effects on the anticyclonic circulation are studied. But I am not quite satisfied with the explanation provided. I will list my concerns later. In section 4.1, the upwelling effects on the cold water mass is discussed, which I think is very important and should be move to section 3.3. More focus should be put to this part. And I found it is hard to follow the tidal effects on the upwelling and the mixed layers (lines 334–360). I will list my questions. Section 4.2 is too short to be a section alone. Instead, as supporting evidence for the tidal and wind effects, it may be better to merge this section into the sections discussing the tidal and wind effects (section 3.5 and 3.6). The same problem raises to section 4.3.*

Since the mechanisms driving the evolution of the Qingdao Cold Water Mass (QCWM) have already been extensively discussed in previous studies (see Introduction), this paper does not aim to revisit that topic. Instead, our focus is on the evolution of the seasonal anticyclonic circulation around the QCWM. Please note that the QCWM is not the direct reason of its formation. A novel aspect of this paper is the evolution of the seasonal anticyclonic circulation around the Qingdao cold water mass.

To our knowledge, the momentum analysis has not used before in the seasonal anticyclonic circulation around the Qingdao cold water mass to test if the geostrophic balance is satisfied or not, so we think we need to proof it in this area. While we believe it is important to include this analysis, we have streamlined the manuscript in response to the reviewer's comment about the section being too lengthy. As a result, we have removed subsection 3.4 and retained only subsection 3.3.

The compensation current (second currents) and upwellings are described in subsection 3.5 and 4.1.

Furthermore, the application of t-test analysis to distinguish between internal variability and external forcing in regional ocean model simulations is a new addition to this study. We emphasize the importance of the t-test and introduce the concept of ensemble simulations in this context. Subsection 4.2 provides a comprehensive discussion of these points.

For clarity and thematic consistency, we prefer to keep this material as an independent section. Addtionally, we have also expanded subsection 4.2 to provide more detailed explanations.

*2. I am a bit confused as to why the authors devoted so much effort to comparing all the forcing terms. Based on Figure 5, the barotropic term appears to dominate in the near-bottom layer, which I believe is likely due to water piling up on the eastern side of the domain under the influence of the southwesterly monsoon. As a result, the landward flow in the near-bottom layers along the coastal region is primarily driven by this pressure gradient force, with minimal influence from other forcing terms. This aligns with what I mentioned earlier regarding the compensating current in my previous comment. So, it seems that too much discussion on other terms (baroclinic, Coriolis, friction) is not necessary. Could you please clarify the rationale and necessity for such extensive discussion of these terms (as presented in Figures 5, 6, 7, 9, and 13)?*

We have rewrote the subsection which describes the wind effects on the seasonal anticyclonic circulation around Qingdao cold water mass by explaining it from the compensating current driven by barotropic pressure gradient force aspects.

*Please see lines 273-304:*

*Xu and Zhao (1999) demonstrated the effect of wind on the seasonal anticyclonic circulation around the Qingdao cold water mass using a two-dimensional numerical model, but a more thorough discussion is needed. Therefore, a no-wind experiment was conducted to examine the effect of wind on the seasonal anticyclonic circulation structure. The results show that wind is a dominant driving force for clockwise circulation. The general magnitude of the current weakens, particularly on the western side of the clockwise circulation (northward current); conversely, the influence of wind on the eastern side of the circulation is minor, as shown in Figs. 6b and 6d. The direction and magnitude of the eastern side are similar to those of the control run.*
*The effect of wind on the seasonal anticyclonic circulation structure around the Qingdao cold water mass can be understood as the seawater piling eastward, under the impact of the southwesterly monsoon. Hence, the landward current (westward) at the western side of the seasonal anticyclonic circulation structure around the Qingdao cold water mass is primarily driven by the pressure gradient force, with minimal influence from other forcing terms. To prove this phenomenon, we examine the surface elevation fields from both the control and no-wind experiments (Fig. 9). The results clearly show that under southwesterly wind forcing (control run), surface waters accumulate to the east. In contrast, the no-wind experiment shows a much flatter sea surface. This finding indicates that the wind-induced water piling on the eastern side is responsible for the enhanced barotropic pressure gradient force. Further diagnostics of the momentum balance around the seasonal anticyclonic circulation confirm that the cross-shelf barotropic pressure gradient force is the dominant term driving the bottom flow along the coast.*
*In Fig. 10, we compare the spatial distributions of the momentum terms in the control (top row) and no-wind (bottom row) experiments. These terms are derived from the x-direction momentum equation, which is particularly relevant since the barotropic pressure gradient force is predominantly directed in the zonal direction. In the control experiment, a strong*

*barotropic pressure gradient is established, corresponding to a pronounced landward current at the western side of the anticyclonic circulation structure surrounding the Qingdao cold water mass. Conversely, the no-wind experiment shows much weaker pressure gradients in the same region. The Coriolis, baroclinic pressure gradient, and vertical friction terms are relatively weak in both experiments. These results support the interpretation that the southwest monsoon induces water piling eastward, thereby establishing a stronger barotropic pressure gradient that dominates the bottom-layer momentum balance and drives the landward current in the control run on the western side of the seasonal anticyclonic circulation; these findings differ from those of the no-wind run.*

*Beyond the local coastal dynamics around the Qingdao cold water mass, previous studies have demonstrated that the wind forcing plays a key role in shaping the large-scale summertime circulation in the Yellow Sea. For example, a wave–tide–circulation coupled model is used to reveal a three-dimensional structure characterized by wind-driven surface flows and compensating near-bottom currents (C. Xia et al. 2006). This basin-scale mechanism is consistent with our findings, suggesting that the landward bottom flow observed in our control experiment is not only locally forced but also part of a broad wind-driven circulation system across the Yellow Sea.*

*3. There are still some findings or points listed in the manuscript that are contradicted to each other.*

Corrected.

**Detailed comments**:

*1. Figure 1. Please overlap the bathymetry used in the model and also provide the coverage of the entire computation domain.*

We have added a diagram in Fig. 1b with the coverage of the entire computation domain and the bathymetry in the model.

*2. Section 2.2. There is a lack of description of the lateral boundary conditions and riverine forcings, which I think are very important to shelf dynamic simulations.*

We have added the information about the lateral boundary conditions and river forcings.

*Lines 96-97: The Huanghe, Huaihe, and Haihe are considered, and the river discharges are sourced from the "China Sediment Bulletin (2019)".*

We use the tidal elevation to introduce the effect of the tides.

*Lines 90-92: We use the tidal elevation to introduce the effects of the tides. The tidal elevation forcing comprises eight major tidal components ($M_2$, $S_2$, $N_2$ $K_2$, $K_1$, $O_1$, $P_1$, and $Q_1$) derived from the TPXO8 database (Egbert and Erofeeva 2002).*

*3. Line 103. Please use the specific years here, e.g., use November 1st, 2017 instead of November 1st of the 9th year. The latter expression may confuse the readers if they count the 2008 as the 1st year.*

We have changed it according to the suggestion (lines 108-109).

*4. Lines 141–142: Evidence is needed to support the claim that the merging of warm and cold water (i.e., the deformation of the cold water mass) is primarily due to increased solar*

*radiation and not other factors, such as mixing caused by the sustained monsoon. Additionally, doesn't surface heating contribute to maintaining the cold water mass near the bottom by reinforcing strong stratification?*

Since the evolution and mechanism of the Qingdao Cold Water Mass have been discussed in previous studies and are not the main focus of this paper, we have added a reference to support our description (lines 147–150).

*5. Lines 142–145. Although the author added my understandings on the cold water mass formation, it is just the knowledge from my previous studies in physical oceanography. The authors need to provide evidence to support this statement. I see that Figure 12 is a helpful clue. And also, the author may need to add more plots in the appendix showing the evolution of this cold water mass from April to June (may be evolution of the transect as depicted in Figure 12?). So, it is better to move Figure 12 to here.*

Figure 12 was initially placed in the Results section. In the last round of review, the reviewer recommended relocating it to the Discussion section. While we are open to adjusting the structure as suggested, we would appreciate further clarification.

*6. Lines 141–142 contradict to Lines 142–145. The former one emphasize the importance of solar radiation, while the latter emphasize the role of mixing and upwelling.*

We deleted the sentences, since it causes confusion and the mechanism of the Qingdao cold water mass is not the main topic of this paper.

*7. Section 3.2. Please confirm the direction of the summer monsoon. According to the Figure A3, it is southwesterly wind, but southeasterly wind is used during the discussion, e.g., Line 166. The inconsistence is found almost all over the manuscript. Please double check.*

Thank you for catching it. Corrected.

*8. Lines 159–161. The southwesterly wind induces the northward current on surface, but the authors indicate that the near bottom northward current is also induced by this wind pattern. I doubt that. This is the reason why I asked, "Is the southeasterly wind strong enough to induce vertically-homogenized the northward currents?" during the last round of revision. It is better to show the current pattern from surface to near bottom. Now, I would argue that it is the compensate current that induces the northward current near bottom but not the direct effects from the surface winds.*

We have rewritten the discussion regarding wind effects on the circulation around the Qingdao cold water mass area. Please see the reply to the major comment 2.

*9. Lines 166–167. This sentence is an incomplete one grammatically. The "southeasterly" contradicts to the previous description and Figure A3. The causality shown confused me. Why do the shallow water and southeasterly monsoon lead to stronger current at the west part of the anticyclonic circulation? Here, the authors may miss the boundary effects but only focusing on the local wind effects. So, how does the current system like over the entire computational domain?*

Ok. We had added a discussion about the other researcher's publication on the effect of wind forcing effects on the entire computational domain. The previous publication is consistent with our new statement that suggests that the landward bottom flow observed in our control

experiment is not only locally forced but also part of a broader wind-driven circulation system across the Yellow Sea.

*Please see lines 298-304:*

*Beyond the local coastal dynamics around the Qingdao cold water mass, previous studies have demonstrated that the wind forcing plays a key role in shaping the large-scale summertime circulation in the Yellow Sea. For example, a wave–tide–circulation coupled model is used to reveal a three-dimensional structure characterized by wind-driven surface flows and compensating near-bottom currents (C. Xia et al. 2006). This basin-scale mechanism is consistent with our findings, suggesting that the landward bottom flow observed in our control experiment is not only locally forced but also part of a broad wind-driven circulation system across the Yellow Sea.*

*10. Lines 171–174. What is the purpose to show the circulation below 25m?*

We would like to show a vertical distribution below the anticyclonic. However, we are open to removing it should the reviewer consider it redundant.

*11. Line 174–175. I don't think the baroclinic pressure gradient forces are contribution to the vertical structure of the current patterns near bottom. As I observed from the Figure 4, current patterns are quite similar over these near bottom layers (northwestward and northward). Isn't is due to the landward compensate currents or the current from the south boundary? A well-known Subei current system usually intrudes into the Yellow Sea in summer and merge with the cold water mass. That is, a larger picture of current systems needs to be discussed.*

Sorry for this confusion, lines 174-175 has been deleted. We have rewritten the wind impacts on the horizontal circulation in section 3.5.

*12. Lines 175–176. I don't understand why the anticyclonic circulation contribute to the formation of the cold water mass near bottom? Does it contradict to Lines142–145?*

Sorry for the confusion, we have deleted this sentence.

*13. Line 194–195. The vertical friction shown in figure 5 is the friction at depth of 25m. So, the wind stress at the surface layer does not contribute to this friction term but the current speed gradient around the 25 m depth does.*

We have deleted this sentence.

*14. Lines 195. I doubt that the surface wind can impact the current pattern near bottom through direct dragging. If so, the whole water column can be well mixed with not cold water exists at near bottom.*

We have deleted line 195 and wind impact on the horizontal circulation has been rewritten in Subsection 3.5.

*15. Lines 190–199. I see that the discussion is conducted mixing the current patterns at surface and near bottom layers, which should be discussed separately as they should not be the same as shown by Figure 3 and Figure A3.*

As suggested by the reviewer, the discussion of the geostrophic balance is too long. Lines 190-199 has been deleted.

*16. Line 200. The expression of the vertical friction force is wrong and same for Line 219.*

Thank you for catching it. Corrected.

*17. Section 3.4. I don't think discussion on geostrophic balance is useful for linking the anticyclonic circulation and the cold water mass.*

As suggested, we have deleted the previous Section 3.4 in the revised manuscript.

*18. Line 227. The term "significant" has statistical meaning. So, "significant" usually come along with statistical tests. Please use another term. Please check throughout the manuscript for this issue.*

We have changed the "significant" to obviously or delete it.

*19. Lines 227–232. The geostrophic balance should not be the theory discussed here as in this region, wind forcing cannot be ignored. Instead, it is the wind stress that causes the pressure gradient forcing (higher SSH on the east than on the west evident by the westward barotropic term) and the compensate current in lower layer but not the violation of geostrophic balance by tidal forcing. The explanation here conflicts with my knowledge. Holp the authors can convince me if I am wrong.*

This part has been deleted.

*20. Lines 233–236. What does the "dynamic reason" refer to? And I don't see the anticyclonic circulation contribute to the evolution of the cold water mass, rather, based on my study on the figures show, both of which (anticyclonic circulation and cold water mass) are the results of the compensate currents induced by the surface wind forcings. Again, please convince me if I am wrong.*

Since the section 3.4 has been deleted, this sentence is not in the manuscript now.

*21. Section 3.5. In-depth discussion is lacking. I suspect that the elimination of tidal forcings results in the changes of the background current systems which affect the anticyclonic circulation near Qingdao. The authors, however, only provide the description of how the anticyclonic circulation changes due to the absence of tides. Additionally, when removing the tidal forcing (that is in the no-tide experiment), how the boundary conditions are configured? The tidal forcing here should be the tidal signal generated within the computational domain, right? If tidal signal is not removed from the open boundary conditions, the no-tide experiment still contains tidal signal generated from outside.*

We have added a discussion of the background current with and without tidal forcing. Please see lines (228-249).

Regarding the question of how tidal forcing is removed in numerical model simulations. Generally, there are two common approaches to generate tidal forcing in ocean numerical models. The first method is through internal tidal potential forcing, where the model calculates tidal motion based on astronomical tidal potential (gravity and Earth-Moon-Sun interactions). This approach is often used in global models. The second, more commonly used method in regional ocean models, is to impose tidal elevation and/or tidal velocity at the open boundaries, based on data from global tidal models such as TPXO, FES, or OTIS. These datasets provide tidal constituents derived from satellite altimetry and other observations. In the model setup

used in this study, tidal forcing was applied at the open boundaries by prescribing tidal elevations based on the TPXO8 dataset.

To remove tidal forcing in the regional model, we simply excluded the tidal elevations at all open boundary grid points. This effectively eliminates the tidal component from the simulation, allowing us to isolate and evaluate the impact of tidal forcing on the regional circulation.

In the case of the Bohai and Yellow Seas, neglecting the astronomical tidal potential (i.e., internal tidal potential forcing) in numerical simulations is generally acceptable. This is because the primary tidal dynamics in this region are overwhelmingly dominated by external tidal forcing from adjacent open oceans, rather than by the local response to the astronomical tidal potential within the model domain. Specifically, the tidal waves entering the Yellow Sea mainly originate from the Pacific Ocean and propagate through the open boundary (e.g., along the East China Sea shelf). As such, the major tidal constituents can be accurately represented by prescribing tidal elevations and/or velocities at the open boundaries using data from global tidal models like TPXO or FES. These boundary conditions are sufficient to reproduce the dominant tidal features and energy in the region.

Therefore, for regional models focused on the Bohai and Yellow Seas, it is common practice to omit internal tidal potential forcing and instead rely solely on boundary-forced tidal constituents. This approach significantly simplifies the model configuration without compromising the accuracy of the simulated tidal dynamics.

Lines 228-249:

*The eastern side of the anticyclonic circulation direction reverses when the tidal forcing is turned off. This effect is related to the tidal forcing impact on the general circulation of the entire Yellow Sea area (Fig. 8). When tidal forcing is considered, this area is dominated by a basin scale anticlockwise gyre in the Yellow Sea at a depth of 25 m. In the eastern part of the Yellow Sea, the main current directions are northward. In the western part of the Yellow Sea, the North Shandong coastal current (NSCC) and the Yellow Sea coastal current (YSCC) are present. Most current directions are southward, except for those located west of 122°E, which are directed westward. This phenomenon is the compensation for surface layer wind transport, which will be discussed in the next subsection. Such observations are in agreement with previous observations and numerical results (Bearsley et al. 1992, Yangagi and Takashi, 1993, Xia 2006). The northward flow in the eastern part of the southern Yellow Sea is a jet-like flow, which is different from the southward flow in the west portion of the southern Yellow Sea; this flow is much weaker and broader. However, when tidal forcing is removed, the overall circulation configuration transitions into a clockwise gyre (Fig. 8). This large-scale circulation change influences the local flow structure around the Qingdao cold water mass. Specifically, the reversal of the eastern branch of the anticyclonic circulation (122.5°-123°E 35°-35.5°N) results from the adjustment of the broad-scale Yellow Sea gyre, highlighting the significant role of background circulation in shaping the local current system.*

*The reason for the background anticlockwise circulation in the Yellow Sea when tidal forcing is considered has been discussed in previous research. (1) In the middle layer of the Yellow Sea (10–40 m), the flow is quasigeostrophic. During spring and summer, strong tidal mixing over the western and central parts of the shelf leads to the formation of a pronounced tidal front, which separates the well-mixed coastal waters from the stratified offshore waters. This front induces strong lateral density gradients, which in turn generate geostrophic currents around the front. This front-associated baroclinic structure promotes the formation of a basin-scale cyclonic (anticlockwise) gyre (C. Xia et al. 2006). (2) The Eulerian residual tidal currents*

*form a cyclonic gyre, implying that these currents strengthen the cyclonic circulation that occurs in the upper layers (C. Xia et al. 2006).*

22. *Line 265. Please be consistent of the difference terms. Based on the caption, Figure 8c is flow differences of no-tide and control runs (no-tide minus control), Figure 8d is flow differences of control and no-wind runs (control minus no-wind). And what does the colored patches represent? Magnitudes? Or directions?*

Sorry for this confusion, we have corrected it. Both are control runs minus no-wind/no-tide runs for the consistency. The color represents the magnitude, and a negative value indicates that the magnitude of velocity is smaller in the control run than the no-tide run. Please see the updated caption of Fig. 6.

23. *Section 3.6. The authors demonstrated that the wind is the dominant driving force for the anticyclonic circulation, which I agree with. But I don't agree with the rest. Firstly, there is a westward shift of the Qingdao cold water mass when wind is removed. It should be pointed out and discussed but the authors did not. Instead, the authors mentioned that the temperature of the cold water decreases by 2oC which is due to the weaker wind mixing when wind is removed. Rather, I think, it should be related to the location changes in cold water mass.*

After changes the wind effects on the horizontal circulation, this part has been deleted already.

24. *Line 309. Based on Figure 12a, the upwelling does not reach the surface but stop at depth around 10m*

We have checked the vertical velocity between 10m to the surface around the rectangle box area. The vertical velocity between 20m to the surface is upward as well. In the previous manuscript, there was a diagram showing the contour of the vertical velocity distribution. This diagram has been removed because the reviewer suggested Fig. 12 has overlapping information with the contour of vertical velocity. But we are open to adding that diagram again.

25. *Lines 312–313. As I observed, the upwelling occurs over the entire transect but with different strength.*

Ok. We have changed the description way. Please see lines 313-315.

*Lines 313-315: Obvious upwelling occurs near the frontal zones of 122.375–122.5°E and 122.625–123°E on the eastern and western sides of the Qingdao cold water mass, respectively.*

26. *Lines 313–315. Yes, it is true. But the convergence is not pronounced. Instead, the most remarkable point is the eastward surface current and the associated secondary westward current at lower layers over the east of the 122E and the strong upwelling system.*

We added a description of convergence and the barotropic and baroclinic pressure gradient comparison between the with- and without- tidal forcing runs, around the obvious upwelling area (lines 329-335).

*Lines 329-335: A comparative analysis between the tidal and non-tidal experiments reveals that both the barotropic and baroclinic pressure gradient forces are significantly intensified when tides are included (Fig. 12). The magnitude of the barotropic pressure gradient force is larger than its baroclinic counterpart. This enhancement in barotropic forcing leads to increased horizontal convergence in nearshore regions, which, through the continuity equation, results in intensified upward motion. Notably, at approximately 122.5°E, surface currents from*

*both the west and east appear to converge (Fig. 11a), as indicated by the opposing surface flow directions. This horizontal convergence is accompanied by a strong upward motion below, supporting the interpretation that tidal forcing enhances barotropic pressure gradients, leading to horizontal convergence and subsequent upwelling in this region.*

*27. Line 317–318. Based on Figure 12a–12b, the upwelling system is strengthened over studied transect but not just the east side when tides are considered, right?*

We have changed the description from the "further strengthening the upwelling intensity on the east side" to "further strengthening the upwelling intensity" (line 319).

*28. Lines 320–321. The westward or eastward flows in the rectangle zone are not the most remarkable point but the upwelling, instead, westward current at lower layers over the east of the 122E may be more interesting.*

The westward and eastward flows are related with the convergence. We are a bit confusion about the comment "westward current at lower layers over the east of the 122°E may be more interesting". Please give us more hints about this point. Thank you.

*29. Line 325. Southeasterly monsoon? The upwelling is contributed by tidal forcings, not by wind forcing? highlight two points: the upwelling system and how it is related to wind and tide. Apparently, the authors did not provide convincible evidence of how tide affect the upwelling here.*

The term "southeasterly monsoon" has been corrected to "southwesterly monsoon." Thank you for pointing this out. Sorry about the misunderstanding that the upwelling is contributed by tidal forcings, not by wind forcing. This misunderstanding likely arose from our wording. We have changed discussion on the tide and wind effects on the upwelling, please see subsection 4.1.

*31. Lines 334–360. I lost here. Instead of believing the tide induce thermal fronts, I would rather believe that the background current system (like the Subei coastal current) is affected by the tides. So, when tides are removed, the background current system change a lot leading to the changes in upwelling and also anticyclonic circulation around the Qingdao cold water mass. Also, I don't know why the authors focus on the relationship between tide and baroclinic pressure gradient force which is a very small term comparing to the barotropic pressure gradient force. Again, the tidal effects can be reflected by changing the background current pattern which, I guess, results in more pronounced changes in the barotropic term than the baroclinic term.*

We have deleted the previous lines 334-360 in the old version.

Previous studies have demonstrated that baroclinic pressure gradient forces can play a critical role in modulating vertical circulation and the evolution of subsurface cold water masses in shallow and seasonally stratified marginal seas like the Yellow Sea. For instance, Xia (2006) and Zhang et al. (2016) showed that tidal mixing significantly alters stratification and induces baroclinic responses, including nearshore upwelling and downwelling circulations.

We have investigated the influence of tidal forcing on upwelling by analyzing the contributions of both barotropic and baroclinic pressure gradients. Please see subsection 4.1.

*32. Figure 13. Please double check the caption.*

Figure 13 has been deleted in the latest version.

*33. Line 369. No. Oceanic modelers use statistic tests a lot.*

We have added a more detailed description in line 379-380: "However, for the ocean regional simulation community, statistical tests are seldom applied to examine the variation between internal and external forcing".

Additionally, we are happy to add reference if the reviewer can give us hints about pervious publications about t-test application in regional seas to distinguish internal variability and external forcings.

*34. Lines 370–380. Is t-test an appropriate test for this study? You only have 4 ensemble members. That is, for example, for the temperature at a given grid point, you are testing if the 4 temperature values in the control run are significantly different (assuming two-side testing) from the 4 temperature values in the no-wind (or no-tide) run. So, sample size is very small. Hope my understanding to your ttest is correct.*

T-test can be used for small samples, even though it would be nice to have more ensemble member. More information about this point can be found in (von Storch & Zwiers, 1999).

*35. Lines 377–380. I don't get it. Could you provide more detailed explanation?*

Ok. We have added more details in lines 386-401.

*Lines 386-401: When such local tests are conducted, it is expected that even if the null hypothesis is valid, at approximately 5% of grid points, the null hypothesis is rejected (multiplicity of tests, cf. von Storch, and Zwiers, 1999). Since the rejection rate is itself a random variable, the false rejection rate can be much larger, but more than 20% is very unlikely. A limitation of univariate tests, such as the t-test, is the problem of multiplicity of tests. This challenge arises when multiple tests are conducted simultaneously across different points in a field without proper adjustment. This issue is discussed in standard textbooks, such as that by von Storch and Zwiers (1999), who built upon previous work (H. V. Storch 1982; Livezey and Chen 1983).*

*The core argument is as follows. For example, if a test has an acceptable false rejection rate (Type I error rate) of 5% when the null hypothesis is true, then repeating the test multiple times while the null hypothesis remains valid will still yield a 5% chance of false rejection in each test. Thus, one would therefore expect false rejections in approximately 5% of the cases on average. However, since the rejection rate itself is a random variable, the actual proportion of false rejections may exceed 5%, although rates significantly higher than 20% are unlikely.*

*The situation becomes more complex when the tests are not independent, such as when analyzing spatially correlated data from a grid. In such cases, nearby grid points exhibit stronger dependencies, suggesting that false rejections are less likely to appear as isolated points and more likely to form spatially coherent patterns.*

*In our analysis, the observed rejection rate is substantially greater than 20% in both scenarios, suggesting that not all rejections can be attributed to the multiplicity effect. Instead, many of these rejections likely reflect genuine signals.*

*36. Lines 394–396. The anticyclonic circulation discussed in this study locate at near bottom, right? The northeastward current induced by summer southwesterly monsoon is at the surface. So, how does this surface current affect the near bottom current system?*

The explanation about the wind impacts on the horizontal circulation has been modified, please see the reply above.

*37. Section 4.3. This section is weak as the authors did not do a great comparison between the finding from this study and the previous studies, e.g., which points agree with the previous findings, and which do not.*

We have expanded the section 4.3 with adding the comparison between previous work results and our results.

*38. Figure A1. The term "evolution" is not expected as only one map is shown.*

Fig. A1 is combined with Fig. 2. In total, it is the map of four month, rather than one-month distribution. We put Fig. A1 in the appendix because it is before the generation of Qingdao cold water mass, and the reviewer suggested that it would be better if we also show the temperature distribution of March.

*39. Figure A3. Blue arrow is for wind while the black for the wind stress, right?*

Yes, the blue arrow is for wind and the black for the wind stress. We have also added one sentence to clarify it.

Reference:

von Storch, H., & Zwiers, F. W. (1999). *Statistical analysis in climate research*. Cambridge University Press.

Xia, C., Qiao, F., Yang, Y., Ma, J., & Yuan, Y. (2006). Three-dimensional structure of the summertime circulation in the Yellow Sea from a wave-tide-circulation coupled model. *Journal of Geophysical Research*, *111*(C11), C11S03. https://doi.org/10.1029/2005JC003218

Xu, D., & Zhao, B. (1999). Existential proof and numerical study of a mesoscale anticyclonis eddy in the Qingdao-Shidao offshore. *Acta Oceanologica Sinica*, *2*, 18–26.

---

## Author Response (AR4)

**General comments:**

The manuscript has been improved; however, several critical issues remain unaddressed. My main concerns lie with the main current patterns in the Yellow Sea, and the effects of wind and tidal forcing on these patterns. The current version lacks a logical, thorough, and convincing analysis of these dynamics. I therefore recommend a major revision.

We would like to thank the editor and reviewers for their time in evaluating our manuscript. We have addressed all the points raised by the reviewers, particularly those that improve the clarity of the manuscript. In cases where we did not accept a suggestion, we have provided a clear justification. In the following, reviewer comments are shown in **black**; our responses are in **blue**.

**Major comments:**

1. What made me confused during the revision for the last two version is the main current patterns in the Yellow Sea in summer. After some literature studies, now, I have some understandings. In Figure 1, the authors show only the YSCC, which is corresponding to the southward current in Figure 8a 122.5E–124E. This is the eastern boundary of the Qingdao cold water mass. However, the northeastward current in the Lu'nan coast (seeing figure below) and the northward current in Subei coastal current were not shown in Figure 1 and discussed thoroughly. These two currents are at the western boundary of the Qingdao cold water mass. The listed 3 currents together generate the summer anticyclonic circulation discussed in this work. Besides, the currents shown in the below figure and the Figure 1 in the manuscript are surface currents, however, the authors aimed to analysis the current patterns at near bottom (25 m). First, the authors should prove that the patterns of these 3 current systems are similar at surface and near bottom. This is what I have pointed out in the last revision. However, in the updated manuscript, the authors still didn't provide detailed analysis on it. Secondly, before the discussion on any effects on the current patterns, the authors should identify and label the current systems in their simulations (i.e., label the current system in Figure 8a). Additionally, the current system in Figure 8a should cover the entire computational domain.

After carefully reviewing the reviewer's concern and re-examining both Fig. 1 and the schematic diagram mentioned by the reviewer, we would like to respectfully point out the following two points: (1) all the currents mentioned in Fig.1a in previous manuscript—the North Shandong Coastal current (NSCC), Yellow Sea Coastal Current (YSCC) and Yellow Sea warm current (YSWC)—are not limited to surface currents (depth<10m), giving our model results and previous studies (Xia et al., 2006; Yu et al., 2010). Therefore, the concern that "Besides, the currents shown in the below figure and the Figure 1 in the manuscript are surface currents, however, the authors aimed to analysis the current patterns at near bottom (25 m)" may not be applicable in this case. (2) We would also like to clarify that our study focuses on the seasonal development of the anticyclonic circulation around the Qingdao Cold Water Mass, which intensifies in spring and typically weakens or disappears by June. This seasonal evolution is consistent with previous observational and modeling studies (Qiu et al. 2025; Huang et al., 2019). However, in the reviewer's comment, the discussion seems to mix the spring and summer circulation regimes, particularly in the sentence: "the listed three currents together generate the summer anticyclonic circulation discussed in this work." Given the strong seasonal cycle of the Yellow Sea circulation, it is important to distinguish between spring and summer patterns.

Even though the two points mentioned above may have been misunderstood, we carefully considered the rest of the reviewer's comment and examined the depth-averaged current structure around the Qingdao Cold Water Mass. The results show a similar anticyclonic circulation pattern as presented in Fig. 3. Therefore, we believe that an additional figure is not necessary in the main text. However, we remain open to including it in the supplementary materials if the reviewer considers it is helpful.

In addition, following the reviewer's suggestion, we have added labels for the northeastward current along the Lu'nan coast and the Subei Coastal Current in Fig. 1a. The simulation results confirm the existence of both currents, with directions consistent with those illustrated in the diagram cited by the reviewer. Please note that the northeastward current along the Lu'nan coast is very close to the shoreline and only appears in shallow waters (depth <10 m), which is why it is not visible in Fig. 8 (25 m depth).

[Figure]

图 1  黄海流系示意图(据文献[6,9,14]重绘)
**Fig. 1  Diagram of coastal current systems in the Yellow Sea(Redrawn after references[6,9,14])**

1. 辽南沿岸流;2. 鲁北沿岸流;3. 青岛—石岛近海的反气旋中尺度涡旋,青岛冷水团;4. 山东南部沿岸的东北向流动;5. 黄海西部沿岸流;6. 苏北沿岸水;7. 夏季苏北沿岸的北向流动;8. 东北向扩展的长江冲淡水;9. 台湾暖流前缘水;10. 黄海暖流;11. 朝鲜半岛西部沿岸流

1. Liaonan coastal current;2. Lubei coastal current;3. Mesoscale anticyclonic eddy in Qingdao-Shidao offshore, Qingdao cold water mass;4. Northeastward current in Lu'nan coast;5. Yellow Sea western coastal current;6. Subei costal current;7. Northward current in Subei coast in summer;8. Northward extension of Yangtze River diluted water;9. Taiwan warm current; 10. Yellow Sea warm current;11. Korean peninsula western coastal current

Figure from Wei et al., (2011)

Wei, Q., Yu, Z., Ran, X., & Zang, J. (2011). *Characteristics of the Western Coastal Current of the Yellow Sea and Its Impacts on Material Transportation*. Advances in Earth Science, 26(2), 145–156. doi: 10.11867/j.issn.1001-8166.2011.02.0145

2. I am lost during the exploration of the wind and tidal effects on the anticyclonic circulation. Based on my understanding in physical oceanography and my careful revision on this

manuscript, I am providing my points of view on how the wind and tides affects the studied anticyclonic circulation.

**Wind effects**. As shown in Figure A3 (please show the wind and wind stress patterns over the entire computational domain), surface currents in the west Yellow Sea are northeastward and eastward. There are at least three effects. (1) Waters piles up on the east side leading to barotropic pressure gradient forces pointing westward. Going down into deeper layers, as the existence of the Qingdao cold water mass, the temperature gradient is pointing westward at the west side of the cold water mass, generating a baroclinic pressure gradient force pointing westward. So, at 25 m depth, both barotropic and baroclinic gradient forces are negative, as denoted by Figure 10b-10c (but I think the authors messed up the signs in these plots). Such **local wind effects** on the deep water will generate a northward current. (2) As the waters move offshore, a basin-wide (at least over west of 123E because over this region, depth is shallower in the west than in the east as shown in Figure 1b) upwelling system will be stimulated. This is the results due to Ekman transports. (3) Summer monsoon in the Yellow Sea and East China Sea should be similarly southwesterly (I am not quite sure if it is true in summer 2019). The currents from the southern boundary are highly affect by East China Sea coastal currents, which are highly affected by the same southwesterly monsoon. This is the **remote wind effects**. The authors, however, did not provide any discussion on it.

Ok. For suggestions (1) and (2), we have shown it in the Section 3.5 and Section 4.1. Regarding the remote wind effects (suggestion (3)), we included it in the outlook, since our model domain does not include the East China Sea. Previous studies used a model domain that did not include most of the East China Sea, however, the models were still able to capture the key features of the Qingdao Cold Water Mass and the surrounding circulation, as confirmed by observational data (Huang et al., 2019; Qiu et al., 2025).

**Tidal effects**. (1) As shown in the Figures 8a-8b, northward coastal currents on the west side of the Qingdao cold water mass are weaken when tides are considered, while the currents (the YSCC) on the east side of the cold water mass turn to northward when tides are turned off. These patterns have been pointed out in the manuscript, which is good. The authors provide discussion on tidal-induced changes in barotropic and baroclinic terms. But what are the specific tidal effects that cause such changes? Residualtidal current? Tidal mixing? Or others? (2) In section 4.1, the authors aimed to discuss the tidal effects on the upwelling. However, discussion on changes in barotropic and baroclinic terms does not answer which tidal processes affect the upwelling system. To my understanding, upwelling systems are compensate current due to the surface water divergence across coastal shelf and the mass balance, but not due to the changes in barotropic and baroclinic conditions at deep layers. Instead, alike the upwelling, the changes in barotropic and baroclinic conditions at deep layers are also the results of changing surface current patterns.

Regarding suggestion (1), the reviewer confirmed our result that the eastside of the anticyclonic circulation turns northward when the tides are turned off. The reversal of the current direction on the east side of the anticyclonic circulation when tides are turned off is linked to the broader-scale Yellow Sea gyre. Unlike the circulation around the Qingdao cold water mass, the tidal effects on the broader-scale Yellow Sea have been studied extensively in previous research (He et al., 2022; Xia et al., 2006). These studies indicate that the strong tidal mixing over the western and central parts of the shelf leads to the formation of a pronounced tidal front. This front induces strong lateral density gradients, which generate currents around the front. Additionally, the residual tidal currents strengthen the cyclonic circulation.

Our model results support this explanation, showing an anticyclonic circulation around the front area in the Yellow Sea (Fig. 8). However, to avoid redundancy with previous publications, we addressed and summarized the effects of tidal mixing and residual currents in section 3.4 (mentioned but not emphasized in the previous version) and focused more on the seasonal anticyclonic circulation around the Qingdao Cold Water Mass, which has not been discussed thoroughly in the literature.

Regarding suggestion (2), contrary to the reviewer's assertion, many studies show that Yellow Sea upwelling is largely tide-driven via baroclinic mechanisms. In particular, Lü et al., (2010) and Sun et al. (2022) demonstrate that strong tidal mixing over sloping topography creates a sharp bottom density front, producing a large cross-front baroclinic pressure gradient that forces an onshore (upwelling) branch of circulation. This "secondary circulation" occurs when the tilted isopycnals at the tidal front induce a pressure gradient near the bottom, drawing deep water upward along the shelf. Numerical simulations confirm that in Yellow Sea tidal-front zones, upwelling is mainly caused by baroclinic processes associated with tidal mixing. Thus, the compensating flow (upwelling) is directly linked to the baroclinic pressure differences generated by tides, not only to the changes of surface current patterns.

The upwelling around the Qingdao Cold Water Mass has not been studied previously. We therefore address the existence of upwelling (lines 310–323, 324–326) and explain it in terms of changes in front intensity (lines 343–354) caused by tidal forcing (lines 355–367). This explanation is consistent with previous research. We also restructured and expanded the explanation to address the reviewer's concerns (lines 361–365).

**Detailed comments:**

1. Figure 1a. Please provide a reference for this plot. It seems like from this work:

Liu, Shichu, et al. "Interannual variation in winter thermal front to the east of the Shandong Peninsula in the Yellow Sea." Journal of Sea Research 193 (2023): 102370.

Ok, added.

2. Line 176-177. Please show evidence to support the baroclinic effects on the differences in current patterns along vertical direction. If not, please remove this sentence.

We have removed the sentence.

3. The arrows in Figure 5 are hard to see.

We have updated a new high-resolution diagram.

4. Line 195-196. The geostrophic balance is hardly violated in the cold water area (122E-123E, 34.5N-36N), which is shown in Figure 5d and is also mentioned in Line 239.

If I understand correctly, the reviewer would like to address that geostrophic balance is maintained in $122-123°E$, $34.5-36°N$. However, this paper we focus on Qingdao cold water mass area, which is outside of $122-123°E$, $34.5-36°N$. We reformulated lines 195-196 to clarify our expression.

5. Line 213-214. Should be "between the control and the no-tide experiment (Fig. 6)"

Ok.

6. Line 219. As shown in Figure 8a and 8b, the velocity is greater in the no-tide experiment than in the control ones.

Ok.

7. Line 221. Without tidal forcing east of $122°E$, the magnitude of the barotropic term also increase.

Ok. After reformulating the lines 220-225, this sentence has been deleted in the new version.

8. Line 219-214. I don't understand what the main conclusion or the main purposes of this paragraph is. I lost here. Please see the major comments 2 for the tidal effects.

It seems that the line numbers (219–214) are not in ascending order. We guess that it maybe refers to lines 219-224. If so, we have tried to conclude the main message of this paragraph in the first sentences.

9. Lines 224-237. Seems that the authors try explain why the YSCC is reversed when tides are off, but I didn't find the logic of for this paragraph and cannot come up with a clear conclusion.
Let me clarify my confusion.
On line 226-227, the anticlockwise gyre is the one around 124-126E, 34.5-26.5N. It is beyond the studied clockwise circulation around the Qingdao cold water mass.

Line 228-230, the norward current west of 122E is not the compensation currents but the northeastward current in the Lu'nan coast and the northward current in Subei coastal current. Please correct me if I am wrong.

Line 231-233, the "northward flow in the eastern part of the southern Yellow Sea" is around 125E, while the "southward flow in the west portion of the southern Yellow Sea" (or the YSCC) is at around 122-124.5E. Do they have any linkages when you compare them?

All the above changes are the so-called changes in broader-scale Yellow Sea circulation. But they still not answer what tidal processes affect changes in broader-scale circulation, and then affect the YSCC. Although the authors list reasons in 238-245 citing other works. But what the authors get from their simulation are still not well addressed.

Regarding the circulation structure, please see our response to Major Comment 1. We have described what we learned from the cited works in lines 240–250.

10. Line 243, where is the mentioned "basin-scale cyclonic gyre"?

We have added "in the Yellow Sea" after the "basin-scale cyclonic gyre" to make it more clear.

11. Lines 284-289. The analysis conflict with the signs shown in Figure 10.

Corrected.

12. Figure 10. Please double check the signs. For example, the positive (eastward) barotropic term in Figure 10b conflict with the westward barotropic term shown in Figure 5b.

Please refer to our response to Minor Comment 11 above.

13. Section 4.1. Please see the major comment 2 related to the discussion of upwelling. As shown in Figures 11a and 11b, surface currents (0-10 m) are eastward moving waters offshore. I would insist that the authors should focus on the adjustment of basin-scale mass balance rather than on the changes in barotropic and baroclinic terms over a transect or a profile (Figure 12) when linking the tidal effects and the upwelling system.

Please see the reply to major comment 2.

14. Line 345-346. My understanding on oceanic front is that front is determined by horizontal gradients of temperature or salinity or density, but not based on gradient on the x-z panel, especially for the Qingdao cold water mass, which has strong horizontal temperature gradient.

The front (in the horizontal direction) is present in our model results, and tidal forcing influences both its shape and intensity. Evidence of the horizontal temperature front has already been provided in our previous responses to the reviewers' comments. In the current figure, we focus on the x–z section to better illustrate the upwelling structure. However, we are happy to include an additional horizontal temperature plot to show the existence of the front if needed.

15. Line 354. As shown in Figures 12a -12b, the vertical friction is near zeros. So, at this location, pressure gradient forces balance with the Coriolis force, which means that the geostrophic balance is met.

In the previous version of the manuscript, we included a vertical distribution of zonal momentum terms along the 35.5°N profile. From that diagram, it was clear that geostrophic balance is not maintained throughout the entire water column. Reviewer 3 agreed with this conclusion after reviewing the figure. However, in the last round, Reviewer 3 suggested deleting the diagram to avoid redundancy. This comment seems to contradict the earlier feedback. To clarify: although geostrophic balance holds approximately at about 15 m depth in a 25 m water column, it breaks down in the deeper layers due to friction. Therefore, conditions at 15 m (the middle layer) cannot represent the full vertical momentum distribution.

16. Figure 12. Why do you show the momentum at depth=15 m when the rest of the analysis is based on the currents pattern on depth =25m?

This is because Fig. 12 describes the momentum balance around the upwelling region, where the maximum upwelling occurs at approximately 15 m depth. The rest of the analysis addresses different scientific questions—specifically, whether the geostrophic balance is maintained around the Qingdao Cold Water Mass, and the respective influences of tidal and wind forcings on the seasonal anticyclonic circulation in that region.

17. Lines 374-375. Distinguish instead of extinct? This is an incomplete sentence.

Corrected.

18. Line 381-382. As I observed on Figures 13-14, 51.29% is for no-tide conditions, while 89.68% for no-wind conditions. Also, rounding to integer is enough.

Corrected.

19. Figure 13-14, are they for depth=25 m? Please update the captions.

Ok.

20. Lines 424-426. The geostrophic balance maintains over the cold water mass.

This comment contradicts Reviewer 3's remarks from previous rounds. As discussed during the previous two rounds of revisions with Reviewer 3, the geostrophic balance does not hold over the Cold Water Mass, and Reviewer 3 agreed with this conclusion. We also deleted part of the geostrophic balance analysis as suggested by Reviewer 3. Additionally, our results on the geostrophic balance show that, around the Qingdao Cold Water Mass, the geostrophic balance is not maintained.